# The North Atlantic mean state in mesoscale eddy-resolving coupled models: a multimodel study

Amanda Frigola[1], Eneko Martin-Martinez[1], Eduardo Moreno-Chamarro[1,2], Margarida Samsó[1], Saskia Loosvelt-Tomas[1], Pierre-Antoine Bretonnière[1], Daria Kuznetsova[1], Xia Lin[3], Pablo Ortega[1]

[1]Barcelona Supercomputing Center, Barcelona, Spain
[2]now at Max Planck Institute for Meteorology, Hamburg, Germany
[3]School of Marine Sciences, Nanjing University of Information Science and Technology, Nanjing, China

*Correspondence to*: Amanda Frigola (amanda.frigola@bsc.es)

**Abstract.** Ocean mesoscale structures, which are parameterized in models with standard resolutions on the order of 1º or coarser, have an impact at larger scales, affecting the ocean mean state and circulation. Here we study the effects of increasing model ocean resolution to mesoscale eddy-resolving scales on the representation of the North Atlantic mean state, by comparing an ensemble of four HighResMIP coupled historical simulations with nominal ocean resolutions of at least 1/10º – corresponding to the models CESM1-CAM5-SE-HR, EC-Earth3P-VHR, HadGEM3-GC31-HH, and MPI-ESM1-2-ER – to a baseline of 39 Coupled Model Intercomparison Project phase 6 (CMIP6) simulations at coarser resolution. We find an improved representation of the Gulf Stream (GS) structure and position in the mesoscale-resolving ensemble, which leads to significantly reduced surface temperature and salinity biases north of Cape Hatteras (NCH). While higher resolution lessens the mean cold–fresh surface biases in the Central North Atlantic (CNA), the improvement is not statistically significant, as some mesoscale-resolving models still present an overly weak North Atlantic Current (NAC). Important differences also occur in the Labrador (LS) and western Irminger Seas (IS). Although the mesoscale-resolving ensemble exhibits larger warm and salty local biases at the surface compared to the low-resolution one, its full-depth profile reveals significantly weaker vertical stratification in the area, closer to observations. This reduced stratification in the high-resolution ensemble is consistent with the presence of stronger (although not significantly stronger) deep water convection in the region. While in the LS the wide range of MLD observational estimates makes model assessment challenging, in the Nordic Seas and along the East Greenland Current, convection in the high-resolution model ensemble is in better agreement with observational records, compared to the low-resolution ensemble. Another clear improvement in the mesoscale-resolving ensemble is found for the representation of the Atlantic overturning in depth-space, which is significantly closer to RAPID observations at 26.5º N than in the low-resolution counterpart; however, it still remains too shallow compared to observations and reanalyses. The subpolar gyre (SPG), as characterized by the barotropic streamfunction, is not significantly stronger in the higher resolution ensemble, although it presents a narrower and locally stronger boundary current.

# 1 Introduction

The North Atlantic is a key region with multiple impacts on the global climate system. One of its main roles is the redistribution of heat from low to high latitudes through the Atlantic Meridional Overturning Circulation (AMOC). At 26.5º N the Atlantic Ocean transports ~1.2 PW of heat, which represents ~60–65 % of the combined contributions from the Atlantic and the Pacific at those latitudes (Ganachaud and Wunsch, 2000; Johns et al., 2023; Lumpkin and Speer, 2007; Trenberth and Fasullo, 2017). Indeed, heat transport by the AMOC explains the milder temperatures in the Northern Hemisphere compared to the Southern Hemisphere (Buckley and Marshall, 2016). The North Atlantic is also an important anthropogenic carbon sink, contributing to reducing atmospheric global warming (Brown et al., 2021). This region exhibits the highest global uptake rate of anthropogenic carbon per area, which is related to enhanced vertical penetration via the AMOC upper cell (Gruber et al., 2019).

Changes in the AMOC in the past have been associated with abrupt changes in climate (Ng et al., 2018), and climate projections indicate consistent AMOC weakening at increased $CO_2$ levels (Jackson et al., 2020), with important effects upon climate, such as Northern Hemisphere drying and cooling, and a southward shift in the intertropical convergence zone (ITCZ; Bellomo and Mehling, 2024; Liu et al., 2020). Thus, considering its fundamental role within the climate system, the dynamics of the North Atlantic need to be appropriately represented in climate models, in order to trustfully evaluate the future impacts of climate change.

The North Atlantic circulation is influenced by a series of elements and processes that are strongly interconnected. The strength and path of the North Atlantic Current (NAC) affect the heat and salinity content of the waters reaching the subpolar North Atlantic (SPNA; Marzocchi et al., 2015). There, the relatively warm and saline AMOC upper limb waters undergo a process of densification associated with 1) surface water mass transformation through air-sea buoyancy fluxes (Petit et al., 2020; Jackson and Petit, 2023) and 2) mixing with denser (colder) waters from the Greenland–Scotland Ridge overflows (Dey et al., 2024), together triggering deep water convection in the SPNA basins (Koenigk et al., 2021). Sinking of deep waters forming the AMOC return flow occurs at the boundaries of the subpolar gyre (SPG) and has been associated with densification of waters along the boundary current (Katsman et al., 2018; Straneo, 2006; Spall and Pickart, 2001).

The horizontal resolution of ocean models is crucial for accurately representing these processes. A realistic bathymetry is key in characterizing ocean throughflows and their properties, in particular those of the Greenland–Scotland Ridge (Katsman et al., 2018; Dey et al., 2024). Additionally, ocean horizontal resolution determines the representation of mesoscale (and submesoscale) features, including ocean eddies. These structures impact the ocean dynamics through their fundamental role in the transport of heat and salt (Sun et al., 2019; Treguier et al., 2012). Ocean eddies' average horizontal scale is smaller at high latitudes, continental shelves, and areas of weak stratification (Hallberg, 2013). Models with ocean resolutions of at least 1/10º are known as mesoscale eddy-resolving models and are capable of resolving mesoscale eddies in extensive areas of the North Atlantic – with limitations in regions of weak stratification or shallow bathymetry. Mesoscale eddy-permitting models, by contrast, have resolutions in the order of 1/4º and are only able to resolve mesoscale eddies in the tropics. In models with

ocean resolutions of 1º or coarser, the contribution of eddies is parameterized instead. In the SPNA, ocean eddies contribute to the downwelling of deep waters along the boundary current of the SPG through advection of density and vorticity from the interior basins (Straneo, 2006; Brüggemann et al., 2017). They also contribute to restratification of Labrador Sea (LS) convective areas at the end of winter (Clément et al., 2023).

Associated with a better characterization of the ocean mesoscale, increasing the ocean resolution to mesoscale eddy-permitting scales has been shown to improve the representation of boundary and frontal currents, such as the Gulf Stream (GS) and NAC, both in terms of location and structure, with significant further improvement at mesoscale eddy-resolving scales (Hewitt et al., 2017; Marzocchi et al., 2015). This better characterization of the GS and NAC leads to reduced sea surface temperature (SST) and salinity (SSS) biases north of Cape Hatteras (NCH) and in the Central North Atlantic (CNA) at mesoscale eddy-resolving scales (Chassignet et al., 2020; Marzocchi et al., 2015). These reduced surface temperature biases are reflected in the atmosphere mean state as well: the winter stormtrack bias generally present at high latitudes in eddy-parameterized models is reduced at mesoscale eddy-resolving scales, associated with a weaker meridional temperature gradient in the North Atlantic (Moreno-Chamarro et al., 2025); and also the local negative bias in precipitation associated with the cold CNA bias is reduced at high resolution (Moreno-Chamarro et al., 2022).

Increasing ocean resolution to (at least) mesoscale eddy-permitting scales has also been shown to improve air-sea interactions in the North Atlantic. More specifically, mesoscale eddy-permitting models, when run together with an atmospheric component of equivalent resolution, exhibit more realistic near surface wind stress divergence and curl fields over the GS and NAC compared to eddy-parameterized models (Tsartsali et al., 2022). Similarly, the representation of the covariance between SST and heat fluxes is improved in that region at mesoscale eddy-permitting scales (Bellucci et al., 2021).

Submesoscale processes also have an impact on the North Atlantic mean state. Tagklis et al. (2020) show a significant reduction in deep water convection in the LS (and an increase in vorticity) when increasing the grid resolution in a regional model from 15 km (mesoscale-permitting in the LS) to 1 km (submesoscale-resolving in the LS). That study finds that the simulated reduction in convection is caused by eddy heat advection from the Irminger Current and by local submesoscale eddy buoyancy fluxes from the LS basin itself. Similarly, restratification of the LS convective areas at the end of winter has been associated with both mesoscale and submesoscale eddies (Clément et al., 2023). Another example of the importance of the submesoscale in the representation of the North Atlantic dynamics is a further eastward penetration of the NAC and its eddy variability at 1/50º resolution, in closer agreement with observations, compared to mesoscale-resolving scales (Chassignet and Xu, 2017). Omitting submesoscale eddies contributions might thus imply biases in the representation of the NAC and deep water convection. Current computational resources allow for multidecadal global coupled runs at mesoscale-resolving resolutions, so that future research efforts should be aimed at parameterizing submesoscale-related processes to the extent possible.

This study has its focus on the North Atlantic, and aims at assessing the impact of explicitly resolving mesoscale ocean eddies in the representation of its mean state in historical simulations, by comparing an ensemble of four Coupled Model Intercomparison Project phase 6 (CMIP6) HighResMIP (Sect. 2.1; Haarsma et al., 2016) coupled mesoscale eddy-resolving

models – namely CESM1-CAM5-SE-HR (Chang et al., 2020), EC-Earth3P-VHR (Moreno-Chamarro et al., 2025), HadGEM3-GC31-HH (Roberts et al., 2019), and MPI-ESM1-2-ER (Gutjahr et al., 2019) – with a second ensemble of 39 CMIP6 coupled non-eddy-resolving models. This work focuses on describing the dynamics of the North Atlantic, as well as the properties that impact them, such as the biases in temperature and salinity, and the vertical stratification in key deep ocean convection areas.

Although a wide range of multimodel studies considering coupled climate models in a North Atlantic context have been published (e.g. Reintges et al., 2024; Jackson and Petit, 2023; Bellomo et al., 2021; Heuzé, 2021; Roberts et al., 2020; Koenigk et al., 2021), only a few of them include mesoscale eddy-resolving simulations (e.g. Koenigk et al., 2021; Roberts et al., 2020) and none of them specifically addresses the impact of resolving mesoscale ocean eddies. In that context, our study stands out for its particular focus on the added value of these eddies, featuring the largest ensemble of coupled mesoscale eddy-resolving simulations considered so far. This ensemble allows us to evaluate more consistently which aspects of the mean climate are improved at that resolution.

This manuscript is structured as follows: the data and methodological approach employed are described in Sect. 2. The main results of the study are presented in Sect. 3, including a characterization of SST and SSS biases in the North Atlantic for the high and low resolution ensembles (Sect. 3.1); the stratification (Sect. 3.2) and mixing in the regions of deep water formation (Sect 3.3); the AMOC streamfunction (Sect 3.4); and the gyre circulations, including the NAC and the SPG (Sect. 3.5). Additionally, in Sect. 3.6, the significance of the differences between the high and low resolution model ensembles is tested using a bootstrap analysis; specific characteristics of the individual high resolution models are analysed in Sect. 3.7; and relations between the previously analysed dynamical and physical properties are investigated in Sect. 3.8. Finally, in Sect. 4 we delve deep into the discussion of the main results, relate them to the current literature, and present our conclusions.

## 2 Data and methods

### 2.1 Model data and methodological approach

In order to assess the impact of increased horizontal resolution on the representation of the North Atlantic mean state, we analyze the outputs from four CMIP6-endorsed HighResMIP (Haarsma et al., 2016) coupled mesoscale eddy-resolving historical simulations (hist-1950; HR-HIST hereinafter) – corresponding to the models CESM1-CAM5-SE-HR (Chang et al., 2020), EC-Earth3P-VHR (Moreno-Chamarro et al., 2025), HadGEM3-GC31-HH (Roberts et al., 2019), and MPI-ESM1-2-ER (Gutjahr et al., 2019) – and compare them to a baseline ensemble of 39 CMIP6 coupled historical runs (Eyring et al., 2016) performed at coarser resolution (LR-HIST hereinafter). We reckon this standard resolution ensemble as a more rigorous benchmark than the low-resolution HighResMIP counterparts of the four eddy-resolving models, given its much larger size. More details on the models considered are provided in Tables 1 and B1.

| HR-HIST models | ocean component | ocean grid | atm. component | atm. grid | reference |
|---|---|---|---|---|---|
| CESM1-CAM5-SE-HR | POP2 | 1/10°; tripolar; 3600x2400 lon/lat; 62 levels; | CAM5.2 | 25 km; 30 levels; | Chang et al. (2020) |
| EC-Earth3P-VHR | NEMO3.6 | 1/12°; ORCA12 tripolar; 4322 x 3059 lon/lat; 75 levels; | IFS cy36r4 | 16 km; 91 levels; | Moreno-Chamarro et al. (2024) |
| HadGEM3-GC31-HH | NEMO-HadGEM3-GO6.0 | 1/12°; eORCA12 tripolar; 4320 x 3604 lon/lat; 75 levels; | MetUM-HadGEM3-GA7.1 | 50 km; 85 levels; | Roberts et al. (2019) |
| MPI-ESM1-2-ER | MPIOM | 1/10°; TP6M tripolar; 3602 x 2394 lon/lat; 40 levels; | ECHAM6.3 | 103 km; 95 levels; | Gutjahr et al. (2019) |
| LR-HIST models | – | 25–250 km (mainly 100 km) | – | 100–500 km (mainly 100–250 km) | – |

**Table 1:** Overview of models used in the current study. Ocean grid details include: nominal resolution; grid type; size of horizontal grid; and number of vertical levels. Details about the individual LR-HIST models can be found in Table B1 in Appendix B.

All models in the HR-HIST ensemble have a nominal ocean resolution of at least 1/10º (~10 km), allowing them to represent the ocean mesoscale in extensive areas of the North Atlantic. By contrast, in LR-HIST, the ocean mesoscale is, at best, only resolved in the tropics and more generally parameterized, since ocean resolution in that ensemble ranges from 25 to 250 km, being 100 km the most common resolution across models. Atmospheric resolution is also generally higher in the HR-HIST ensemble compared to LR-HIST, ranging from 15 to 100 km for HR-HIST, and from 100 to 500 km for LR-HIST (Tables 1 and B1). We note the heterogeneity in model components employed across the ensembles, which might avoid a dominant contribution of specific individual model biases within the ensembles.

Model selection criteria is based on the availability of three-dimensional temperature and salinity, and the overturning mass streamfunction (either msftmz or msftyz) for the Atlantic Ocean as output variables. Although in most ocean grids the y-grid direction might differ from the meridional direction at high northern latitudes, in general the overturning mass streamfunction calculated along lines of constant y (msftyz) provides a good approximation of that calculated along lines of constant latitude (msftmz) (Griffies et al., 2016). In our study, we select msftmz over msftyz when available. The models for which only msftyz is available as output variable (marked with an * in Fig. 9) present significant grid rotation only from ≥ 40º N northwards (except for MPI-ESM1-2-ER, which presents rotation already at 30º N). For caution, we restrict our analysis to latitudes below 40º N, which nevertheless allows us to extract valuable information. Some of the LR-HIST models, as downloaded from the Earth System Grid Federation ESGF data portal, present unrealistic values for the barotropic streamfunction (BSF), reflecting problems in the integration and/or different integration approaches (not shown). Such models have been discarded from our analysis. No significant offset in the BSF (including in the Southern Ocean region) is observed between the models kept in our BSF analysis and they all present a realistic BSF structure in the North Atlantic. Differences originating from different integration assumptions are thus expected to be small in our restricted model sample.

Forcing fields in the HighResMIP coupled historical simulations (HR-HIST ensemble) are almost identical to those in the CMIP6 historical simulations (LR-HIST ensemble). The only significant difference concerns land use, which is fixed in time in HighResMIP and representative of the present-day period (around year 2000) (Haarsma et al., 2016), and time-varying in CMIP6 historical simulations (Eyring et al., 2016). We do not expect this to cause important differences in the ocean variables considered in our analysis. It is also worth noting that HighResMIP historical simulations are run without interactive aerosols, but this is also the case for several CMIP6 historical simulations (e.g. CMCC-CM2-HR4, EC-Earth3, FGOALS-f3-L, GISS-E2-2-G, IPSL-CM6A-LR, MPI-ESM1-2-HR).

More significant differences concern model initialization and spin-up. Due to the high computational costs of high-resolution modelling, in HighResMIP, initial conditions for historical runs are taken from a short spin-up (~30–50 years) with fixed 1950's radiative forcings and ocean initial conditions (Haarsma et al., 2016), instead of from a long pre-industrial control representative of 1850 conditions. Thus, in the case of the HighResMIP experiments, the substantially shorter spin-up and historical period covered (1950–2014) can leave some lingering drifts. Nevertheless, in some mesoscale eddy-resolving HighResMIP simulations, the ocean seems to equilibrate faster (Moreno-Chamarro et al., 2025; Roberts et al., 2019) compared to their lower resolution counterparts.

Data analyses are carried out using the Earth System Model Evaluation Tool (ESMValTool v2.10.0), a Python package designed for model intercomparison purposes (Andela et al., 2023a, b; Righi et al., 2020). We note that the CMIP6 model ICON-ESM-LR was excluded from some of the analyses due to an incompatibility of the data with ESMValTool. Climatologies of monthly means are computed for the last 35 years of the historical runs (1980–2014) to reduce the potential effect of model drifts, which are expected to be larger in the earlier years of the HighResMIP simulations.

Individual model data are plotted on the original grids provided in the Earth System Grid Federation (ESGF) portal (e.g. in Figs. 7, 9, 12, or in the vertical profiles of individual models in Figs. 5 and 11a). Instead, when calculations of multi-model means or model bias metrics are required, model data are regridded onto the corresponding observations/reanalysis grid or onto a common regular grid (in Fig. 8). Regridding methods employed are linear/bilinear (for one-/two-dimensional data, respectively; Figs. 5–6, 10–11, and 13) and nearest neighbour interpolation (closest source point; Figs 1–4, and 8). We note that each Fig. is produced using one and only one interpolation scheme. In all cases, a visual comparison between regridded and original data is performed, to ensure the suitability of the interpolation scheme.

## 2.2 Specific diagnostics

Potential density anomalies with respect to a reference pressure of 0 dbar ($\sigma_0$) are calculated from temperature and salinity monthly means, using the polynomial approximation of the TEOS-10 equation of state for Boussinesq models (Roquet et al., 2015). Mixed layer depth (MLD) is defined and calculated as the shallowest depth level at which monthly potential density $\sigma_0$ exceeds by a threshold of 0.03 kg m$^{-3}$ its value at a reference depth of 10 m, as described in de Boyer Montégut et al. (2004).

This method is preferred over employing direct MLD model outputs that use instantaneous values and a range of different
definitions, to ensure a consistent comparison across models and observations.

## 2.3 Observational references

Observational and reanalysis data are employed to evaluate model performance. For temperature and salinity, the Met
Office Hadley Centre EN.4.2.2 dataset (EN4; Good et al., 2013) with the Gouretski and Reseghetti (2010) expendable
bathythermograph and Gouretski and Cheng (2020) mechanical bathythermograph corrections is used, which has a resolution
of 1º. We opt for this three-dimensional dataset to jointly assess biases at the surface and depth. We also use it to derive an
observational reference for the MLD that is physically consistent with EN4 salinity and temperature fields. Additionally,
temperature and salinity data from the observational analysis ARMOR3D (Guinehut et al., 2012) at a resolution of 1/4° are
employed to complement the EN4-derived MLD estimates. The increased resolution of the ARMOR3D dataset compared to
EN4 allows for a detailed characterization of the MLD at the boundary current of the SPG. We note though that EN4 data
spans the entire historical period covered in our analysis (i.e. 1980–2014), while ARMOR3D data only covers from 1993
onwards, which might introduce some temporal effects in the derived climatologies.

For the Atlantic overturning mass streamfunction and barotropic streamfunction, ORAS5m reanalysis data (Tietsche et al.,
2020) are used as a first reference for model validation (1/4° resolution; period 1980–2014). ORAS5m is an improved version
of the 5th ECMWF ocean reanalysis system ORAS5 (Zuo et al., 2019), with reduced SST nudging and increased weight to
coastal observations. This version improves the representation of the AMOC streamfunction and leads to reduced biases in
winter reforecasts of the North Atlantic. As a second reference, in addition to ORAS5m, GLORYS12 reanalysis data at
mesoscale eddy-resolving resolution (1/12º; period 1993–2014; Lellouche et al., 2018) are also employed to add robustness to
our analyses. In the case of ORAS5m, the overturning and barotropic streamfunctions are calculated from velocity fields in
the original reanalysis ocean model grid, while for GLORYS12, they are calculated based on the regridded velocity fields
available from Copernicus. We note that the use of regridded fields for volume transport calculations might introduce some
errors related to, for example, the estimates of grid cell areas. However, the GLORYS12 data still constitute a valuable
qualitative reanalysis reference. As a complementary reference of direct observational data, the climatological vertical profile
of the RAPID array is employed to validate the simulated Atlantic overturning streamfunction at 26.5° N. RAPID is a
monitoring programme providing time series of AMOC based on temperature, salinity and pressure profiles from a mooring
array crossing the Atlantic from west to east at 26.5º N (Johns et al., 2023; Moat et al., 2023). The climatology employed
corresponds to the period April 2004 – December 2014.

Monthly averaged absolute dynamic topography data (sea surface height above geoid) are also employed from AVISO
observations at 1/4° resolution for the period February 1993 – December 2014.

## 2.4 Bootstrapping analysis

In order to test the statistical significance of the differences in means between the HR-HIST and LR-HIST ensembles for the different climatologies analysed in the manuscript, we apply bootstrapping to different single number metrics, such as, the temporal mean of maximum (max.) Atlantic overturning strength at 26.5º N, SPG strength, or spatially averaged surface biases over specific regions (see Sect. 3.6 and first column in Table A1 for more details). More specifically, significance is assessed by calculating the 95% confidence interval (CI) of the distribution of the differences in means between the two ensembles. A description of the bootstrapping algorithm is provided right below: (i) first, a single number metric, such as the climatological max. Atlantic overturning strength at 26.5º N is selected; (ii) then, one random sample of models from the HR-HIST ensemble and one from the LR-HIST ensemble are selected for that specific metric, allowing for model repetition (replacement) within the samples; (iii) the multi-model mean values of that specific metric for the HR-HIST and LR-HIST samples are calculated separately; (iv) then, the difference between those two mean values (HR-HIST sample mean - LR-HIST sample mean) is calculated; (v) subsequently, steps (ii)–(iv) are iterated to get $10^4$ samples, obtaining a distribution of differences in means; (vi) finally, the 95% CI of that distribution is calculated. If the CI obtained does not contain the value zero, the difference in means is considered significant.

As a first analysis, bootstrapping is applied using maximum ensemble sizes in both the LR-HIST and the HR-HIST samples, i.e. by taking LR-HIST ensemble samples with the same cardinality as the LR-HIST ensemble (card(LR-HIST)), and HR-HIST ensemble samples with the same cardinality as the HR-HIST ensemble (card(HR-HIST); second column in Table A1). Subsequently, a second analysis is performed after reducing the size of the LR-HIST ensemble samples to card(HR-HIST), which is considerably smaller than card(LR-HIST) (third column in Table A1). This second analysis is aimed at investigating whether the differences in means between the HR-HIST and LR-HIST ensembles are still significant when the LR-HIST ensemble is considered as subsamples of size card(HR-HIST), or whether, on the contrary, there is a significant amount of subsets of models of size card(HR-HIST) within the LR-HIST ensemble that are similar to the HR-HIST ensemble in terms of the analysed metric.

If the CI obtained in this second analysis does contain the zero value, which means that the LR-HIST ensemble is not significantly different from the HR-HIST ensemble when the former is conceived as subsets of size card(HR-HIST), the analysis is repeated by gradually increasing the size of the LR-HIST ensemble samples, until a CI falling entirely to the right (or to the left) of zero is obtained. This allows us to determine the minimum size of the LR-HIST ensemble samples required for the difference in means between the ensembles to become significant (last column in Table A1): the larger the value of the required LR-HIST sample sizes, the closer the two ensembles are in terms of the analysed metric, and the larger the degree of intersection between the two ensembles (see examples in Sect. 3.6).

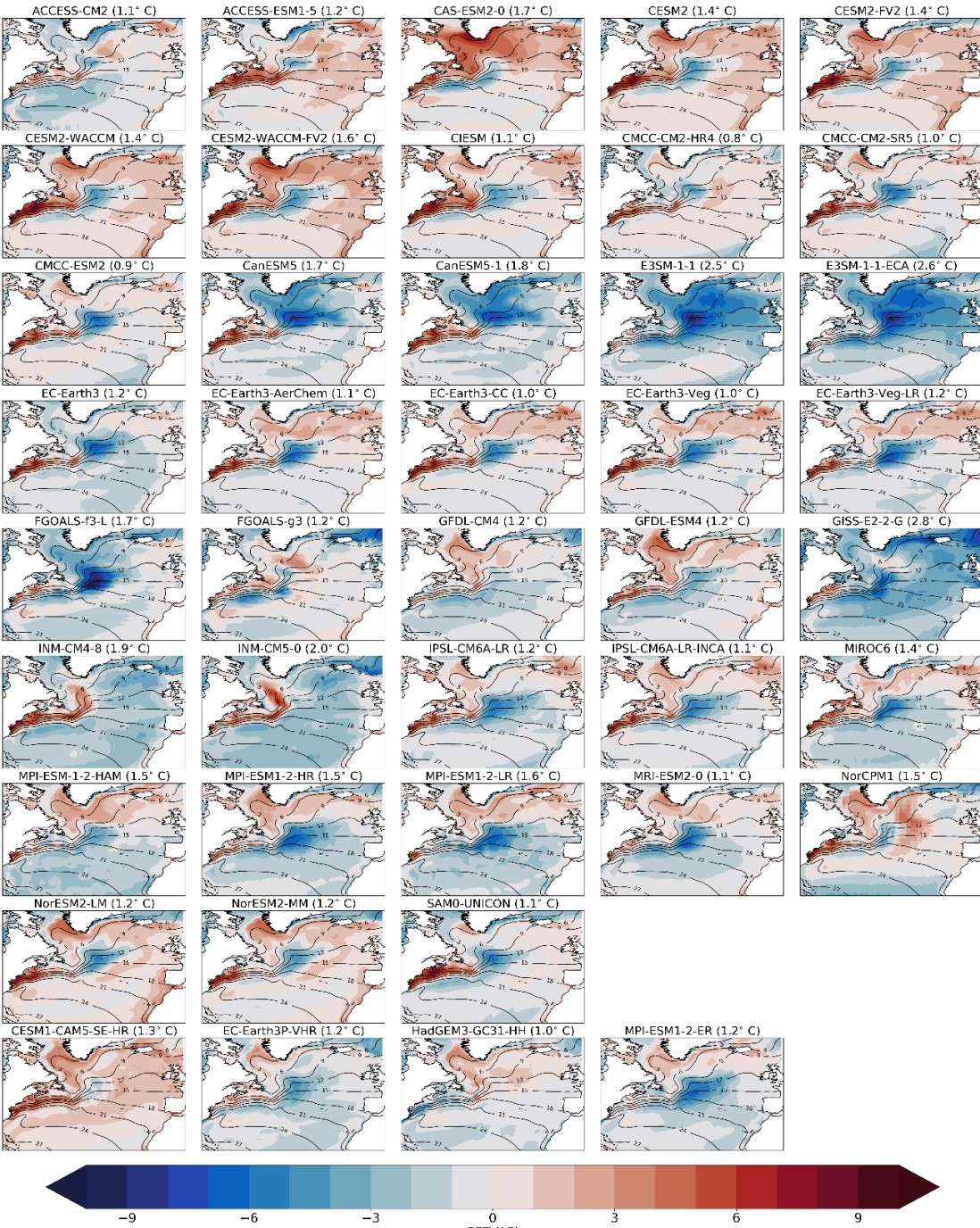

**Figure 1.** Sea surface temperature (SST) bias (shading; in ºC) for the individual LR-HIST (rows 1–8) and HR-HIST (last row) models with respect to EN4, for the period 1980–2014. EN4 climatology shown in contour lines (in ºC). Values in parenthesis in each subfigure header show the spatially averaged absolute mean bias for each individual model.

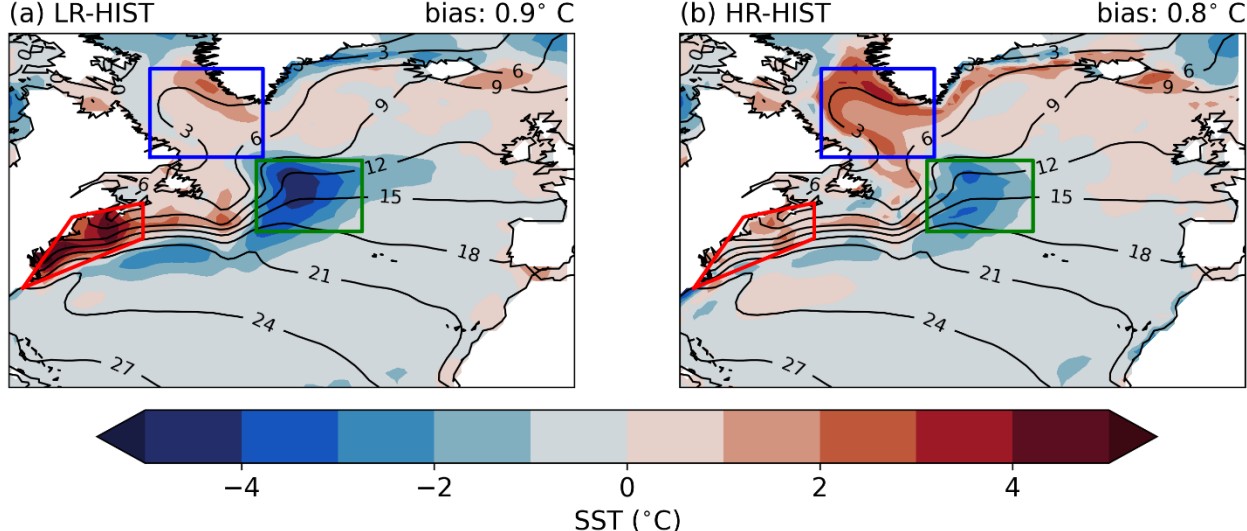

**Figure 2.** SST bias for the multi-model mean of the (a) LR-HIST and (b) HR-HIST ensembles. Plotting details as in Fig. 1. Coloured polygons delineate the main bias regions addressed in the manuscript: north of Cape Hatteras (NCH) in red [edges: (78º W, 34º N), (61º W, 41º N), (61º W, 46º N), (71º W, 44º N)], Central North Atlantic (CNA) in green (30º–45º W, 42º–52º N), and Labrador Sea (LS) in blue (44º–60º W, 52.5º–65º N).

## 3 Results

### 3.1 Sea surface biases

Temperature and salinity biases, through their impact on the zonal and vertical density gradients, are important for the realism of the ocean circulation and deep water formation in the North Atlantic. The mean SST biases of the individual LR-HIST and HR-HIST models are shown in Fig. 1, and their respective multi-model means in Fig. 2. In general, the HR-HIST ensemble mean displays warmer surface waters in the SPNA, compared to the LR-HIST one. The LR-HIST ensemble shows two main SST biases of opposite sign and similar magnitude. The first is a warm bias located along the North American coast, at NCH, with temperatures 2–5º C warmer than observations (Fig. 2). This bias has previously been associated with a misrepresentation of the position of the GS separation from the coast (Marzocchi et al., 2015). The other is a cold bias in the CNA (2–5º C), which earlier studies have linked to an unrealistic position of an overly weak NAC (Marzocchi et al., 2015) and an underestimation of the horizontal heat transport into the CNA domain (Lin et al., in review). The NCH bias has been shown to have an important impact on the global atmospheric circulation, through a Rossby wave response to local changes in vertical motion in the troposphere (Lee et al., 2018); the CNA bias has an effect on local precipitation (Moreno-Chamarro et al., 2022). In the HR-HIST mean, the NCH and CNA biases are reduced compared to LR-HIST (the significance of these reductions is analised in detail in Sect. 3.6). By contrast, the HR-HIST mean shows a positive bias of 1–3º C in the LS, which is weaker in the LR-HIST mean (Fig. 2; see also Sect. 3.6). We note, however, that the range of LS biases is larger in the LR-

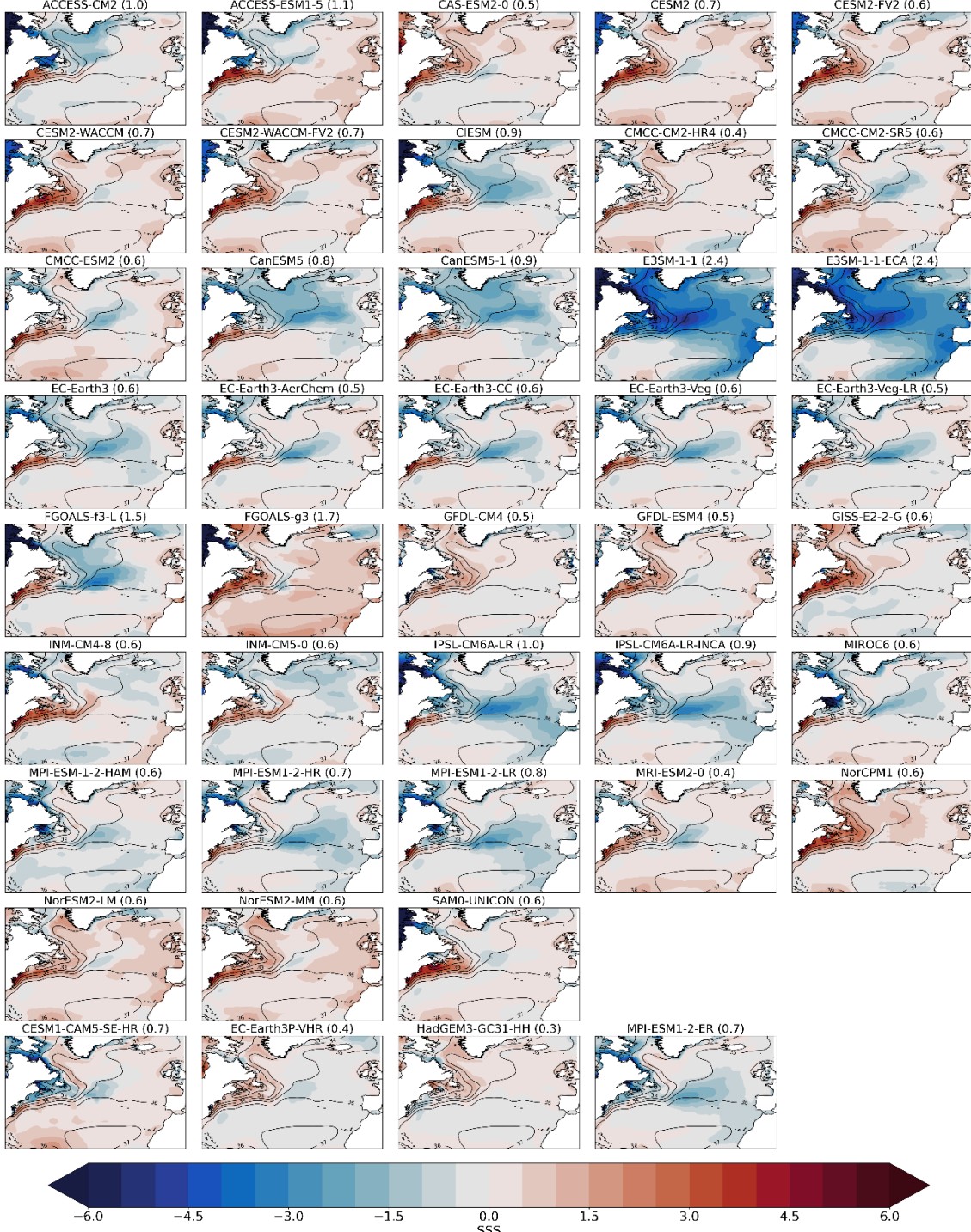

**Figure 3.** As in Fig. 1 but for the sea surface salinity (SSS) biases.

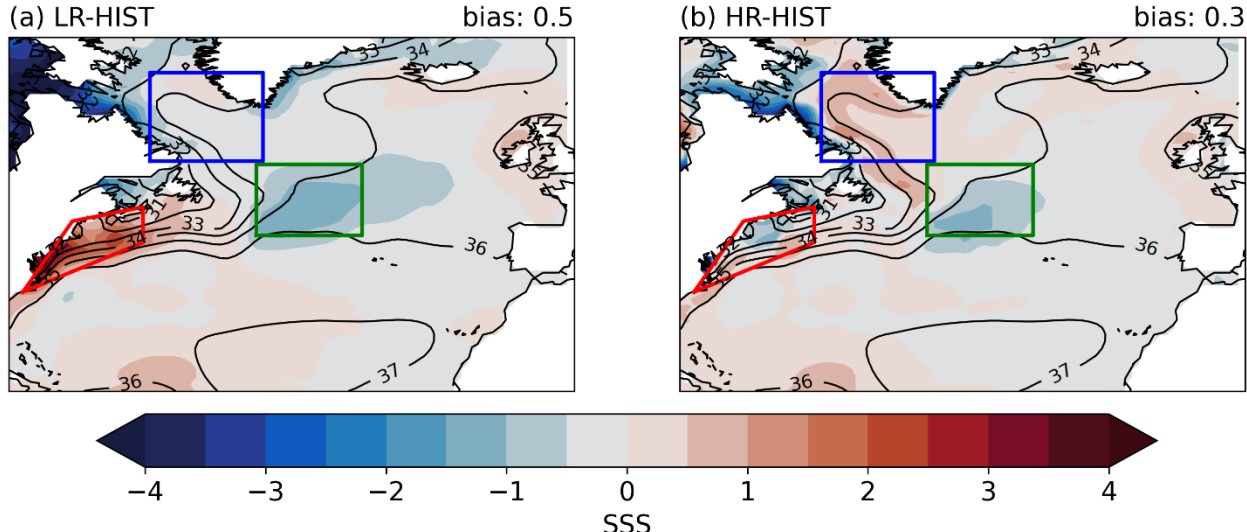

**Figure 4.** As in Fig. 2 but for the SSS biases.

HIST ensemble compared to the HR-HIST one, with some LR-HIST models showing larger positive biases than the HR-HIST ones (Figs. 1 and 16a; e.g. CAS-ESM2-0 and CESM2-WACCM-FV2), while other LR-HIST models present substantial biases of opposite sign (e.g. CanESM5-1 and E3SM-1-1).

Analogously to the SSTs, SSS biases from the individual models and the corresponding ensemble means are described in Figs. 3 and 4. The multi-model mean SSS biases show a similar pattern to the temperature ones (Fig. 4). LR-HIST presents a positive salinity bias of 1–3 at NCH, and a negative bias of 0.5–1.5 in the CNA. Note that salinities are presented on the practical salinity scale throughout the manuscript, with no associated units. In contrast to the SST biases, the SSS CNA negative bias is not a common feature in all LR-HIST models, although it is indeed dominant across them (Fig. 3). For HR-HIST, the NCH and CNA biases are reduced with respect to LR-HIST (see Sect. 3.6 for an analysis of significance), although a positive bias of 0.5–1 appears in the LS that is not present in the LR-HIST ensemble mean, probably because biases of different models compensate with each other. We note also that in the LS, models tend to show SST and SSS biases of the same sign, with CIESM, GISS-E2-2-G, INM-CM4-8, and INM-CM5-0 as exceptions. This might lead to a compensating contribution to the surface density biases. Despite the apparent LS degradation for HR-HIST, the spatially-averaged absolute SSS biases in the North Atlantic are substantially lower in HR-HIST simulations than in LR-HIST (i.e. 0.3 in HR-HIST vs 0.5 in LR-HIST), supporting the overall beneficial effect of the enhanced resolution.

## 3.2 Stratification in the Labrador/Irminger Sea

LS and Irminger Sea (IS) vertical water properties are important for deep water formation and connected to AMOC strength (Ortega et al., 2021). In Fig. 5, vertical profiles of temperature, salinity, and density for the Labrador/Irminger Sea (LIS) box shown in Figs. 7 and 8, are plotted and compared to EN4 observations, to characterize their related biases and assess the

differences across ensembles. The LIS domain contains the LS and the western part of the IS, and is defined to cover the area of weakest stratification in the North Atlantic (Ortega et al., 2021). Although the HR-HIST multi-model mean temperature profile (Fig. 5a, thick red curve) displays a larger surface bias in the LIS box compared to the LR-HIST one, when the whole vertical column is considered, HR-HIST temperature profiles are closer to EN4 than LR-HIST, as supported by the respective root mean square errors (RMSEs) in the vertical dimension against EN4 (Fig. 6a). The increase in vertical correlation against EN4 observed in the HR-HIST mean temperature profile compared to the LR-HIST mean profile (Fig. 6a) is not significant, which is related to a low correlation in the HadGEM3-GC31-HH model (see Sect. 3.6). The EC-Earth3P-VHR and MPI-ESM1-2-ER models exhibit the most realistic temperature profiles of all models, with CESM1-CAM5-SE-HR showing a very good vertical structure but a relatively high RMSE.

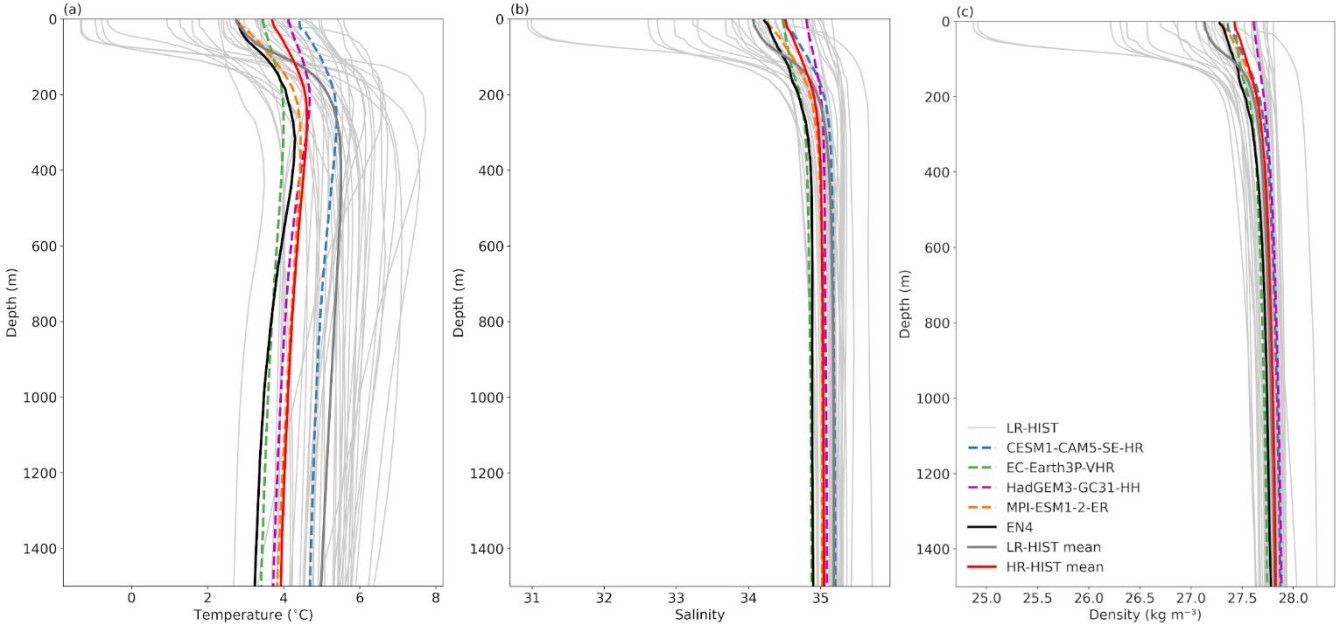

**Figure 5.** Vertical profiles of (a) temperature (in ºC), (b) salinity, and (c) density (in kg/m³), averaged over (35º–60º W, 50º–65º N), the Labrador/Irminger Sea (LIS) box shown in Figs. 7 and 8.

In terms of salinity, the HR-HIST ensemble displays a positive bias in the whole water column in the LIS box, while the mean for LR-HIST exhibits a negative bias in the upper 150 m, where it remains closer to the EN4 reference (Fig. 5b). However, below 150 m, the LR-HIST ensemble mean salinity bias turns positive and is stronger than for HR-HIST. Overall, the vertical salinity profile exhibits a more realistic shape in HR-HIST, a higher correlation coefficient, and a slightly smaller RMSE against EN4 (Figs. 5b, 6b; see Sect. 3.6 for an analysis of significance). The previous temperature and salinity profiles determine the realism of the climatological density profile in the LIS box in both ensembles. Figure 5c shows multi-model mean density biases of opposite sign and similar magnitude in the top ~100 m for HR-HIST and LR-HIST, and positive biases of comparable magnitude below. More interestingly, HR-HIST exhibits a more realistic density stratification, which is largely overestimated in LR-HIST. Indeed, the density profile for HR-HIST is closer in shape to the EN4-derived one, as supported

by Pearson correlation coefficients in Fig. 6c, which are very close to one in all HR-HIST models, and by the relatively small RMSEs. Although some individual LR-HIST models present similar (or improved) density profiles compared to HR-HIST, the LR-HIST ensemble shows significant spread, with many models being far from observations both in terms of correlation and RMSE (Fig. 6c). We note that the use of vertical correlation coefficients to assess resemblance between two vertical profiles should come in conjunction with other metrics, such as RMSEs, or direct visual inspection of profiles, as a high correlation coefficient alone does not ensure a small distance between curves.

The comparatively lower density stratification in HR-HIST is explained by a relative reduction in salinity stratification (which is partly counterbalanced by the relative increase in temperature stratification; Fig. 5). The improved shape of the HR-HIST mean density profile is expected to impact on the vertical mixing, which is addressed in the next section.

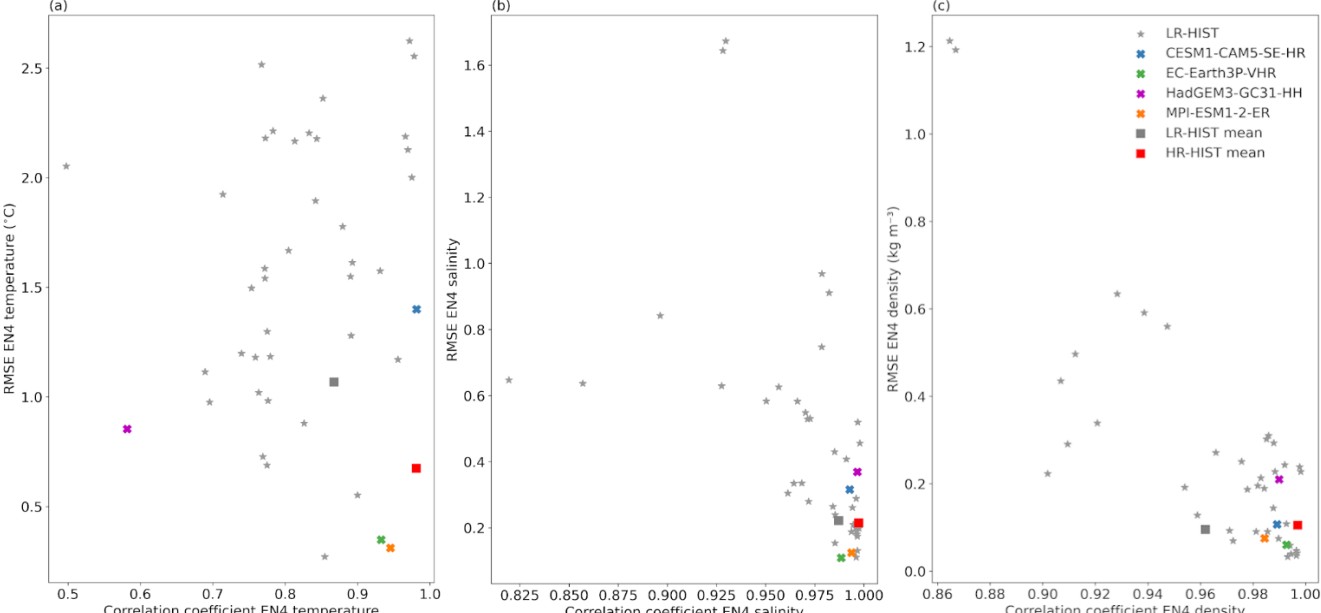

**Figure 6.** Pearson correlation coefficient (horizontal axis) and Root Mean Square Error (RMSE) (vertical axis; units as in Fig. 5) for the vertical (a) temperature, (b) salinity and (c) density profiles in Fig. 5 against EN4. Profiles are averaged over the region (35º–60º W, 50º–65º N), the LIS box shown in Figs. 7 and 8.

## 3.3 Mixed layer depth

MLD in the North Atlantic is generally used as a proxy for deep water convection, it has been shown to correlate with the AMOC strength (Li et al., 2019, Martin-Martinez et al., 2025) and it achieves its maximum in March. We note that results from a recent observational study suggest that AMOC variability depends on combined density anomalies from different areas of the SPNA rather than from one location alone (Li et al., 2021), suggesting that MLD analyses in the SPNA should not be limited to one single region (e.g. the central LS). The March climatology for the two multi-model ensembles, and EN4- and ARMOR3D-derived values, is shown in Figs. 7 and 8. In the LS, the multi-model mean of HR-HIST shows a deeper (although

not significantly deeper; see Sect. 3.6) mixed layer than the LR-HIST mean (Fig. 8), consistent with a relatively weaker density stratification (Sect. 3.2). If we check the individual models (Fig. 7) we note that all the HR-HIST models show deep mixed layers in the LS, while ~25 % of the LR-HIST models show little or no convection. LS mixed layers in the HR-HIST mean (1800–2000 m) are closer to EN4 estimates (2000–2200 m), whereas in ARMOR3D (1000–1200 m) they are in the same range as in the LR-HIST mean (1000–1200 m). For the overlapping time interval of ARMOR3D and EN4 (i.e., for 1993–2014), EN4 values for the LS are still larger (1800–2000 m; not shown) compared to ARMOR3D. The wide range in the observation-derived estimates for the LS in our analyses leads us to review the literature for observational studies. Work by Holte et al. (2017) based on individual Argo density profiles shows mixed layers down to 1400–1800 m in the LS for the 2000–2016 period. Time-varying estimates of winter maximum MLDs in the LS obtained from Argo floats, the AR7W line, and moored measurements, suggest values mostly around 1100–1500 m in the 2002–2015 interval (Yashayaev and Loder, 2016), showing an intensification in recent years, with a record value of 2100 m in 2016 (Yashayaev and Loder, 2017). Therefore, the MLD values in our HR-HIST (1800–2000 m) ensemble mean are slightly too large compared to observational studies, and the LR-HIST (1000–1200 m) ensemble mean values are slightly too shallow. The differences in the MLD estimates obtained across the different studies might arise from the different temporal and spatial characteristics of the profile data, differences in the time intervals analysed, and from the different methodological approaches employed. The yearly estimates by Yashayaev and Loder (2016; 2017) are winter maximum values of "aggregate" maximum convection depths, defined as the 75[th] percentile of the depth of the base of the pycnostad in the set of available individual LS profiles at each time. Holte et al. (2017)'s MLD estimates (shown in their Fig. 3a) correspond to individual Argo profiles and are obtained with a density algorithm (Holte and Talley, 2009) that uses a combination of methods and elements (including temperature and density threshold methods, gradient methods, estimates of thermocline linear fits, etc). In our study, instead, MLD values are a climatology of March MLD monthly means obtained from gridded temperature and salinity data through a density threshold method.

Notice that the convection area along the East Greenland Current, in the western IS, is also deeper in the HR-HIST ensemble with respect to LR-HIST, better resembling the EN4 and ARMOR3D patterns. A remarkable feature in the ARMOR3D dataset is a distinct stripe of deep mixing (1200–1400 m deep) attached to the shelf along the East Greenland Current, which is also slightly visible in some of the individual HR-HIST models (e.g. HadGEM3-GC31-HH; Fig. 7) and might be absent in EN4 due to its coarser resolution.

Additional deepening of the mixed layer at higher resolution is found in the Nordic Seas, where the multi-model means reach depths of 1400–1600 m for HR-HIST versus only 1000–1200 m for LR-HIST. ARMOR3D and EN4-derived values are the largest, reaching down to 3000–3200 m and 2200–2400 m, respectively. We note that the core of the deep mixing area in the Nordic Seas in HR-HIST is displaced westward compared with EN4, ARMOR3D and LR-HIST.

In the eastern IS and in the Iceland basin, the MLD shows values down to 600–800 m for the LR-HIST mean, 400–600 m for the HR-HIST mean, and 600–800 m for ARMOR3D. EN4 values are the largest, reaching down to 800–1000 m, due to stronger mixing south-west of the Denmark Strait in the pre-Argo EN4 data (Fig. A4). The latter could be related to natural variability

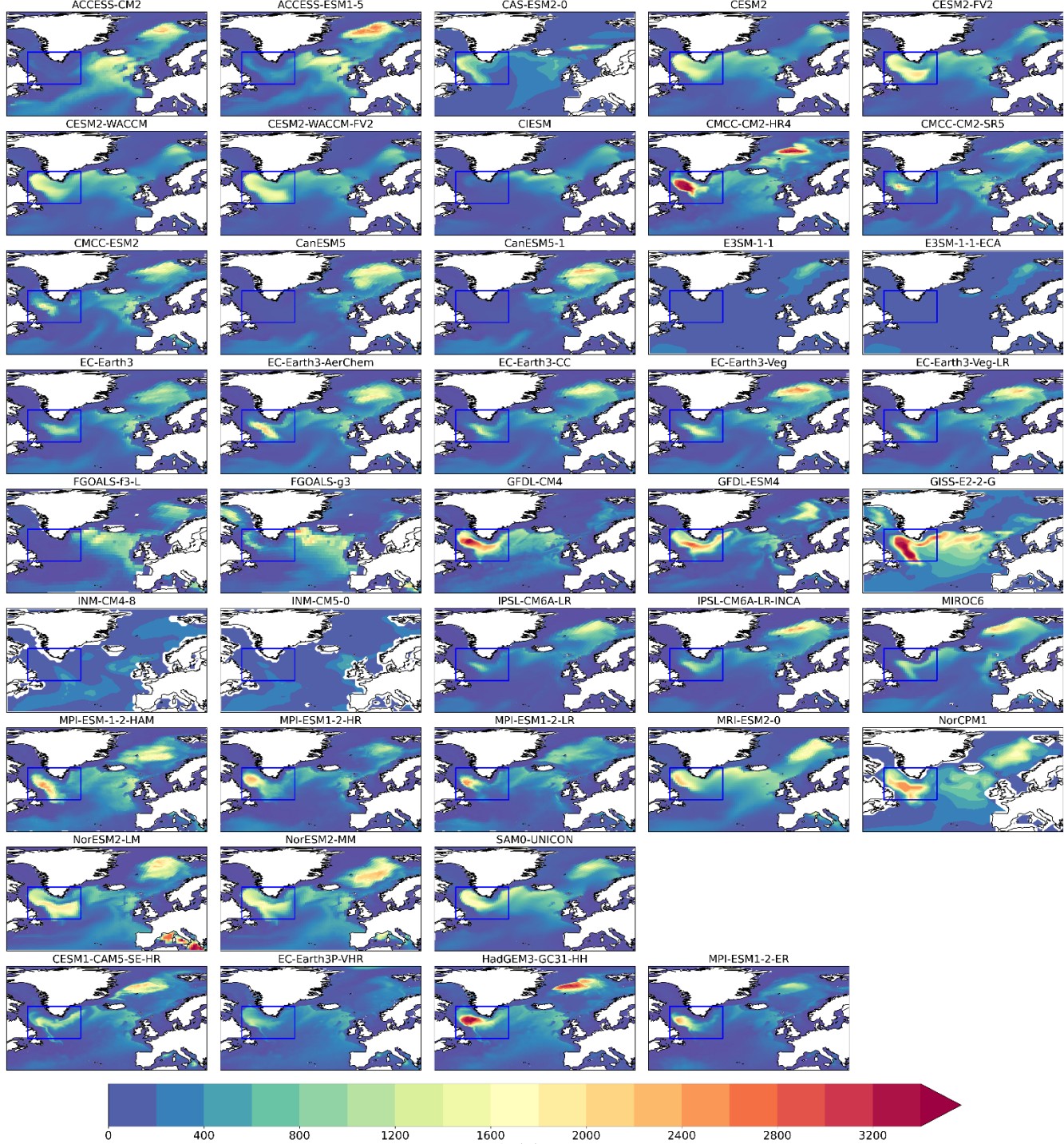

**Figure 7.** March mixed layer depth (MLD; in m) for LR-HIST (rows 1–8) and HR-HIST models (row 9). MLD is calculated using the potential density threshold method described in de Boyer Montégut et al. (2004), with a threshold value of 0.03 kg m$^{-3}$ and a reference depth of 10 m. The blue box (35º–60º W, 50º–65º N) indicates the region used in the vertical profiles calculations in Sect. 3.2.

but also to the more restricted availability of EN4 profiles before 1999. Next section will address if these overall improvements

in deep mixing for HR-HIST are accompanied by an enhanced representation of the AMOC streamfunction.

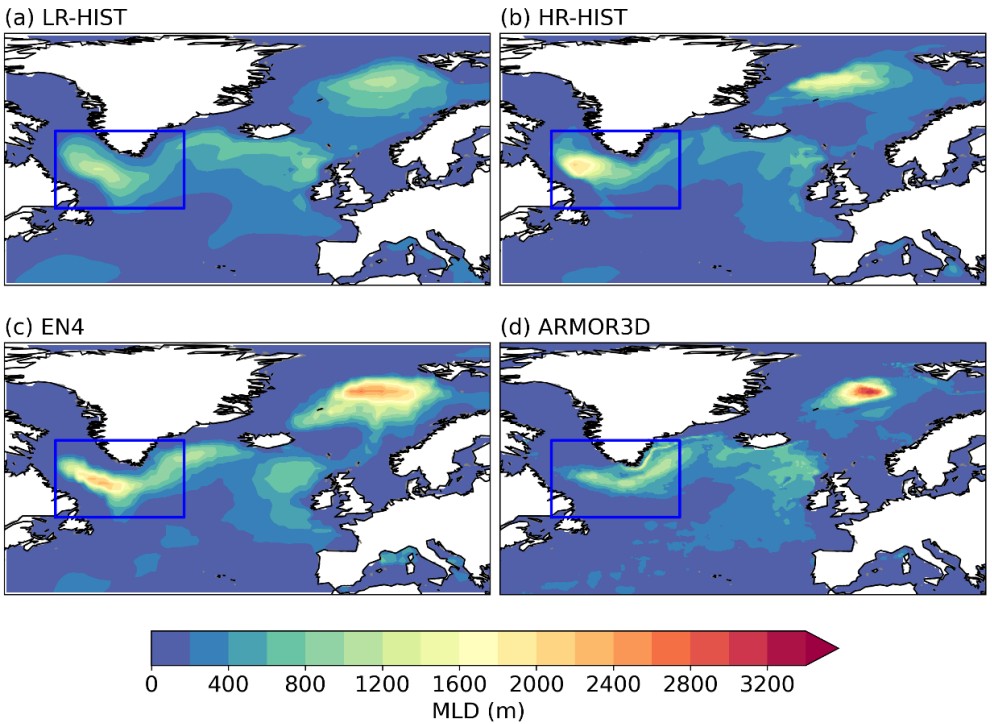

**Figure 8.** March MLD (in m) by groups, for (a) LR-HIST, (b) HR-HIST, (c) EN4, and (d) ARMOR3D. In all cases, MLD has been calculated
from temperature and salinity fields using the density threshold method of 0.03 kg m$^{-3}$ described in the manuscript. The time interval covered
is 1980–2014 in (a), (b), (c), and 1993–2014 in (d). Plotting details as in Fig. 7.

## 3.4 Atlantic overturning in depth-space

The AMOC streamfunction is a measure of the northward ocean volume transport, integrated zonally over the Atlantic

basin and cumulatively from the top of the ocean as a function of depth. The Atlantic overturning streamfunction in the depth-

space for the individual models and reanalysis is shown in Fig. 9, and their LR-HIST and HR-HIST ensemble means are

compared with ORAS5m and GLORYS12 reanalyses in Fig. 10. The Atlantic overturning in depth-space is weaker, although

not significantly weaker (see Sect 3.6), in the multi-model HR-HIST mean compared to the LR-HIST one. In ORAS5m, it is

weaker than in HR-HIST and LR-HIST, and the opposite is true for GLORYS12 (Fig. 10). In both LR-HIST and HR-HIST,

the upper cell of the AMOC streamfunction is shallower compared to ORAS5m and GLORYS12, with the return branch

reaching depths of 3000 m in models vs. depths below 3500 m in reanalysis data. In the individual-model plot (Fig. 9), we

observe that members are much more homogeneous in terms of overturning structure and intensity within HR-HIST than

within LR-HIST, with maximum climatological values close to 35º N in all four HR-HIST models. The AMOC streamfunction

in the HR-HIST models, ORAS5m and GLORYS12 present a sharp feature at the latitude of the maximum (~ 35º N), as

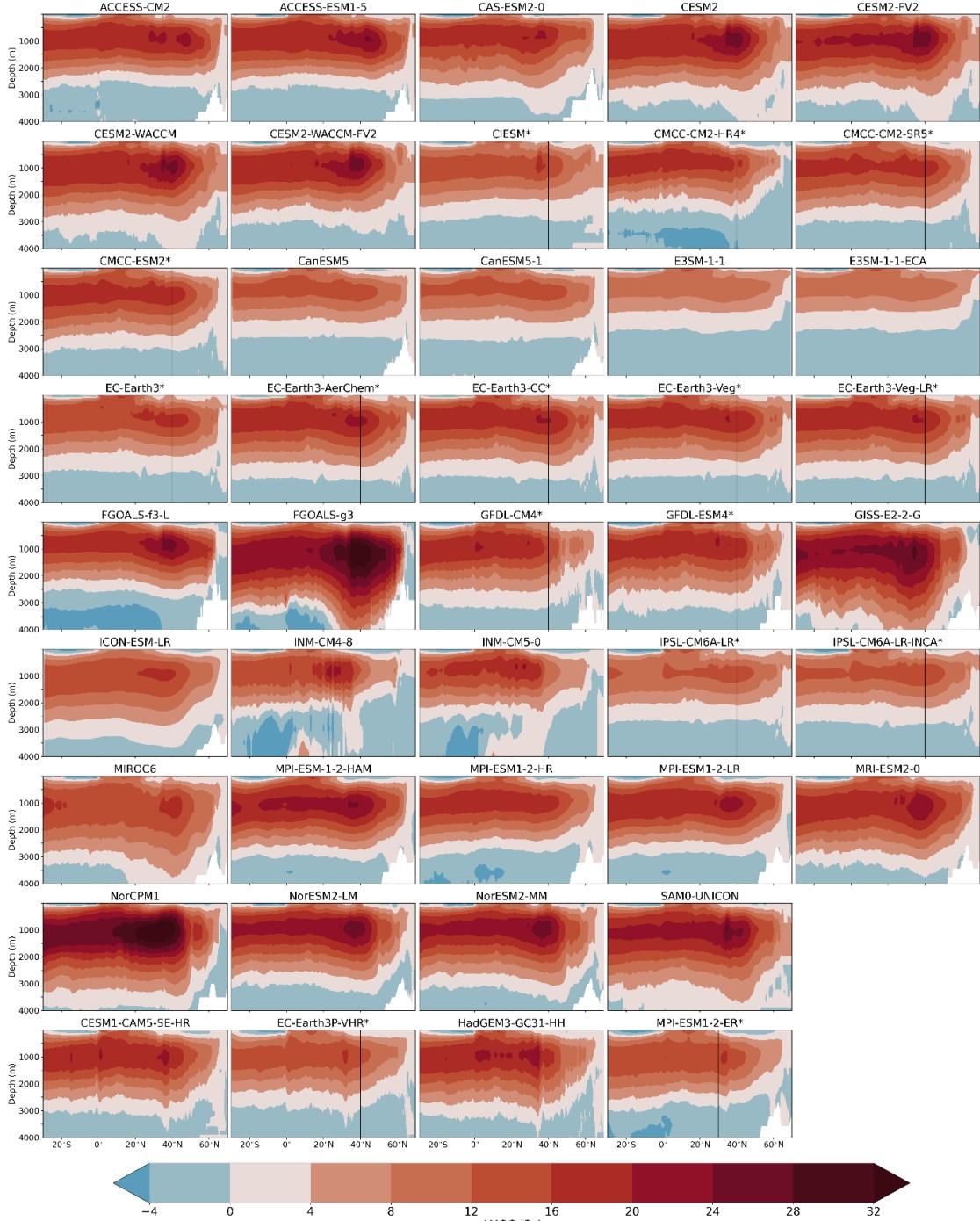

**Figure 9.** Atlantic overturning streamfunction in depth-space (in Sv) for LR-HIST (rows 1–8), and HR-HIST models (last row). The asterisks indicate when msftyz is displayed. Values at latitudes north of the black vertical lines might be affected by grid rotation (Sect. 2.1).

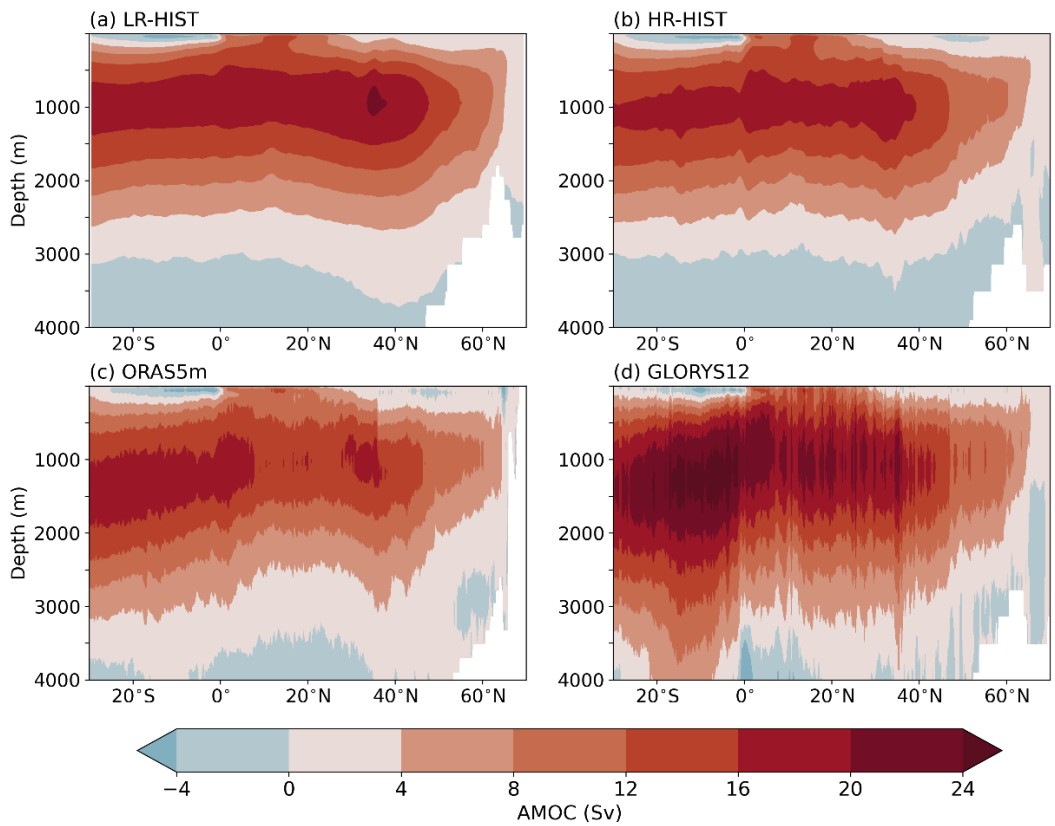

**Figure 10.** Atlantic overturning streamfunction in depth-space (in Sv) by groups, for (a) LR-HIST, (b) HR-HIST, and (c) ORAS5m and (d) GLORYS12 reanalyses. The time interval covered is 1980–2014 in (a), (b), (c), and 1993–2014 in (d).

opposed to a more horizontally uniform flow in the LR-HIST models, which might be related to the higher resolution of the HR-HIST and reanalysis models compared to the LR-HIST models (Figs. 9, 10; Sect. 4).

In order to compare model results with direct observational evidence, we examine the Atlantic overturning streamfunction at 26.5º N, where RAPID volume transports are available for the 2004–2014 period (Fig. 11a). The vertical profile at 26.5º N indicates a weaker, although not significantly weaker (see Sect. 3.6) overturning for the multi-model mean of HR-HIST (max. 17 Sv) compared to LR-HIST (max. 18.8 Sv) which is also closer to RAPID observations (max. 16.8 Sv) and ORAS5m reanalysis (max. 15.3 Sv). GLORYS12 (max. 20.3 Sv) seems to overestimate AMOC strength in relation to RAPID. The HR-HIST mean profile shows a particularly good fit with the RAPID array above ~1000 m, although, in general, the overturning profile is too shallow both for LR-HIST and HR-HIST compared to RAPID. HR-HIST models remain relatively close to the RAPID data (see Sect 3.6), and the main outliers both in terms of under- and overestimation of Atlantic vertical overturning are LR-HIST models (Fig. 11a). Some differences between models and observations might stem from the methodologies used to derive the AMOC profiles. While in models the AMOC streamfunction is obtained by integrating model velocities, which are simulated at every grid point, this approach is not possible with observations, since direct velocity measurements are scarce.

The calculation of the upper mid-ocean return transport in RAPID is based on the zonal gradient of dynamic heights from
density profiles, which makes use of a reference depth (4820 m), representing a level-of-no-motion (Roberts et al., 2013;
McCarthy et al., 2015; Danabasoglu et al., 2021). Some studies report sensitivity of the estimated RAPID profile, particularly
in the deep ocean, to the choice of this reference depth (Fig 3.2 in McCarthy et al., 2015; Fig. S3 in Roberts et al, 2013), which
might explain some of the differences between the RAPID and model profiles in the deep ocean (Fig. 11). However,
uncertainties related to the choice of a reference depth are within the range of the accuracy of the RAPID method, and
uncertainties in deep transport are a current topic in the literature (McCarthy et al., 2015). A model-based study also suggests
that estimating the AMOC via RAPID's physical assumptions could lead to an underestimation of up to 1.5 Sv in its mean
value at ~900 m depth (Sinha et al., 2018) compared to its real strength, a result that is, however, not supported in a more
recent study based on a different ocean model (Danabasoglu et al., 2021). Model and reanalysis overturning profiles at 26.5º
N in our study are computed from full velocities and do not include the application of a uniform volume transport compensation
term. This might be relevant for the GLORYS12 overturning profile, which is based on regridded velocities (see Sect. 2.3).
We would like to note the existence of the software package Meridional ovErTurning ciRculation diagnostIC (METRIC), for
calculating RAPID observations-equivalent AMOC diagnostics in models using different model output variables (Castruccio,
2021; Danabasoglu et al., 2021).

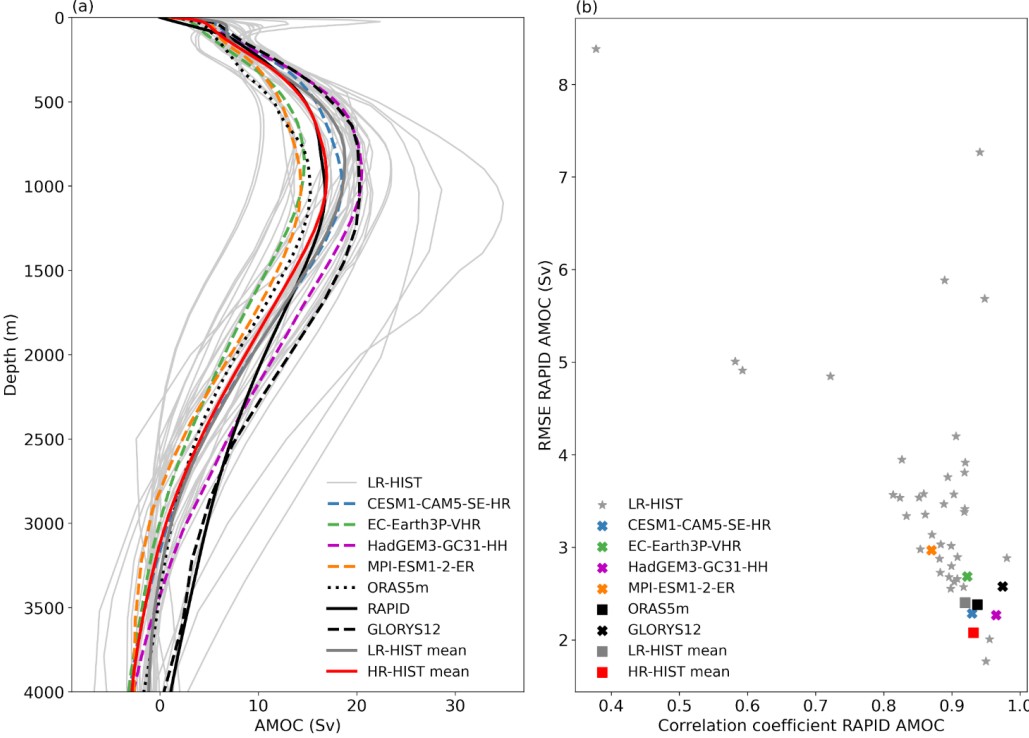

**Figure 11.** (a) Climatological Atlantic overturning profile in depth-space at 26.5º N (in Sv). (b) Pearson correlation coefficient (horizontal
axis) and Root Mean Square Error (RMSE; in Sv) (vertical axis) of vertical profiles in (a) against RAPID. In both subplots, models and
ORAS5m reanalysis data correspond to the interval 1980–2014, RAPID observations are averaged over the period April 2004–December
2014, and GLORYS12 reanalysis over the interval 1993–2014.

To perform a quantitative comparison between the two ensembles, we extend the analysis of the Atlantic overturning profiles by computing two metrics that measure the degree of agreement of the different models with RAPID observations, as diagnosed by the Pearson correlation (x-axis in Fig. 11b) and the RMSE (y-axis in Fig. 11b) across the vertical dimension. Figure 11b confirms that, although none of the HR-HIST models is systematically better than all the LR-HIST ones, HR-HIST models lie within the range of best performing models, both in terms of vertical correlation and RMSE against RAPID, with HR-HIST models concentrated close to the bottom-right corner of the figure (see Sect. 3.6). To complement this analysis of the impact of resolution on the overturning circulation, next section looks at the impact on the gyre circulations in the North Atlantic.

**3.5 Gyre circulations**

In this section, the main gyre circulations of the North Atlantic are examined, as described by the BSF, a measure of the vertically integrated volume transport. The gyre circulations play a key role in climate in terms of northward ocean heat and freshwater transport and deep water formation. In order to validate the position of the GS and NAC in models, we also plot the zero contour line of absolute dynamic topography from AVISO observations, which delimits the intergyre boundary (dashed lines in Figs. 12 and 13). Ideally, this zero line in observations would overlap the zero line of the model BSF, as for ORAS5m in Fig. 13c.

In the multi-model mean of LR-HIST, the GS separates too far north from the American coast compared to AVISO, which implies that its NCH bias region (Fig. 13a, red polygon) is only influenced by warmer, more saline waters of southern origin, in contrast with ORAS5m and GLORYS12, where entrainment of colder, fresher waters from the north occurs. This can therefore explain the positive temperature and salinity biases in NCH described in Sect. 3.1.

Instead, in the multi-model mean of HR-HIST, the GS separates from the coast further south compared to LR-HIST, and the NCH region is partially influenced by waters of northern origin, as in the reanalyses (Fig. 13b, c, d). These results can explain why the HR-HIST models show comparatively reduced SST and SSS biases with respect to LR-HIST in that area (Figs. 2 and 4). Furthermore, HR-HIST displays an improved GS structure (e.g. in the Florida Current) with a narrower and locally stronger current than LR-HIST (Fig. 13b), in closer agreement with ORAS5m and GLORYS12, although this could be partly explained by the fact that these reanalyses were produced with a mesoscale eddy-permitting/resolving ocean. We note that the entrainment of waters of northern origin in the GS region reaches slightly too far south compared to AVISO in the HR-HIST ensemble mean, and also in GLORYS12. In the case of HR-HIST, this is mainly seen in HadGEM3-GC31-HH, and to a lesser extent in EC-Earth3P-VHR (CESM1-CAM5-SE-HR BSF data was not available and could not be included in Figs. 12 and 13).

The NAC is too zonal in most LR-HIST models (Fig. 12). In the LR-HIST ensemble mean, the CNA region is only touched at its southern edge by the warmer/saltier NAC waters, remaining predominantly exposed to the influence of the SPG (Fig.

13a). By contrast, for HR-HIST, ORAS5m, and GLORYS12, the SPG has a more restrained influence on that region in favor of a less zonal NAC (Fig. 13b, c, d), which could explain the reduced cold and fresh biases at high resolution in the CNA for HR-HIST (Sect. 3.1). Notice though that the NAC is still slightly too zonal in the MPI-ESM1-2-ER model in the CNA, and slightly too meridional in GLORYS12 and HadGEM3-GC31-HH.

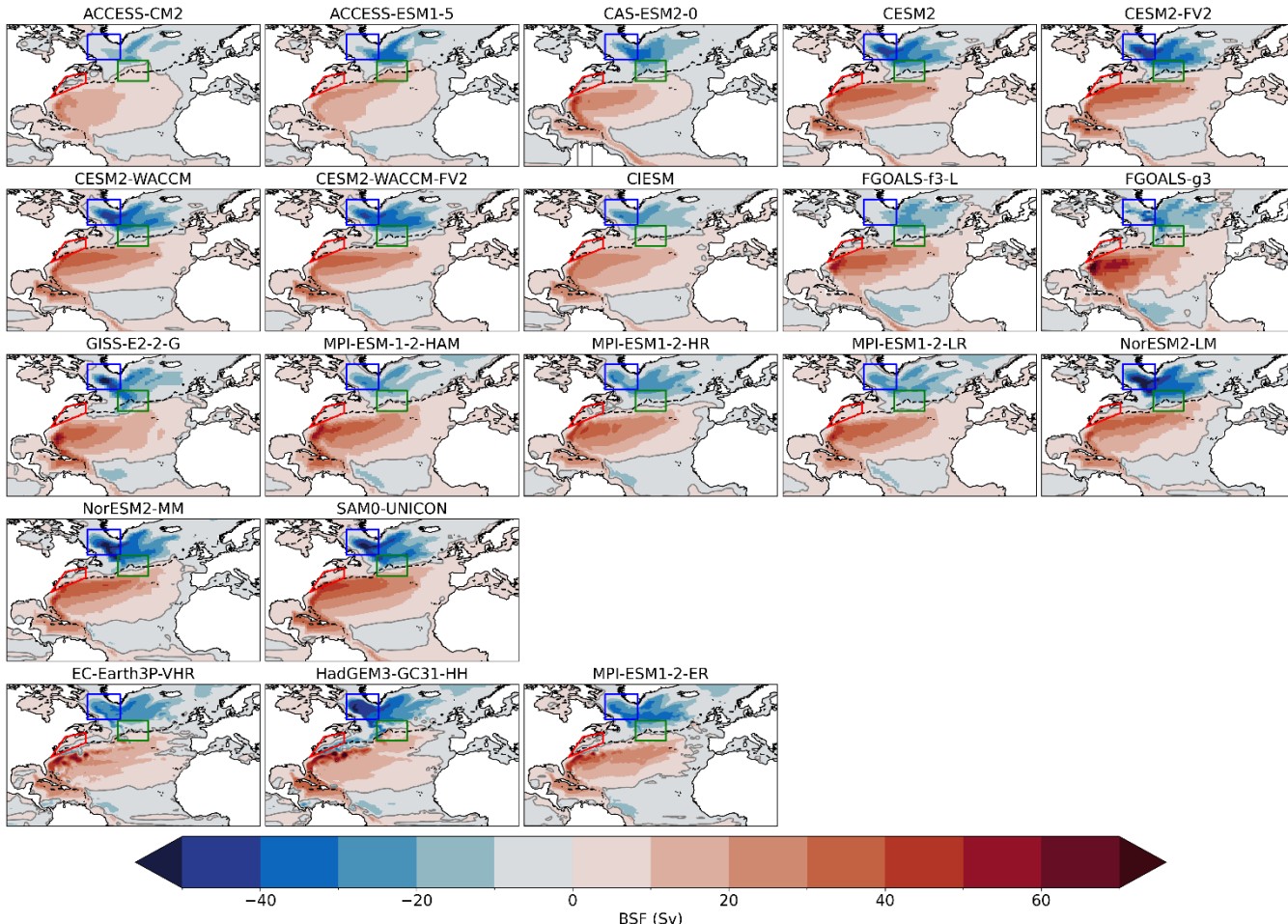

**Figure 12.** Barotropic streamfunction (BSF; in Sv) for LR-HIST (rows 1–4) and HR-HIST models (last row). Zero contour of absolute dynamic topography from AVISO observations (dashed black line) corresponding to the period 1993–2014 is also shown. Boxes as in Figs. 2 and 4: NCH in red, CNA in green, LS in blue. Note: some models did not present a BSF output (e.g. CESM1-CAM5-SE-HR) or this presented unrealistic values, hence the reason fewer models are shown in this figure.

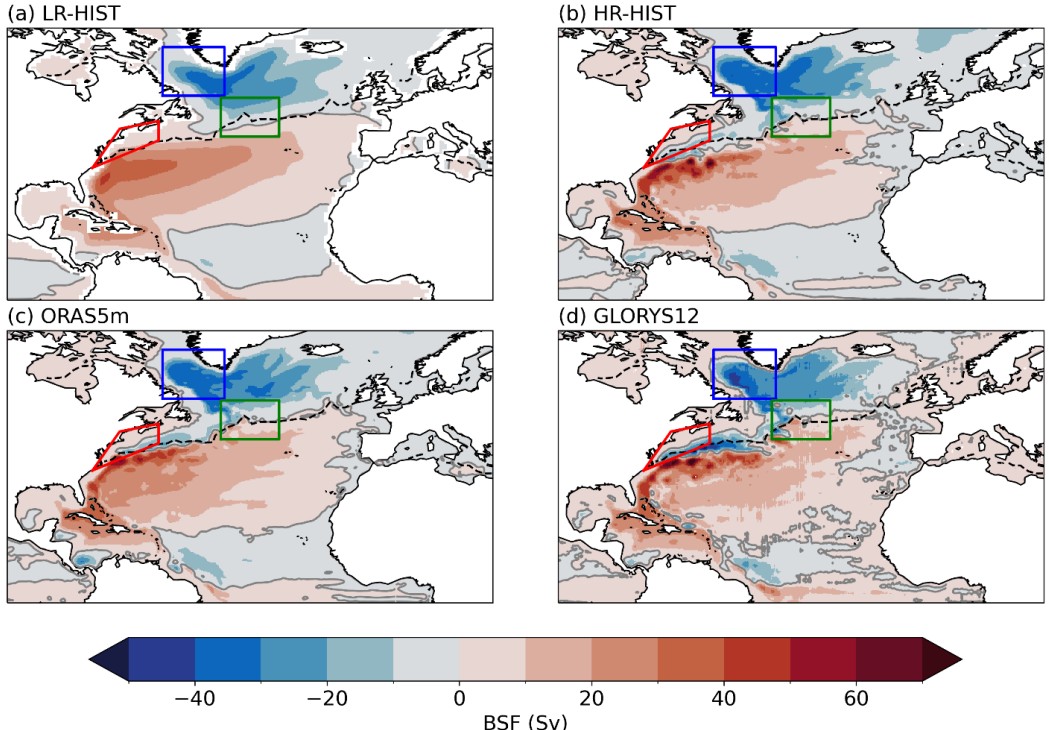

**Figure 13.** BSF (in Sv) by groups for (a) LR-HIST, (b) HR-HIST, (c) ORAS5m and (d) GLORYS12. The time interval covered is 1980–2014 in (a), (b), (c), and 1993–2014 in (d). Plotting details as in Fig. 12.

Although the SPG is stronger in the HR-HIST mean compared to the LR-HIST mean (Fig. 13), the difference in means in SPG strength (calculated as the absolute value of the minimum of the BSF in the LIS box) is not significant in our ensembles (see Sect. 3.6). We note that some LR-HIST models like NorESM2-LM and SAM0-UNICON have gyres of comparable intensity to HadGEM3-GC31-HH, which has the strongest SPG across the available HR-HIST models (Fig. 12). On the other hand, the HR-HIST SPG presents a narrower and locally stronger boundary current, as well as an improved structure in the west (in the LS), closer to that in the ORAS5m and GLORYS12 reanalyses. Although the above-mentioned gyre strengthening is not significant, a stronger SPG in HR-HIST is consistent with the stronger convection identified in Sect. 3.3, as this latter enhances the presence of dense waters in the deep ocean, subsequently strengthening the baroclinic pressure gradient that drives the gyre (Yashayaev and Loder, 2009). A larger sample of HR-HIST models is required to further investigate the relationship between model resolution and SPG strength and structure.

### 3.6 Testing the significance of differences between ensembles

In order to test the significance of the results described in the previous sections (Sects. 3.1–3.5), bootstrapping (see Sect. 2.4 for a detailed description of the method) is applied to the following single number metrics (first column in Table A12): SST and SSS biases in the LS, CNA, and NCH regions (Sect. 3.1); max. Atlantic overturning streamfunction at 26.5º N, as

well as RMSE and Pearson correlation of Atlantic overturning profiles at 26.5º N with respect to RAPID (Sect. 3.4); RMSE and Pearson correlation of temperature, salinity and density profiles in the LIS box with respect to EN4 (Sect. 3.2); max. MLD in the LIS box (Sect. 3.3); and max. SPG strength, calculated as the absolute value of the minimum of the BSF in the LIS box (Sect. 3.5). Our goal is to investigate whether the differences in means between the HR-HIST and LR-HIST ensembles associated with these metrics are significant.

When bootstrapping is applied to LS SST and SSS biases employing LR-HIST samples of the size of the total LR-HIST ensemble (see Sect. 2.4), the CI obtained does not include the value zero, which means that the difference in means between the two ensembles is significant for those metrics (second column in Table A1). By contrast, when the size of the LR-HIST ensemble samples is reduced to the size of the HR-HIST ensemble (i.e., to four), the CI does include zero (third column in Table A1). Sizes of 19 and 25 for the LR-HIST ensemble samples are required for SST and SSS, respectively, for the difference in means to become significant (last column in Table A1). This is due to the fact that several models within the LR-HIST ensemble present LS SST and SSS biases of comparable magnitude to those of the HR-HIST ensemble (Fig. A3). We would like to note, though, that results are significant when the whole LR-HIST ensemble size is considered, and that even in the case of a reduced LR-HIST sample size, the corresponding CIs are clearly centered to the right of zero (Table A1). This implies that the LR-HIST ensemble has a better performance than the HR-HIST ensemble in regard to LS SSTs and SSSs.

When the targeted metrics are the CNA SST and SSS biases, the reduction observed in the respective HR-HIST ensemble means is not significant (for all LR-HIST ensemble subsample sizes; Table A1). We note though that CIs are rather centered to the right of zero. In the case of SSTs, the lack of significance is associated with the cold biases in MPI-ESM1-2-ER and EC-Eart3P-VHR still present in that area (Fig. A3; Sect. 3.7). For SSSs, the lack of significance is related to the fact that several LR-HIST models have a similar performance to the HR-HIST models in that region. Also, we note that the MPI-ESM1-2-ER model still presents a significant SSS bias in the CNA region (Fig. A3; Sect 3.7).

As for the NCH region, the bootstrapping analysis shows that both SST and SSS biases are significantly reduced in the HR-HIST ensemble, compared to the LR-HIST ensemble (Table A1). However, in the case of SSTs, samples of at least size 8 are required from the LR-HIST ensemble for significance to be achieved, which is related to the warm bias present in the CESM1-CAM5-SE-HR model (Fig. A2; Sect. 3.7).

In the case of max. Atlantic overturning at 26.5º N, the reduction in strength in the HR-HIST ensemble is not significant (Table A1), since several LR-HIST models show values within the range of the HR-HIST ensemble or even lower (Fig. 11a). We note, though, that the CIs of the difference in means are rather centered to the left of zero. Interestingly, the Atlantic overturning profile distance to RAPID, as measured by the RMSE, is significantly reduced in the HR-HIST ensemble (for all LR-HIST ensemble subsample sizes; Table A1). The increase in correlation to the RAPID curve in the HR-HIST ensemble becomes significant when the LR-HIST samples considered have a minimum size of 14, due to several LR-HIST models presenting correlation values within the same range of the HR-HIST ensemble (Fig. 11b).

In terms of temperature profiles in the LIS box, RMSEs relative to EN4 are significantly reduced in the HR-HIST ensemble compared to LR-HIST, even when considering small subsamples of size four in the LR-HIST ensemble in the bootstrapping

analysis (Table A1). The reduction in RMSEs is particularly pronounced in the EC-Earth3P-VHR and MPI-ESM1-2-ER models (Fig. 6a; Sect. 3.7). The increase in correlation with respect to the EN4 temperature profile in the HR-HIST ensemble is not significant, which is due to the low correlation exhibited by HadGEM3-GC31-HH (Fig. 6a; Sect. 3.7). By removing this model from the bootstrapping calculations, the correlation becomes significant even with a reduced LR-HIST subsample size (not shown). Regarding the salinity and density profiles, improvements in the HR-HIST ensemble related to both RMSE and correlation to EN4 become significant already with relatively small LR-HIST sample sizes (Table A1).

The increase in max. MLD and in SPG strength observed in the HR-HIST ensemble compared to LR-HIST are not significant (Table A1), since several LR-HIST models present values within the same range displayed in the HR-HIST ensemble (Figs. 7, 12 and 14b). We note though that in the case of max. MLD, CIs are centered to the right of zero.

## 3.7 Characteristic features in the HR-HIST models

After analysing the improvements in the representation of the North Atlantic mean state associated with the use of mesoscale-resolving models, in this section we aim at describing inter-model differences within the HR-HIST ensemble.

CESM1-CAM5-SE-HR stands out among the HR-HIST models for having the largest (warm) SST biases in the LS (2.52 ºC; Fig. A3) and NCH (3.79 ºC; Fig A2) regions, and the smallest (cold) SST bias in the CNA region (-0.02 ºC; Fig. A3). Figure 1 shows that this model has a general warm bias over the North Atlantic, which is also present at the subsurface, as shown in the temperature vertical profiles of the LIS region (Fig. 5). Despite this warm bias, CESM1-CAM5-SE-HR presents one of the best fits to observations (together with HadGEM3-GC31-HH) in terms of Atlantic overturning profiles at 26.5º N (Fig. 11b).

As for MPI-ESM1-2-ER, this model is characterized by large negative SST (-4.45 ºC) and SSS (-1.25) biases in the CNA (Fig. A2), which might be linked to a weak and still too zonal NAC (Fig. 12), and are consistent with a weak vertical overturning (Figs. 11a, 14a). However, this model lies within the best performing mesoscale eddy-resolving models in terms of vertical profiles in the LIS region (Fig. 6). Similarly, EC-Eart3P-VHR presents the second largest cold bias in the CNA among HR-HIST models (-2.94 ºC) (Fig. A2), after that of MPI-ESM1-2-ER, which might be again related to a weak AMOC (Fig. 11a). We note though that this model exhibits the most realistic density profiles in the LIS region within the HR-HIST ensemble (Fig. 6c).

Regarding HadGEM3-GC31-HH, this model shows the largest LS surface salinity bias (0.66) and the second largest LS surface temperature bias (1.62 ºC) within the HR-HIST ensemble (Fig. A3), which is consistent with a weak stratification (Fig. 5c) and overly strong convection (Fig. 7) in the LIS region. This could be related to the fact that the NAC is slightly too meridional in the east compared to AVISO (see Fig. 12). Nevertheless, HadGEM3-GC31-HH presents the weakest SSS bias (0.17) in the CNA, the second weakest SST bias (0.39 ºC) in the CNA (Fig. A3), and the second best fit to RAPID AMOC at 26.5º N (Fig. 11b) among HR-HIST models.

Interestingly, the HR-HIST models with the largest LS SST biases (Fig. 1) are those with a stronger than observed vertical overturning (CESM1-CAM5-SE-HR and HadGEM3-GC31-HH; Fig. 11a), which suggests that the LS SST bias might be linked to northward heat transport through the NAC. Nevertheless, the fact that models with a large negative SST bias in the CNA region (Fig. 1) and a weaker than observed overturning (Fig. 11a; MPI-ESM1-2-ER and EC-Eart3P-VHR) still present a positive SST bias in the LS (Fig. 1) suggests that additional mechanisms, apart from heat transport through the NAC, might contribute to the LS SST bias.

### 3.8 Relations between dynamical and physical properties

In order to explore the relationships between the different ocean dynamical structures in the North Atlantic, correlations between AMOC streamfunction strength, SPG strength, and March MLD are analysed in Fig. 14. Our results show a correlation ($r=0.47$; $p<0.01$) between max. AMOC streamfunction at 26.5º N and max. MLD in the LIS region (Fig. 14a), with stronger overturning values associated with deeper mixed layers, in agreement with Li et al. (2019). We note though that in their case, the obtained correlation value is larger ($r=0.83$; their Fig. 11), which might be related to a smaller (more homogeneous) model ensemble and perhaps the use of mean MLD values instead of max. values in that study. Also max. MLD and SPG strength are significantly correlated in our analysis ($r=0.47$; $p=0.04$; Fig. 14b), consistent with results by Koenigk et al. (2021) and with the paradigm described by Straneo (2006), which shows that the SPG is partially driven by the difference in density between the LS interior basin and the LS boundary current, as well as by surface winds. A positive correlation between SPG strength and max. overturning at 26.5º N exists ($r=0.57$; $p<0.01$; Fig. 14c), which supports the relationship between deep water sinking and velocities at the SPG boundary current described again by Straneo (2006; equation (17) therein). Note that although the correlations described here are consistent, this consistency does not always translate into full observational fidelity, meaning that some models may agree with observational constraints for one metric but not for the other. For example, whereas several models with max. AMOC values within the range of observational estimates display max. MLD values which are also close to observations, some others exhibit MLD values that are too large compared to observations (Fig. 14a).

Surface water properties in the North Atlantic mid- and high-latitudes are related to dynamical properties through, for example, their close link with vertical stratification. We explore these relationships below, focusing on water properties in the LS region (similar results are obtained for the CNA region; not shown). A high positive correlation (despite a heterogeneous model ensemble) between LS SSS and Atlantic overturning strength ($r=0.66$; $p<0.01$; Fig. 15a), max. MLD ($r=0.70$; $p<0.01$; Fig. 15b), and SPG strength ($r=0.75$; $p<0.01$; Fig. 15c) suggests a key role of surface salinities in the dynamics of the North Atlantic. These correlations are consistent with increased SSS in the SPNA leading to reduced stratification and increased deep water convection and sinking, which in turn reinforces the AMOC and leads to larger salinity transport into the SPNA through the NAC. Kostov et al. (2023) proposes an additional mechanism by which LS SSSs affect AMOC strength, which is consistent with our results. That study shows that negative SSS anomalies in the western LS cause negative density anomalies in the upper ocean in the east SPNA, which in turn lead to a decrease in the southward AMOC transport along the Deep Western Boundary Current. Interestingly, the negative density anomalies in the east SPNA in their analysis are caused by advection of

602 negative salinity anomalies from the western LS, but also by a slowdown of the NAC, which leads to reduced heat loss by the
603 ocean in the east SPNA and thus reduced water mass transformation. The NAC slowdown is associated with SSH anomalies
triggered by the same surface salinity anomalies in the western LS (Kostov et al., 2023; Fig. S3 therein).

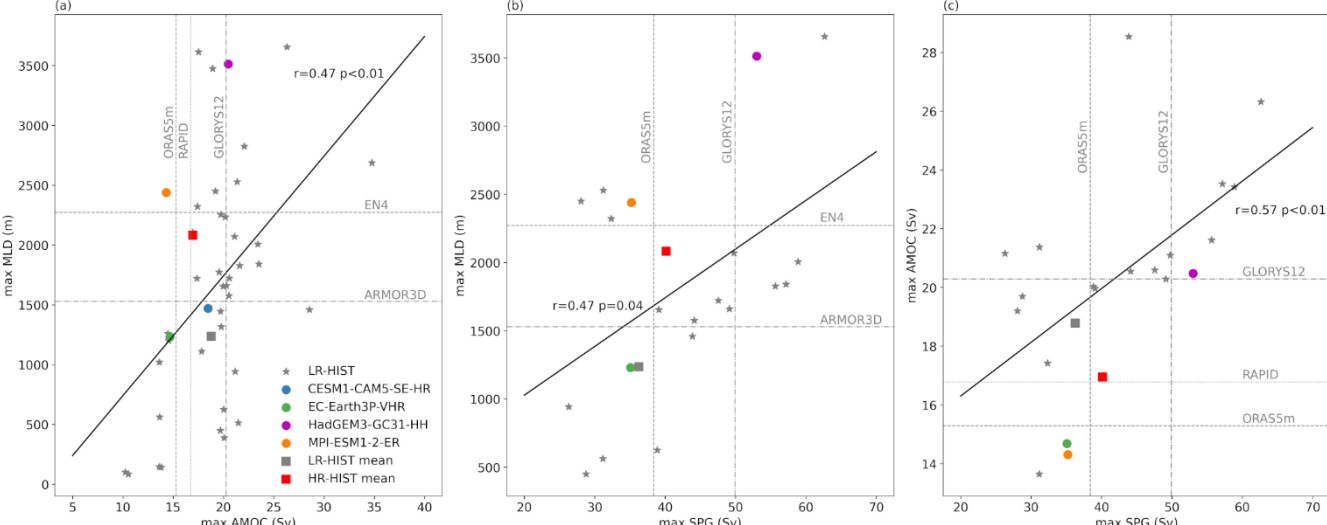

**Figure 14.** Scatterplots of (a) max. Atlantic overturning vs max. MLD, (b) SPG strength vs max. MLD, and (c) SPG strength vs max. overturning. The corresponding Pearson correlation coefficients and their p-values are shown next to the fit lines. Dashed/dotted vertical and horizontal lines indicate observational/reanalysis values. Units for max. MLD, overturning, and SPG strength are m, Sv, and Sv, respectively. Max. MLD is calculated in the LIS box (35º–60º W, 50º–65º N), which is shown in Figs. 7 and 8. Max. SPG strength is calculated as the absolute value of the minimum of the BSF, also in the LIS box. Max. overturning is calculated at 26.5º N. Correlation coefficients and fit lines are based on the composite of the LR-HIST and HR-HIST ensembles (LR-HIST + HR-HIST).

Correlations between LS SSTs and the different North Atlantic dynamical metrics have also been analysed and are lower
and/or not significant compared to those of SSSs (Fig. A1). The positive correlation between LS SST and overturning strength
(r=0.36; p=0.02; Fig A1a) might be explained by increased heat transport to the SPNA by a stronger overturning. Regarding
the correlation between LS SSTs and max. MLD (r=0.48; p<0.01; Fig. A1b), it could be associated with the fact that models
with higher LS SSTs have less sea ice and/or increased surface water mass transformation associated with heat loss to the
atmosphere (Kostov et al., 2023), leading to increased sea surface cooling and thus deeper mixed layers. Alternatively, it could
be associated with models with larger LS SSTs having also larger LS SSSs (Sect. 3.1), which would also lead to reduced
stratification and deeper mixed layers.
Horizontal resolution might play a role in the representation of the relationships between the different dynamical and
physical properties in the North Atlantic through differences in model dynamics. For example, work by Katsman et al. (2018)
shows differences in deep water sinking mechanisms at mesoscale-permitting resolutions (see Sect. 4 for further details), which

might affect relationships involving overturning. In this study we cannot properly assess whether such relationships change

with resolution given the limited size of the HR-HIST ensemble, but future studies might be able to address it as new

mesoscale-resolving simulations become available.

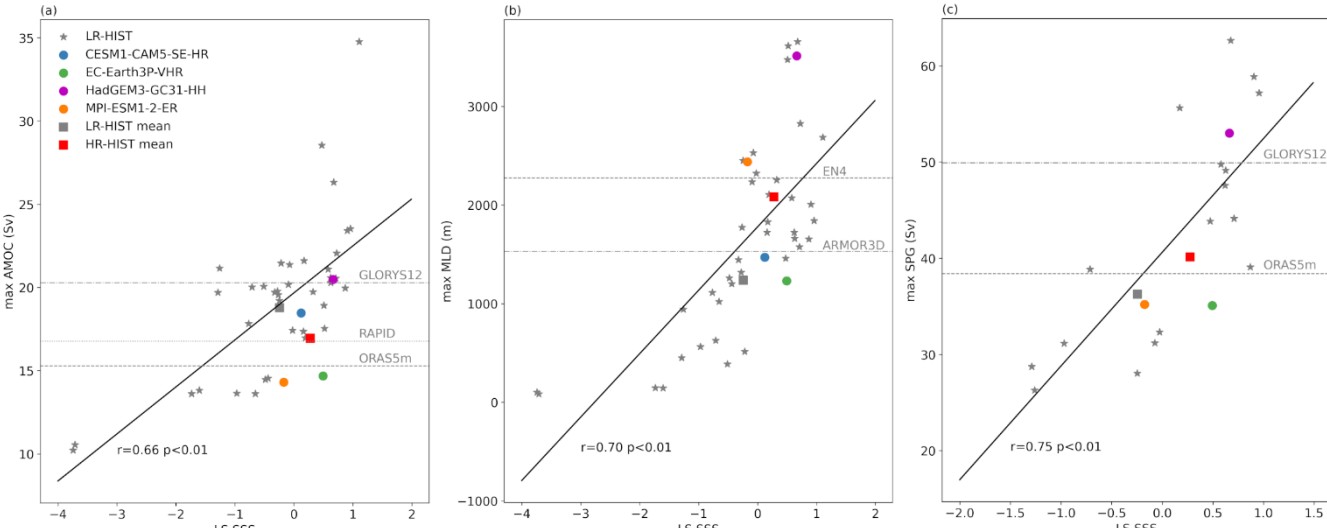

**Figure 15.** Scatterplots of LS SSS vs (a) max. Atlantic overturning, (b) max. MLD, and (c) max. SPG strength. LS SSS biases are calculated

in the LS box (44º–60º W, 52.5º–65º N), which is shown in Fig. 2. Remaining plotting details as in Fig. 14.

To conclude our analysis, we investigate potential relations between SSSs (and SSTs) in the different bias regions identified

in Sect. 3.1, namely those between the NCH and CNA regions (Fig. A2), and between the CNA and LS regions (Fig. A3).

Since the NCH region falls within the GS domain in LR-HIST models, we might expect the NCH and CNA biases to be

correlated for those models, due to their ultimate link with the GS/NAC dynamics. Indeed, SSSs between the two regions show

a significant correlation, if we restrict our analysis to the LR-HIST ensemble (Fig. A2b). Interestingly, when HR-HIST models

are included in the calculations, the SSS correlation between the NCH and CNA decreases/loses its significance ($p=0.06$),

which we argue is due to the fact that NCH is outside the GS domain in HR-HIST models. The correlation in SSTs between

the NCH and CNA regions is not significant for LR-HIST models though, which could be related to some damping of the SST

signal through interactions with the atmosphere (Fig. A2a).

Scatterplots of SSS biases between the LS and CNA regions indicate a strong correlation between them ($r = 0.86$, $p < 0.01$;

Fig. A3b), which suggests a potential link between LS salinity biases and the NAC, through the effect of the NAC on the

northward salinity transport. We note, however, that the LS and CNA are also connected through the SPG circulation, which

could also partly explain why their SSS biases are related. The correlation between the SST biases of the LS and CNA regions

is also significant, although weaker compared to the SSS biases ($r = 0.54$, $p < 0.01$; Fig. A3a), which could be related to a

damping of the SST signal through interactions with the atmosphere, or to mixing with Arctic waters.

## 4 Main conclusions and discussion

In this study, we analyse the impact of increasing horizontal resolution on the representation of the North Atlantic mean state, by comparing two ensembles of coupled historical simulations: four HighResMIP experiments at mesoscale eddy-resolving scales (HR-HIST ensemble; at least 1/10º nominal resolution) and 39 CMIP6 experiments with eddy-parameterized and some eddy-permitting ocean resolutions (LR-HIST ensemble).

The main biases of key thermodynamic and dynamical variables for the North Atlantic are analysed for the two ensembles. In particular we examine i) the main surface temperature and salinity biases; ii) stratification and iii) deep water convection; iv) the representation of the Atlantic overturning streamfunction; and v) the gyre circulations, including the GS, NAC, and the SPG. Additionally, we test the significance of the differences between ensembles, analyse specific model features within the HR-HIST ensemble, and study relationships between dynamical and physical properties in a North Atlantic context. In the following, the main findings of the paper are described and their implications discussed in light of the previous literature.

Three main SST and SSS bias regions are found in the simulations, located at North Cape Hatteras, the Central North Atlantic, and the Labrador Sea, which show differences across the two ensembles. In the NCH region, we find significantly reduced positive temperature and salinity surface biases for the multi-model HR-HIST mean with respect to LR-HIST, associated with a more southward position of the GS separation, in agreement with previous individual model studies (Roberts et al., 2019; Gutjahr et al., 2019; Marzocchi et al., 2015).

Then, the CNA cold and fresh biases in the multi-model LR-HIST mean are also reduced in HR-HIST – as in Gutjahr et al. (2019) and Marzocchi et al. (2015) – which describes a less zonal NAC and a more restricted influence of SPG waters in that region, in closer agreement with observations and reanalysis. Sein et al. (2018) found similar results when comparing HighResMIP coupled simulations obtained with the AWI-CM model, showing that, ultimately, an increase in ocean resolution shifted the NAC path northward, with no significant influence of the atmospheric resolution. However, our bootstrapping analysis indicates that the reduction in the CNA surface biases is not statistically significant in our HR-HIST ensemble (Sect 3.6), as some of the HR-HIST models (MPI-ESM1-2-ER and EC-Earth3P-VHR) still present an overly weak NAC (as represented in the BSF), with a reduced eastward penetration compared to reanalyses (Figs. 12, 13; Sect 3.7).

In the HR-HIST multi-model mean, the LS region stands out for a warm and salty bias. The LR-HIST ensemble mean shows also a warm bias in the LS, although this is weaker in magnitude compared to the one in the HR-HIST ensemble mean. No salty bias is present in the LS in the LR-HIST ensemble mean, although some individual LR-HIST models do show salty biases in that region, comparable in magnitude to those of the HR-HIST ensemble, but their signal is compensated in the multi-model mean by models with biases of opposite sign.

In terms of vertical stratification, we find improved temperature and salinity profiles in the broader LIS box for HR-HIST compared to LR-HIST, with the HR-HIST ensemble mean curve closer in distance and shape to EN4. The warm and salty biases present at the subsurface in both ensembles over most of the column (below ~150 m) are reduced in the HR-HIST

ensemble mean. On the other hand, as already discussed, surface biases (above ~150 m) are more pronounced in the HR-HIST

ensemble mean compared to the LR-HIST ensemble mean.

The origin and dynamical impacts of the LS biases are a current matter of debate (Jackson et al., 2023; Lin et al., 2023;

Bruciaferri et al., 2024; Menary et al., 2015; Roberts et al., 2020). These biases have an effect on LS deep water convection

through their decisive influence on vertical stratification and, therefore, correcting them might help obtain more realistic

present-day AMOC estimates and reliable future projections. Work by Lin et al. (2023), for example, indicates that models

with a strong AMOC present a warmer and saltier LS and experience a larger decrease in AMOC strength in future projections.

In regard of the impacts of the LS temperature and salinity biases, results by Jackson et al. (2023) show significant shoaling

of the LS mixed layers when temperature and salinity in the IS and Icelandic basin subsurface (below 1000 m) in the

HadGEM3-GC1 model are restored to observed values at run time. Thus biases in the LS are linked to biases in the IS and

Icelandic basin, which in turn are influenced by the transport of overflow waters from the Nordic Seas. Ocean models using

fixed vertical levels (z-models), as the ones in this study, present difficulties in correctly representing the temperature and

salinity of the Arctic overflows downslope the Greenland-Scotland Ridge (Bruciaferri et al., 2024; Colombo et al., 2020;

Jackson et al., 2023). Bruciaferri et al. (2024) show that the embedding of local terrain-following coordinates in the area of the

Arctic overflows in ocean models leads to improved stratification in the IS and Icelandic basin (which might thus reduce biases

in the LS) and improved transport in the AMOC lower limb.

Our study hints that LS and CNA biases might be actually related to each other through northward salinity/heat transport

by the NAC, as supported by the correlations between the SSS (and SST) biases of those two regions. Note that northward

transports depend both on NAC strength as well as path (Jackson et al., 2023).  Studies such as Chang et al. (2020) and Roberts

et al. (2019) report increased heat transport by the AMOC in mesoscale eddy-resolving models, further supporting the idea of

increased northward transport as a potential origin for the LS biases. A connection between LS salinity biases and the NAC is

further supported by Kostov et al. (2023; Sect. 3.8), who suggests a positive feedback exists between LS salinities and NAC

strength. Furthermore, we hypothesize that the increased (reduced) biases at the surface (subsurface) for HR-HIST in the LIS

area could also be related to the fact that ocean mesoscale eddies increase vertical (upwards) heat and salt transports in the

ocean (Hewitt et al., 2017).

Despite the surface biases described above, density profiles in the LIS area, which is the key property controlling the

vertical mixing, are also improved in the HR-HIST mean with respect to LR-HIST, showing comparatively reduced

stratification. This is consistent with deeper, although not significantly deeper mixed layers in the LS and along the east

Greenland coast in HR-HIST compared to LR-HIST. The deeper mixed layers along the east Greenland coast and also in the

Nordic Seas in HR-HIST compared to LR-HIST, are in better agreement both with EN4- and ARMOR3D-derived values.  In

the LS the wide range of observation-derived MLD estimates, with values of 1000–1200 m in ARMOR3D and 2000–2200 m

in EN4, makes model assessment challenging. Analyses from additional observational studies show values between 1100–

1500 m (Yashayaev and Loder, 2016) and down to 1400–1800 m (Holte et al., 2017) in the LS, suggesting that HR-HIST

mean values for the LS (1800–2000 m) might be slightly too deep, and LR-HIST mean values (1000–1200 m) slightly too shallow.

The AMOC streamfunction in HR-HIST exhibits a better fit with RAPID observations and ORAS5m reanalysis compared to LR-HIST, although it is weaker than in GLORYS12. Additionally, the HR-HIST AMOC streamfunction presents sharper features, as shown, e.g., in Sein et al. (2018), better resembling reanalysis data. Nevertheless, the AMOC in both the LR-HIST and HR-HIST ensemble means is too shallow compared to RAPID and reanalyses, in agreement with previous modeling studies by Roberts et al. (2020) and Hirschi et al. (2020).

The role of resolution in Atlantic overturning strength is a current matter of debate in the literature, with different individual model studies pointing at different results. Winton et al. (2014) report AMOC strengthening with increased ocean resolution for the GFDL CM2.6 and CM2.5FLOR models at 0.1º and 1º resolution, respectively. They also find AMOC strength is sensitive to horizontal friction and mesoscale eddy parameterizations. Hewitt et al. (2016) show strengthening in the mean AMOC at a concomitant increase in ocean (from 1/4º to 1/12º) and atmospheric resolution (from 60 to 25 km) in the GC2.1 model. Similar results are found by Moreno-Chamarro et al. (2025) for the EC-Earth3P model when increasing ocean and atmospheric resolution from 0.25º to 0.08º and from ~54 km to ~12 km, respectively. On the other hand, a study assessing the separate effects of enhanced atmospheric and ocean resolution on AMOC behaviour with the AWI-CM model, describes a weakening at increased atmospheric resolution (from 1.9º to 0.9º) associated with reduced winds, but both a weakening at ~45º N and a strengthening at ~20º N related to ocean grid refinement (from 1º to 1/4° nominal resolutions, and the latter with grid refinements in eddy-rich areas; Sein et al., 2018). Furthermore, Gutjahr et al. (2019) show little difference in AMOC strength between MPI-ESM1-2-HR and MPI-ESM1-2-ER, which use the same atmosphere and vertical mixing parameterization yet different ocean resolution (0.4º vs 0.1º, respectively). That study also shows AMOC can be very sensitive to the vertical mixing scheme.

Multimodel studies on this topic have also been conducted (Roberts et al., 2020; Hirschi et al., 2020). Hirschi et al. (2020) analyze 28 model configurations (22 ocean-only and six coupled configurations) with ocean resolutions ranging from 2º to 0.05º and find increased AMOC strength at eddy-resolving scales (their Fig. 2). Roberts et al. (2020) compare the AMOC streamfunction in HighResMIP simulations with seven different coupled models, not finding a consistent effect of enhancing ocean and/or atmospheric resolution on the AMOC strength in depth-space. This also applies for the two simulations at mesoscale eddy-resolving scales in that study, performed with HadGEM3-GC31 and CESM1.3 (our CESM1-CAM5-SE), the first showing a stronger AMOC in depth-space than its low resolution counterpart, and the second showing a weaker AMOC instead. Interestingly, in that study results converge towards a stronger AMOC in the mesoscale eddy-resolving simulations when density coordinates are used instead.

Our results show a weaker Atlantic overturning in depth-space at mesoscale eddy-resolving scales, although the difference in strength between the HR-HIST and LR-HIST ensembles is not significant. We note that our AMOC profiles of mesoscale eddy-resolving models at 26.5º N display similar values to those in Roberts et al. (2020) and Hirschi et al. (2020). The differences in our results compared to those of Roberts et al. (2020) and Hirschi et al. (2020) lie rather in the characteristics of

the low resolution model ensembles, which in those studies show considerable lower overturning values compared to ours, probably in relation with the high sensitivity of AMOC to model schemes and parameterizations (see e.g. Winton et al., 2014; Gutjahr et al., 2019 above). Nevertheless, all three multimodel studies – Hirschi et al. (2020), Roberts et al. (2020), and our study – point at an improved AMOC mean-state representation at enhanced resolution.

Whereas the SPG is stronger in the HR-HIST ensemble mean compared to the LR-HIST mean, in line with results by Hirschi et al. (2020) comparing a range of ocean resolutions from 1º to 0.08º, our results suggest that this strengthening is not significant. We note though differences in the structure between resolutions, since near the continental boundaries, the SPG is narrower and locally stronger in the HR-HIST mean compared to the LR-HIST mean (e.g. in the LS). The link between model resolution and SPG strength and structure should be further investigated in future studies employing a larger HR-HIST ensemble size.

Although a link exists between AMOC strength and SPNA densities/mixed layers (Ortega et al., 2021; Menary et al., 2020; Martin-Martinez et al., 2025; and our study), in our study, the deeper mixed layers in the multi-model HR-HIST mean with respect to LR-HIST despite a weaker Atlantic overturning, reflect a different representation of deep water sinking mechanisms in high resolution models, as described in Katsman et al. (2018). That study shows that deep water sinking in mesoscale eddy-permitting models occurs only at the continental slopes – at the boundary current of the SPG – and not also in the open ocean where MLDs reach their maximum depths, as in 1º ocean models. The sinking mechanism described for mesoscale eddy-permitting models can be explained by buoyancy loss along the boundary current path, triggering a cross-shore baroclinic flow and subsequent sinking forced by mass conservation (Katsman et al., 2018; Straneo, 2006; Spall and Pickart, 2001). The more realistic SPG structure near the continental boundary e.g. in the LS, and more realistic MLDs along the East Greenland Current, might thus be key factors for the improvement in modeled AMOC strength in HR-HIST.

To summarize, we find significantly reduced surface biases in the NCH region in our HR-HIST ensemble, compared to LR-HIST. Although the NAC path and strength are generally improved in the HR-HIST ensemble mean, some of the HR-HIST models still present an overly weak NAC. In terms of vertical stratification in the LIS area, the HR-HIST ensemble is significantly closer to EN4 observations, compared to the low resolution ensemble. Additionally, the representation of deep water convection in the HR-HIST ensemble is in better agreement with observation-derived estimates in the East Greenland Current and also in the Nordic Seas, whereas in the LS the range of observational values is wide and HR-HIST estimates fall at its upper end. Finally, the Atlantic overturning streamfunction is significantly closer to RAPID observations at 26.5º N.

The AMOC is a fundamental element in global climate through its role in 1) the distribution of heat to high latitudes and 2) the carbon cycle, both leading to important impacts on the atmosphere. Working towards an improved representation of the AMOC, as achieved in mesoscale-resolving models, is therefore paramount to produce more reliable climate change projections. Next efforts in North Atlantic climate modelling should aim at further investigating the origin of the SPNA biases, at embedding parameterizations of submesoscale processes, improving the representation of transports from the Nordic Seas overflows, and further refining the representation of the NAC in mesoscale-resolving models.

## Appendix A: Additional figures and tables

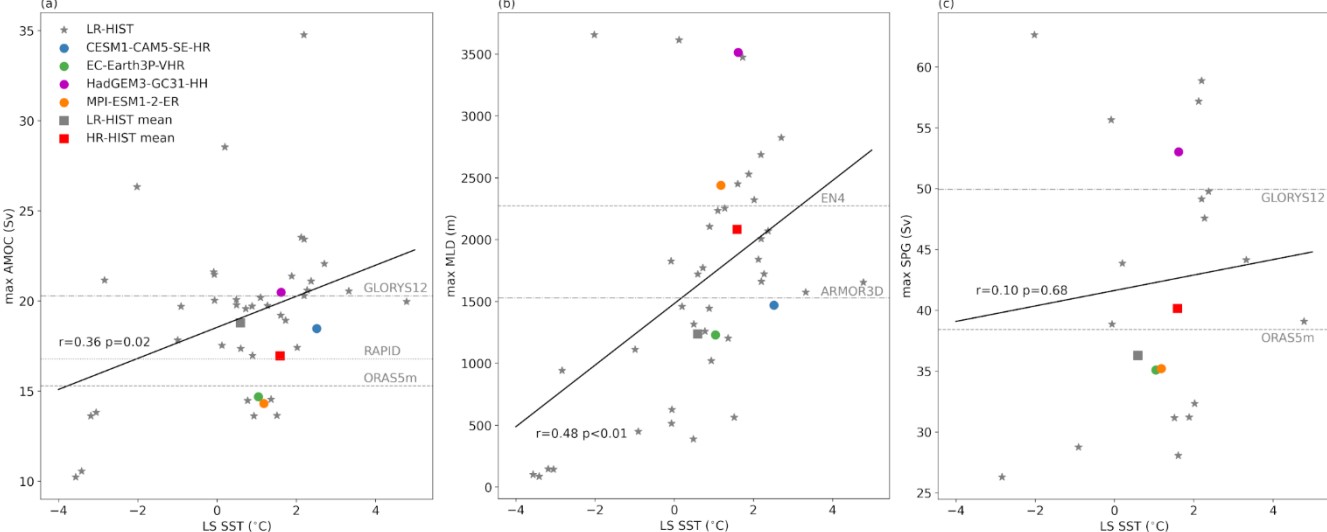

**Figure A1.** Scatterplots of LS SST vs (a) max. Atlantic overturning streamfunction at 26.5º N, (b) max. MLD, and (c) max. SPG strength. LS SST biases are calculated in the LS box (44º–60º W, 52.5º–65º N), which is shown in Fig. 2. Remaining plotting details as in Fig. 14.

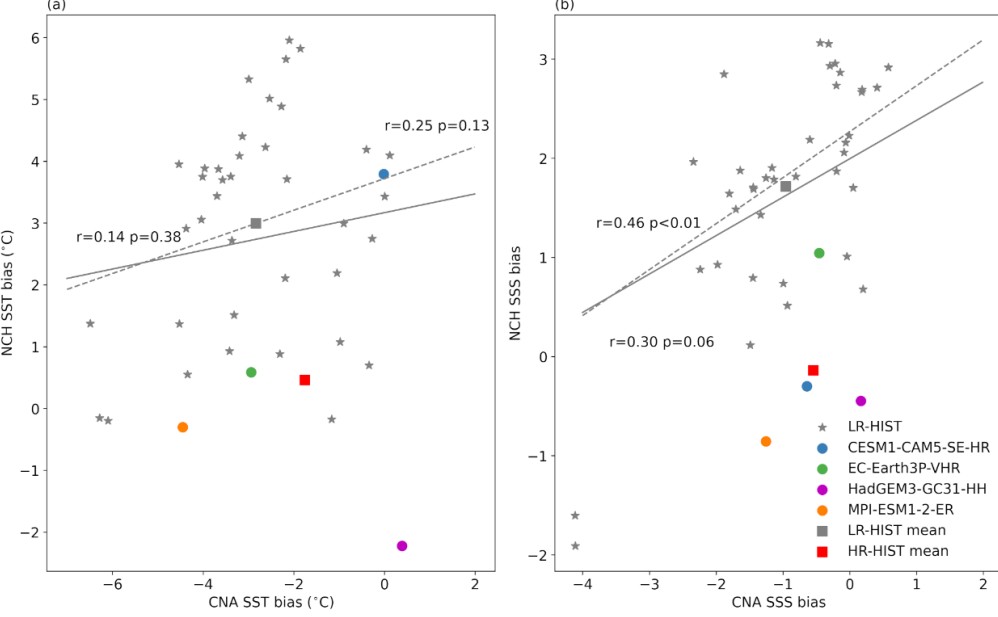

**Figure A2.** Scatterplots of (a) SST (in ºC) and (b) SSS biases between the Central North Atlantic (CNA) and North Cape Hatteras (NCH) regions defined in Fig. 2. Biases are calculated as spatially averaged temporal means in the model minus the corresponding EN4 values in each of the selected boxes. The corresponding correlation coefficients and their p-values are shown next to the fit lines. Dashed lines are regression lines obtained after removal of HR-HIST models. Note: all regression lines and values in (b) are calculated excluding the two outliers at the bottom left of the figure.

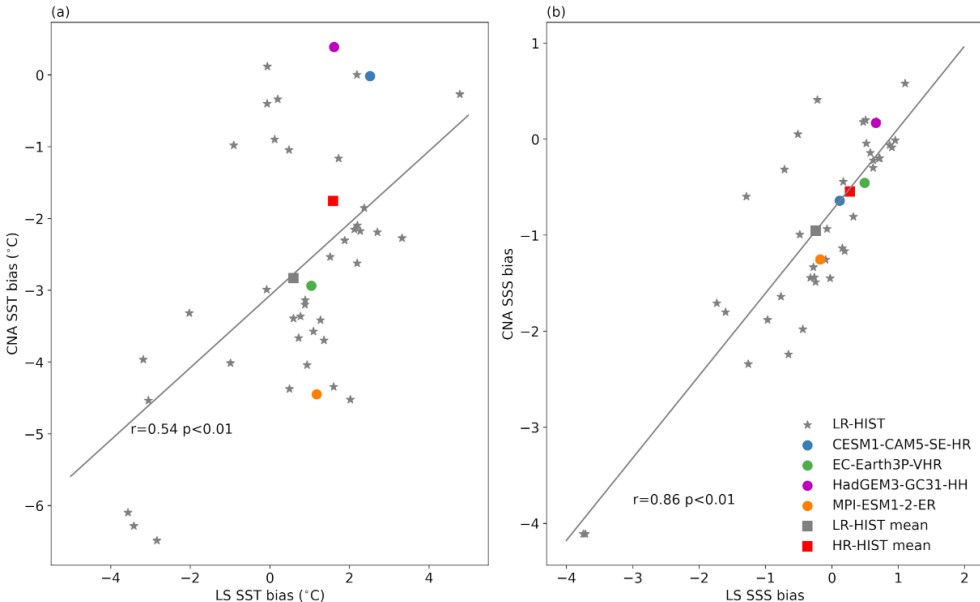

**Figure A3.** Scatterplots of (a) SST (in °C) and (b) SSS biases between the Labrador Sea (LS) and Central North Atlantic (CNA) regions defined in Fig. 2. The corresponding Pearson correlation coefficients and their p-values are shown next to the fit lines. Correlation coefficients and fit lines are based on the composite of the LR-HIST and HR-HIST ensembles (LR-HIST + HR-HIST).

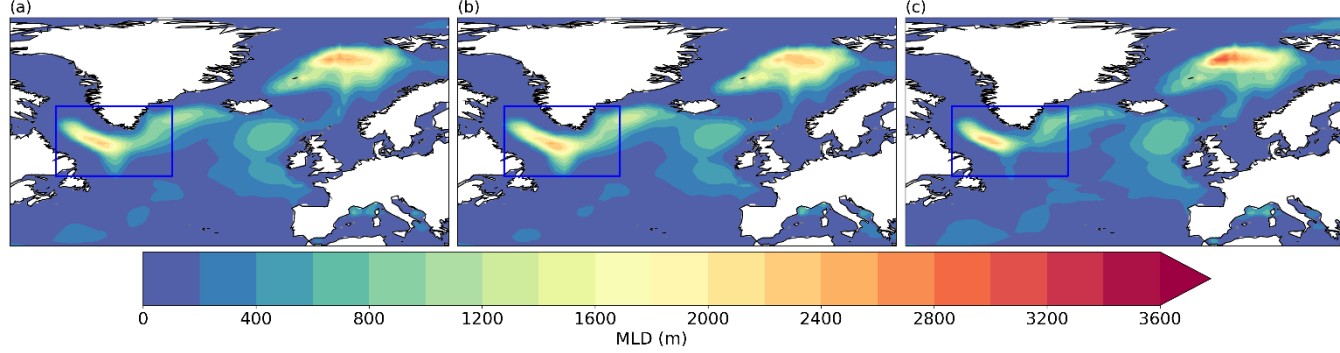

**Figure A4.** EN4-derived March MLD (in m) for the (a) 1980–2014, (b) 1980–1999 (pre-Argo), and (c) 2000–2014 (Argo) periods. Calculated from temperature and salinity as in Fig. 8.

| Metric | CI with a total LR-HIST ensemble size | CI with a reduced LR-HIST ensemble size | min. LR-HIST ensemble size required for 95% sign. |
|---|---|---|---|
| LS SST (ºC) | **[0.21 1.86]** | [-0.79 3.11] | 19 |
| LS SSS | **[0.04 1.00]** | [-0.39 1.78] | 25 |
| CNA SST (ºC) | [-1.05 3.07] | [-1.43 3.66] | – |
| CNA SSS | [-0.18 1.01 ] | [-0.63 1.70] | – |
| NCH SST (ºC) | **[-4.58 -0.16]** | [-5.19 0.23] | 8 |
| NCH SSS | **[-2.54 -1.02]** | **[-2.99 -0.38]** | 4 |
| max AMOC (Sv) | [-5.10 0.66] | [-7.75 2.61] | – |
| AMOC RMSE (Sv) | **[-1.59 -0.60]** | **[-2.69 -0.05]** | 4 |
| AMOC correl | **[0.01 0.11]** | [-0.02 0.20] | 14 |
| temp profile RMSE (ºC) | **[-1.31 -0.39]** | **[-1.57 -0.12]** | 4 |
| temp profile correl | [-0.15 0.16 ] | [-0.16 0.20] | – |
| salt profile RMSE | **[-0.41 -0.09]** | [-0.68 0.04] | 8 |
| salt profile correl | **[0.01 0.04]** | [-0.00 0.07] | 6 |
| density profile RMSE (kg m$^{-3}$ ) | **[-0.26 -0.06]** | [-0.48 0.03 ] | 8 |
| density profile correl | **[0.01 0.04 ]** | [-0.00 0.06] | 7 |
| max MLD (m) | [-310.52 1530.64] | [-703.04 1858.65] | – |
| max SPG (Sv) | [-11.17 9.94] | [-17.10 14.09] | – |

**Table A1:** (First column) Single numeric metrics analysed, with units in parenthesis; (Second column) 95% confidence interval (CI) of the differences in means between the HR-HIST and LR-HIST ensembles, calculated from a distribution of bootstrapping samples with repetition. The size of the samples coincides with the total size of their respective ensembles; (Third column) Analogous to the second column but in this case the size of the LR-HIST samples coincides with the total size of the HR-HIST ensemble; (Fourth column) Minimum size of the LR-HIST samples in the bootstrapping required to obtain a CI not containing the value zero. Text in bold indicates when this is the case.

| | ocean component | ocean grid | atm. component | atm. grid |
|---|---|---|---|---|
| **HR-HIST** | | | | |
| CESM1-CAM5-SE-HR | POP2 | 1/10°; tripolar; 3600x2400 lon/lat; 62 levels; | CAM5.2 | 25 km; 30 levels; |
| EC-Earth3P-VHR | NEMO3.6 | 1/12°; ORCA12 tripolar; 4322 x 3059 lon/lat; 75 levels; | IFS cy36r4 | 16 km; 91 levels; |
| HadGEM3-GC31-HH | NEMO-HadGEM3-GO6.0 | 1/12°; eORCA12 tripolar; 4320 x 3604 lon/lat; 75 levels; | MetUM-HadGEM3-GA7.1 | 50 km; 85 levels; |
| MPI-ESM1-2-ER | MPIOM | 1/10°; TP6M tripolar; 3602 x 2394 lon/lat; 40 levels; | ECHAM6.3 | 103 km; 95 levels; |
| **LR-HIST** | | | | |
| ACCESS-CM2 | ACCESS-OM2 | 100 km; GFDL-MOM5 tripolar; 360 x 300 lon/lat; 50 levels; | MetUM-HadGEM3-GA7.1 | 250 km; 85 levels; |
| ACCESS-ESM1-5 | ACCESS-OM2 | 100 km; MOM5 tripolar; 360 x 300 lon/lat; 50 levels; | HadGAM2 | 250 km; 38 levels; |
| CAS-ESM2-0 | LICOM2.0 | 100 km; 362 x 196 lon/lat; 30 levels; | IAP AGCM 5.0 | 100 km; 35 levels; |
| CESM2 | POP2 | 100 km; gx1v7 displaced pole; 320x384 lon/lat; 60 levels; | CAM6 | 100 km; 32 levels; |
| CESM2-FV2 | POP2 | 100 km; gx1v7, displaced pole; 320 x 384 lon/lat; 60 levels; | CAM6 | 250 km; 32 levels; |
| CESM2-WACCM | POP2 | 100 km; gx1v7 displaced pole; 320 x 384 lon/lat; 60 levels; | WACCM6 | 100 km; 70 levels; |
| CESM2-WACCM-FV2 | POP2 | 100 km; gx1v7 displaced pole; 320 x 384 lon/lat; 60 levels; | WACCM6 | 250 km; 70 levels; |
| CIESM | CIESM-OM | 100 km; mod. POP2 displ. pole; 320 x 384 lon/lat; 60 levels; | CIESM-AM (modified CAM5) | 100 km; 30 levels; |
| CMCC-CM2-HR4 | NEMO3.6 | 25 km; ORCA0.25; 1442 x 1051 lon/lat; 50 levels; | CAM4 | 100 km; 26 levels; |
| CMCC-CM2-SR5 | NEMO3.6 | 100 km; ORCA1 tripolar; 362 x 292 lon/lat; 50 levels; | CAM5.3 | 100 km; 30 levels; |
| CMCC-ESM2 | NEMO3.6 | 100 km; ORCA1 tripolar; 362 x 292 lon/lat; 50 levels; | CAM5.3 | 100 km; 30 levels; |
| CanESM5 | NEMO3.4.1 | 100 km; ORCA1 tripolar; 361 x 290 lon/lat; 45 levels; | CanAM5 | 500 km; 49 levels; |
| CanESM5-1 | NEMO3.4.1 | 100 km; ORCA1 tripolar; 361 x 290 lon/lat; 45 levels; | CanAM5.1 | 500 km; 49 levels; |
| E3SM-1-1 | MPAS-Ocean (v6.0) | 30-60 km; oEC60to30 unstructured; 60 levels; | EAM (v1.1) | 100 km; 72 levels; |

**Table B1.** Overview of individual models used in the current study. Ocean grid details include: nominal resolution; grid type; size of
horizontal grid; and number of vertical levels. Expanded version of Table 1 (part1).

| | ocean component | ocean grid | atm. component | atm. grid |
|---|---|---|---|---|
| E3SM-1-1-ECA | MPAS-Ocean (v6.0) | 30-60 km; oEC60to30 unstructured; 60 levels; | EAM (v1.1) | 100 km; 72 levels; |
| EC-Earth3 | NEMO3.6 | 100 km; ORCA1 tripolar; 362 x 292 lon/lat; 75 levels; | IFS cy36r4 | 100 km; 91 levels; |
| EC-Earth3-AerChem | NEMO3.6 | 100 km; ORCA1 tripolar; 362 x 292 lon/lat; 75 levels; | IFS cy36r4 | 100 km; 91 levels; |
| EC-Earth3-CC | NEMO3.6 | 100 km; ORCA1 tripolar; 362 x 292 lon/lat; 75 levels; | IFS cy36r4 | 100 km; 91 levels; |
| EC-Earth3-Veg | NEMO3.6 | 100 km; ORCA1 tripolar; 362 x 292 lon/lat; 75 levels; | IFS cy36r4 | 100 km; 91 levels; |
| EC-Earth3-Veg-LR | NEMO3.6 | 100 km; ORCA1 tripolar; 362 x 292 lon/lat; 75 levels; | IFS cy36r4 | 250 km; 62 levels; |
| FGOALS-f3-L | LICOM3.0 | 100 km; tripolar; 360 x 218 lon/lat; 30 levels | FAMIL2.2 | 100 km; 32 levels; |
| FGOALS-g3 | LICOM3.0 | 100 km; tripolar; 360 x 218 lon/lat; 30 levels; | GAMIL3 | 250 km; 26 levels; |
| GFDL-CM4 | GFDL-OM4p25 | 25 km; GFDL-MOM6 tripolar; 1440 x 1080 lon/lat; 75 levels; | GFDL-AM4.0.1 | 100 km; 33 levels; |
| GFDL-ESM4 | GFDL-OM4p5 | 50 km; GFDL-MOM6 tripolar; 720 x 576 lon/lat; 75 levels; | GFDL-AM4.1 | 100 km; 49 levels; |
| GISS-E2-2-G | GISS Ocean | 100 km; GO1; 360 x 180 lon/lat; 40 levels; | GISS-E2.2 | 250 km; 102 levels; |
| ICON-ESM-LR | ICON-O | 50 km; icosahedral/triangles; 40 levels; | ICON-A | 250 km; 47 levels; |
| INM-CM4-8 | INM-OM5 | 100 km; shifted North Pole; 360 x 318 lon/lat; 40 levels; | INM-AM4-8 | 100 km; 21 levels; |
| INM-CM5-0 | INM-OM5 | 50 km; shifted North Pole; 720 x 720 lon/lat; 40 levels; | INM-AM5-0 | 100 km; 73 levels; |
| IPSL-CM6A-LR | NEMO-OPA | 100 km; eORCA1.3 tripolar; 362 x 332 lon/lat; 75 levels; | LMDZ | 250 km; 79 levels; |
| IPSL-CM6A-LR-INCA | NEMO-OPA | 100 km; eORCA1.3 tripolar; 362 x 332 lon/lat; 75 levels; | LMDZ | 250 km; 79 levels; |
| MIROC6 | COCO4.9 | 100 km; tripolar; 360 x 256 lon/lat; 63 levels; | CCSR AGCM | 250 km; 81 levels; |
| MPI-ESM-1-2-HAM | MPIOM1.63 | 250 km; bipolar GR1.5; 256 x 220 lon/lat; 40 levels; | ECHAM6.3 | 250 km; 47 levels; |
| MPI-ESM1-2-HR | MPIOM1.63 | 50 km; tripolar TP04; 802 x 404 lon/lat; 40 levels; | ECHAM6.3 | 100 km; 95 levels; |

(The row label **LR-HIST** spans the left margin of all rows above.)

**Table B1.** Continuation (part2).

|  |  | ocean component | ocean grid | atm. component | atm. grid |
|---|---|---|---|---|---|
| **LR-HIST** | MPI-ESM1-2-LR | MPIOM1.63 | 250 km; bipolar GR1.5; 256 x 220 lon/lat; 40 levels; | ECHAM6.3 | 250 km; 47 levels; |
|  | MRI-ESM2-0 | MRI.COM4.4 | 100 km; tripolar; 360 x 364 lon/lat; 61 levels; | MRI-AGCM3.5 | 100 km; 80 levels; |
|  | NorCPM1 | MICOM1.1 | 100 km; displaced pole; 320 x 384 lon/lat; 53 levels; | CAM-OSLO4.1 | 250 km; 26 levels; |
|  | NorESM2-LM | MICOM | 100 km; tripolar; 360 x 384 lon/lat; 70 levels; | CAM-OSLO | 250 km; 32 levels; |
|  | NorESM2-MM | MICOM | 100 km; tripolar; 360 x 384 lon/lat; 70 levels; | CAM-OSLO | 100 km; 32 levels; |
|  | SAM0-UNICON | POP2 | 100 km; displaced pole; 320 x 384 lon/lat; 60 levels; | CAM5.3 with UNICON | 100 km; 30 levels; |

**Table B1.** Continuation (part3).

*Code and data availability.* The ESMValTool code will be made available in the revised version. Model data used in this study can be found on the Earth System Grid Federation (ESGF) site (https://esgf-ui.ceda.ac.uk/cog/search/cmip6-ceda/), except for: 1) HadGEM3-GC31-HH AMOC and BSF data, which are available upon request via the CEDA-JASMIN platform (https://www.ceda.ac.uk/services/jasmin/), 2) CESM1-CAM5-SE-HR AMOC data, which are available from NCAR's Climate and Global Dynamics lab (https://www.cgd.ucar.edu/) upon request, 3) MPI-ESM1-2-ER data, which are archived by the Max Planck Institute for Meteorology and can be obtained by contacting publications@mpimet.mpg.de, and 4) EC-Earth3P-VHR data, which will be published on ESGF soon, while at the moment are available upon request from the Barcelona Supercomputing Center (BSC). ORAS5m data are available upon request from the ECMWF file storage system. AVISO absolute dynamic topography data (MADT-H) can be directly downloaded from this link https://www.aviso.altimetry.fr/en/data/products/sea-surface-height-products/global/gridded-sea-level-anomalies-mean-and-climatology.html. RAPID AMOC data can be downloaded from https://rapid.ac.uk/data/. EN4 data are available from https://www.metoffice.gov.uk/hadobs/en4/index.html. ARMOR3D and GLORYS12 data can be downloaded from https://doi.org/10.48670/moi-00052 and https://doi.org/10.48670/moi-00021, respectively.

*Author contributions.* AF carried out the analysis with ESMValTool and wrote the manuscript. SLT and EMM provided technical support and guidance in ESMValTool. SLT made improvements for memory usage in ESMValTool. EMM wrote the ESMValTool code to calculate density and mixed layer from temperature and salinity, and GLORYS12 BSF and AMOC streamfunctions from velocities. EMC performed the ECEarth-3P HighResMIP runs, provided scientific guidance, as well as

input to the draft. PO provided scientific guidance, as well as input to the draft. Xia Lin provided input to the draft. MS, AF, PAB, and DK performed the data assemblage and formatting.

*Competing interests.* The authors declare that they have no conflict of interest.

*Acknowledgements.* The authors thank Nikolay Koldunov for his ESMValTool arctic ocean diagnostics (Khosravi et al., 2022) shared on GitHub (https://github.com/ESMValGroup/ESMValTool/tree/main//esmvaltool/diag_scripts/arctic_ocean) under an Apache 2.0 license. We would also like to thank Steffen Tietsche and Hao Zuo for kindly providing the ORAS5m reanalysis data; Dian Putrasahan and Katja Lohmann for sharing all the MPI-ESM1-2-ER data; Malcolm Roberts for the HadGEM3-GC31-HH AMOC and BSF data; as well as Gokhan Danabasoglu, James Hurrell, Frederic Castruccio, and Gary Strand for the CESM1-CAM5-SE-HR AMOC data. Additionally, we thank the editor, Karen Heywood, and two anonymous reviewers for their guide and constructive comments that have greatly contributed to improving the manuscript. Further acknowledgements go to Albert Vila and David Vicente for the technical and HPC support at BSC, and to the BSC itself for providing the computing resources. AF thanks Malcolm Roberts and Michael Lai for the science discussions and seminars at Met Office.

*Financial support.* This research has been funded by the Spanish STREAM project (grant no. PID2020-114746GB-I00) and the Horizon Europe EERIE project (grant no. 101081383).

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
