# Peer review of "The North Atlantic mean state in mesoscale eddy-resolving coupled 1"

_EGUsphere, 2025_

## Referee Comment (RC1)

**Summary**

Frigola et al. investigate the role of resolving mesoscale ocean eddies on the representation of the mean state & circulation of the North Atlantic Ocean in coupled climate models. The authors present a largely qualitative comparison of four coupled historical simulations with nominal ocean resolutions of at least 1/10° to an ensemble of 39 coupled simulations configured at coarser horizontal resolution. The study concludes that the vertical stratification & deep convection in the subpolar North Atlantic, and both the meridional overturning and barotropic circulations agree more closely with ocean observations at mesoscale eddy-resolving resolution. The manuscript is generally well written and includes a valuable final discussion; however, I have significant concerns regarding: (1) the use of the phrase 'eddy-resolving', (2) the ocean observations and reanalysis products used to evaluate model performance, and (3) the originality of the study & its wider implications. I would not recommend the manuscript for publication until major revisions have been made to address each of the comments below.

**General Comments**

- ***Use of 'eddy-resolving':*** In both the title and throughout the manuscript, the authors use 'eddy-resolving' to refer to coupled model simulations with sufficiently fine horizontal ocean resolution to resolve **mesoscale** eddies. Given the recent emergence of both submesoscale permitting / resolving ocean model configurations (e.g., Chassignet & Xu, 2017; Lévy et al., 2010; Pennelly & Myers, 2020; Pennelly & Myers, 2022; Li et al., 2023) and a growing awareness that representing or resolving submesoscale processes is integral for the accurate simulation of North Atlantic mean state (see Jackson et al, 2023 for a review), I would strongly recommend that the authors refine their use of 'eddy-resolving' to 'mesoscale resolving' throughout. Similarly, I would suggest revising the manuscript title to: 'The North Atlantic Ocean mean state in mesoscale eddy-resolving coupled models: a multi-model study' or equivalent. Furthermore, the authors should explicitly address the role of submesoscale features and the implications of their (justifiable) absence in current generation coupled climate models for the North Atlantic mean state; for example, their important role in restratifying the Labrador Sea (Clément et al., 2023; Frajka-Williams et al., 2014) and reducing deep convection (e.g., Tagklis et al. (2020)) and Gulf Stream penetration (e.g., Chassignet and Xu (2017) and Chassignet et al. (2020)). The manuscript would also benefit from greater effort to contextualise inter-model differences within the LR & HR ensembles; for example, do all HR simulations use the same z-level vertical coordinate system & what impact would any difference have on the representation of the overflows (e.g., Colombo et al.,

2020, Bruciaferri et al., 2024) and Labrador Sea stratification downstream (MacGilchrist et al. 2020).

▪ **_Datasets used for model validation:_** My second major concern is both the choice of ocean observations and reanalysis datasets used to validate the ocean model components & the details absent from their methodology descriptions. More specifically, I could not find justification for why a coarse resolution ocean analysis product (EN4.2.2) is used to validate mesoscale-resolving ocean models, when at least eddy-permitting resolution products are available (e.g., ARMOR3D [https://doi.org/10.48670/moi-00052] or ASTE [Nguyen et al., 2021]). While no observational product is a 'true' representation of reality, I would argue it is more appropriate to compare the property fields simulated in mesoscale-resolving models with observational products that can, at least partially, represent them. Similarly, the authors could have used the mesoscale-resolving GLORYS12 ocean reanalysis product (https://doi.org/10.48670/moi-00021 - on its original NEMO grid) rather than the eddy-permitting ORAS5m reanalysis product to validate the mean meridional overturning stream function in depth-space. There are also some important details missing from Section 2.3 of the methodology; for example, are model property fields regridded onto the observations or vice versa for validation? And what type of interpolation is used: bilinear, conservative etc? The current use of interpolating colour contour plots does not make this obvious to readers.

▪ **_Original contribution to our understanding_**: My final concern is regarding the manuscript's original contribution to our understanding of the representation of the North Atlantic Ocean in coupled climate model simulations. The authors do a good job of placing their largely qualitative findings into wider context in Section 4, however, I still remain unsure which of the study's findings are original since the impact of model resolution on sea surface biases is addressed in Roberts et al., 2019, Gutjahr et al., 2019 & Marzocchi et al., 2015, and the strength and structure of the AMOC in Talandier et al., 2014, Roberts et al., 2020, Hirschi et al., 2019, Jackson et al., 2019, Jackson et al. 2023 (see references within) and Reintges et al., 2024. To progress beyond identifying differences between ensembles and ocean observations, the study should place greater emphasis on the reasons why these differences exist, including those differences between ensemble members; for example, why is HadGEM3-GC3.1-HH often an outlier in the HR ensemble? The authors begin to address this in Section 4 by identifying an interesting 'potential link between LS salinity biases and the NAC, through the effect of the NAC on the northward salinity transport' and I would strongly encourage them to pursue this further since developing

diagnostics to better understand common model biases would be a valuable contribution of this research.

**Specific Comments**

**Abstract**

Lines 10: Suggest clarifying what 'standard resolution models' are? This description could be clearer for readers.

**Introduction**

Line 39-45: Suggest revising this paragraph from one long sentence to demonstrate the interconnectivity between water mass processes. As it stands, deep convection, surface forced water mass transformation and densification along the SPG boundary are highlighted separately, yet both deep convection and boundary current densification are a result of surface forced water mass transformation. It may be beneficial to frame this discussion in terms of surface forced water mass transformation and mixing and their importance for deep convection and dense water formation – and the role of horizontal ocean model resolution in representing these processes.

Lines 46-55: Suggest including a brief discussion of submesoscale eddies in this paragraph and using mesoscale-resolving models to be more precise (see general comments above).

Lines 59-61: The Introduction appears to depend heavily on the single model study of Marzocchi et al. (2015), however, the role of ocean model resolution on the Gulf Stream position is also explored in the more recent studies of Chassignet and Xu (2017) and Chassignet et al. (2020). Suggest extending the references cited here.

Lines 78-79: Suggest rephrasing this sentence to more accurately reflect the number of multi-model comparisons that have been performed; for example, Jackson et al. (2022), Jackson et al. (2023), Reintges et al. (2024) all consider coupled climate models in a North Atlantic context.

**Methods**

Lines 112-119: Are the three-dimensional temperature and salinity fields used in the study stored on the original model grid or the regularly interpolated tracer fields? Here and throughout, suggest being more precise in the use of AMOC. The AMOC is a phenomenon and the overturning stream function in depth-space is a diagnostic used to understand one aspect of this phenomenon. Suggest using vertical overturning or overturning in depth-space throughout since the diapycnal overturning is not considered in this study (although I would argue is a more relevant diagnostic to

consider due to its close relationship to sea surface property biases explored later in this study).

Lines 141-144: The authors make a strong case for diagnosing the MLD from the time-averaged potential density anomaly field following de Boyer Montegut et al. (2004), so I was surprised that the authors did not then compare this result to the available de Boyer Montegut et al. (2004) MLD climatology.

Lines 152-156: How is the AMOC & barotropic stream function calculated in the ORAS5m reanalysis data? Is the calculation performed on the original model grid or using interpolated model fields?

**Results**

Figures 2-4: Here and throughout the manuscript text, suggest discussing the statistical significance of the differences between the LR & HR ensembles. For example, there is considerable discrepancies between SST & SSS bias within the HR ensemble, especially around the NAC. Is the improvement in SST bias in the Central North Atlantic region in the HR ensemble simply due to the warm bias exhibited by HadGEM3-GC31-HH counteracting the cold biases in the other ensemble members?

Lines 255-256: Is the correlation between MLD (deep convection) and AMOC (assuming you are referring to vertical overturning strength) in models in Martin-Martinez et al.? This relationship is much less clear in observations (see Li et al., 2021 for discussion in relation to the OSNAP observing system).

Lines 264-265: Given that EN4.2.2 is too coarse & has insufficient observational data (Argo etc.) to resolve subpolar boundary currents, can you really assess if the convection region along the Irminger Sea western boundary current is better represented in the HR ensemble?

Figure 9: Is the AMOC vertical overturning stream function in the HR ensemble statistically significantly different from the LR ensemble, given the wide range of AMOC mean states shown in Figure 9 (LR ensemble panels).

Lines 293-294: When comparing to model results to the RAPID-MOCHA array is the 2004-2022 period used or the 2004-2014 period overlapping the end of the historical simulations?

Lines 305-306: Suggest being more specific on the differences in the methodological approach; was the RAPID overturning stream function calculated using the METRIC package or are you comparing the model 'truth' to the RAPID calculation applied to observations?

Lines 328-344: When discussing the Gulf Stream path, the predominant focus is on the location of separation with only limited commentary on the current's structure. Suggest

undertaking a more detailed evaluation of the Gulf Stream structure, including exploring its eastward penetration using surface eddy kinetic energy following Xu and Chassignet (2017). Alternatively, an observational product such as COPERNICUS-GLOBCURRENT could be used to validate the model surface current velocities.

Lines 353-356: This is an interesting point, suggest exploring the relationship between interior dense water formation and SPG dynamics further as a potential explanation for ensemble spread / differences.

**Discussion and Conclusions**

Lines 388-391: The current discussion of Labrador Sea biases should be revised to cite more recent perspectives (e.g., review by Jackson et al., 2023; Li et al., 2023; Rühs et al. 2021).

Lines 391-394: This is an interesting hypothesis to link the NAC and LS salinity biases. Suggest reading Kostov et al. (2023, 2024) on the connection between the NAC and LS convection to extend these ideas further in the manuscript as suggested in General Comments.

Lines 408-412: Given the limitations you have identified with the EN4.2.2 product, why not use a 'purpose-built' global mixed layer climatology, such as the LOPS-IFREMER MLD product (https://doi.org/10.17882/98226)? Note, this is still a coarse resolution product, so using ARMOR3D may be more appropriate to compare to HR models.

Lines 467-470: Suggest revising this summary to focus on why the HR ensemble shows improvements compared to ocean observations, and highlight next steps forward in coupled climate modelling; for example, what are the implications of this improved representation of the ocean mean state in mesoscale-resolving ocean models on the atmosphere and societally relevant indicators? This invokes a wider question of whether the improvements in the North Atlantic mean state are sufficient to justify the additional computational cost, and to what extent the mean state determines the ocean's future trajectory in coupled models.

---

## Author Comment (AC1)

Thank you for the manuscript "The North Atlantic mean state in eddy-resolving coupled models: a multimodel study". Understanding how increasing resolution improves the mean state is an important topic and I welcome this study.

We would like to sincerely acknowledge the reviewer for their constructive comments that have greatly contributed to improve this manuscript. Please, find below the answers to all the points raised in the comments (in blue). Some of the changes in the manuscript have already been included in green.

However, I have some major comments so have to recommend major revisions. There are a lot of statements throughout about quantities being larger/smaller in the HR ensemble compared to the LR ensemble, however this isn't really tested. It seems to be based on differences in the ensemble mean without taking into account the fact that the two ensembles are quite different in size and possibly quality. If 4 members were picked at random from the LR ensemble, what is the probability that they look like the HR ensemble? It should also be borne in mind that the CMIP6 ensemble includes some models which are a long way from the observations. The authors should statistically test their assertions – one way of doing this is by using single number metrics (for instance the AMOC strength, SST in the CNA etc) and testing whether the ensembles are statistically different by: randomly picking 4 members from the LR ensemble, calculating the ensemble mean, then repeating until you have a distribution from the LR ensemble. This will show whether the HR ensemble is really different from the LR ensemble, or whether there are some LR members which have similar properties.

To address this point, we have applied "bootstrapping" to different single number metrics associated with the variables analyzed in the manuscript: SST and SSS biases in the LS, CNA, and NCH regions; max. AMOC, as well as AMOC RMSE and AMOC correlation to RAPID at 26.5ºN; RMSE and correlation of temperature, salinity and density profiles in the LIS box compared to EN4; max. MLD in the LIS box; and max. strength of the SPG (first column in Table T1).

We have allowed repetition (replacement) in the bootstrapping samples from both the LR and HR ensembles, to better describe the variability of the ensembles. Significance has been assessed by calculating the 95% confidence intervals (CIs) of the distribution of the differences in means between the two ensembles: mean(HR) - mean(LR).

First, bootstrapping has been applied using maximum ensemble sizes in both the LR and the HR ensembles (second column in Table T1). Here is the pseudocode:

*size_LR* = total size of the LR ensemble

*size_HR* = total size of the HR ensemble (it is usually 4)

*distribution* = {}

for i from 0 to 9999:

    *sample_LR* = random sample from the LR ensemble, of size size_LR, taken with replacement

*sample_HR* = random sample from the HR ensemble, of size size_HR, taken with replacement

*difference* = mean(*sample_HR*) - mean(*sample_LR*)

add *difference* to *distribution*

*confidence_interval* = [2.5th percentile of *distribution*, 97.5th percentile of *distribution*]

Subsequently, another analysis has been performed after reducing the size of the LR ensemble samples to the size of the HR ensemble, i.e. by assigning *size_LR* = total size of the HR ensemble (third column in Table T1).

If the CI obtained in this second analysis did not contain 0, we repeated the analysis by gradually increasing the size of the LR ensemble samples until a CI falling entirely to the right (or to the left) of zero was obtained (last column in Table T1).

For the LS SST and SSS biases, when bootstrapping is applied employing LR samples with the total LR ensemble size, the difference in means between the two ensembles is significant (i.e. the CI obtained does not include 0; Table T1). By contrast, when the size of the LR ensemble samples is reduced to the size of the HR ensemble (i.e., to 4), the CI does include 0. Sizes of 19 and 25 for the LR ensemble samples are required for SST and SSS, respectively, for the difference in means to become significant. This happens because there are several models in the LR ensemble with LS SST and SSS biases of comparable magnitude to those of the HR ensemble (see Fig. A2 in the original manuscript). We would like to note, though, that results are significant if the whole LR ensemble size is considered, and that even in the case of a reduced LR sample size, the corresponding CI is clearly centered to the right of 0 (Table T1).

The reduction in the CNA SST and SSS biases observed in the HR ensemble mean is not significant, as the CI of the difference in means between the HR and LR ensembles does contain 0 (for all LR ensemble subsample sizes)(Table T1). We note though that CIs are notably centered to the right of 0. In the case of SSTs, the lack of significance is associated with the cold biases in MPI-ESM1-2-ER and EC-Eart3P-VHR still present in that area (Fig. A2). For SSS, the lack of significance is related to the fact that several LR models have a similar performance to the HR models in that region. Also, we note that the MPI-ESM1-2-ER model still presents a significant SSS bias in the CNA (Fig. A2).

As for the NCH region, the analysis shows that both SST and SSS biases are significantly reduced in the HR ensemble compared to the LR ensemble (Table T1). However, in the case of SSTs, samples of at least size 8 are required from the LR ensemble to achieve this significance, which is due to the warm bias that CESM1-CAM5-SE-HR presents in that area (Fig. A1).

Regarding max. AMOC at 26.5 ºN, although CIs are notably centered to the left of 0, the reduction in strength in the HR ensemble is not significant (Table T1), since several LR models show values within the range of the HR ensemble (Fig. 11a). Interestingly, the distance to the RAPID profile (as measured by the RMSE) is significantly reduced in the HR ensemble (for all LR ensemble subsample sizes; Table T1). The increase in correlation to the RAPID curve in the HR ensemble becomes significant when samples considered in the bootstrapping from the LR ensemble have a

minimum size of 14, due to several LR models presenting correlation values in the same range of the HR ensemble (Fig. 11b).

| Metric | total LR ensemble size | reduced LR ensemble size | min. LR size for 95% sign. |
|---|---|---|---|
| LS SST (ºC) | **[0.21 1.86]** | [-0.79 3.11] | 19 |
| LS SSS | **[0.04 1.00]** | [-0.39 1.78] | 25 |
| CNA SST (ºC) | [-1.05 3.07] | [-1.43 3.66] | - |
| CNA SSS | [-0.18 1.01 ] | [-0.63 1.70] | - |
| NCH SST (ºC) | **[-4.58 -0.16]** | [-5.19 0.23] | 8 |
| NCH SSS | **[-2.54 -1.02]** | **[-2.99 -0.38]** | 4 |
| max AMOC (Sv) | [-5.10 0.66] | [-7.75 2.61] | - |
| RMSE AMOC (Sv) | **[-1.59 -0.60]** | **[-2.69 -0.05]** | 4 |
| correl AMOC | **[0.01 0.11]** | [-0.02 0.20] | 14 |
| RMSE temp profile (ºC) | **[-1.31 -0.39]** | **[-1.57 -0.12]** | 4 |
| correl temp profile | [-0.15 0.16 ] | [-0.16 0.20] | - |
| RMSE salt profile | **[-0.41 -0.09]** | [-0.68 0.04] | 8 |
| correl salt profile | **[0.01 0.04]** | [-0.00 0.07] | 6 |
| RMSE density profile (kg m-3 ) | **[-0.26 -0.06]** | [-0.48 0.03 ] | 8 |
| correl density profile | **[0.01 0.04 ]** | [-0.00 0.06] | 7 |
| max MLD (m) | [-310.52 1530.64] | [-703.04 1858.65] | - |
| max SPG (Sv) | [-11.17 9.94] | [-17.10 14.09] | - |

**Table T1**: The first column indicates the single numeric metrics analyzed (units in parenthesis). The second column shows the 95% CI of the differences in means between the HR and LR ensembles, calculated from a distribution of bootstrapping samples with repetition. The size of the samples coincides with the total size of their respective ensembles. The third column is analogous to the second one but in this case the size of the LR samples coincides with the total size of the HR ensemble. The fourth column indicates the minimum size of the LR samples in the bootstrapping required to obtain a CI not containing the value 0. Text in bold indicates when this is the case.

In terms of temperature profiles in the LIS box, RMSEs relative to EN4 are significantly reduced in the HR ensemble compared to LR, even when considering small subsamples of size four in the bootstrapping analysis. The reduction in RMSEs is particularly pronounced for the EC-Earth3P-VHR and MPI-ESM1-2-ER models (Fig. 6a). The increase in correlation with respect to the EN4 temperature profile in the HR ensemble is not significant, which is due to the low correlation exhibited by HadGEM3-GC31-HH (Fig. 6a). By removing this model from the bootstrapping calculations, correlation becomes significant even with a reduced LR subsample size (not shown). Regarding the salinity and density profiles, improvements in the HR ensemble related to both RMSE and correlation to EN4 become significant already with relatively small LR sample sizes (Table T1).

The increase in max. MLD observed in the HR ensemble compared to LR is not significant (Table T1), since several LR models present max. MLD within the same range displayed in the HR ensemble (Fig. 7). We note though that CIs are again centered well to the right of 0.

The increase in the SPG strength in the HR ensemble is also not significant (Table T1), again because several LR models present values within the same range as the HR ensemble (Fig. 12).

The manuscript will be edited to reflect all the findings described in this point.

My other concern is that the analysis here is quite basic and doesn't really show much that is new. The authors could include more analysis of scatterplots of the metrics they analyse against each other and discuss the implications for how biases affect each other. There have been a number of studies looking at how resolution affects the North Atlantic. The novelty of this paper seems to be having multiple models at eddy-resolving rather than permitting resolution. What are the implications of going to eddy-resolving resolution?

We have added scatterplots relating some of the metrics in Table T1, as well as a discussion of the associated findings in the updated version of this manuscript. More specifically, the new analyses include scatterplots of modelled Labrador Sea SSS biases versus AMOC strength, mixed layer depth, and SPG strength, as well as scatterplots of mixed layer depth vs AMOC strength, and vs SPG strength (please, see Figure F1 below as an example).
Some of the implications of going to eddy-resolving resolution have already been discussed in the previous point about bootstrapping.

[Figure]

**Figure F1.** Scatterplot of max. SPG strength (in Sv) vs max. MLD (in m), both referred to the LIS box (shown in Fig. 8). Pearson correlation coefficient and p-value are shown next to the fit line. Horizontal dashed and dot-dashed lines show EN4 and ARMOR3D observation-based values, respectively. Vertical dashed and dot-dashed lines show ORAS5m and GLORYS12 reanalysis values, respectively.

**Minor**

-L11 'important role in featuring global ocean dynamics' – what does this mean?

This sentence has been rephrased for clarification and the impact on modelled climate of resolving ocean mesoscale structures is discussed in the introduction (old lines 56-70).

New text: "Ocean mesoscale processes, which are parameterized in models with standard resolutions on the order of 1º or coarser, have an impact at larger scales, affecting the ocean mean state and circulation."

Old text: "Ocean mesoscale structures, which are parameterized in standard resolution models, play an important role in featuring global ocean dynamics."

-L19 'weaker than for lower resolution models' This is rather misleading – the way this is reported in the abstract suggests that it is a result of the resolution change. As the authors discuss, studies have shown that the impact of increased resolution varies from model to model. The result here that the AMOC is weaker in the LR ensemble is likely because of some very strong models in the LR ensemble.

We have removed this statement.

New text: "the Atlantic Meridional Overturning Circulation (AMOC) is closer to RAPID observations."

Old text: "the Atlantic Meridional Overturning Circulation (AMOC) is weaker than for lower resolution models and closer to RAPID observations"

-L63 'to a'-> 'with a'

Changed, also at L64.

-L105 Include resolution in km to compare with other values.

Done.

New text: "of at least 1/10º (10 km)."

-L173 I can't see the paper 'in review' though other studies (e.g. Marzocchi et al 2015) have suggested that this bias is because of the location of the NAC, rather than the strength.

We agree with the reviewer that also the location of the NAC plays a role in the magnitude of the CNA bias. We have edited the text accordingly. We would like to point out that Fig. 8 in Marzocchi et al., 2015 shows larger velocities in the North Atlantic at increased ocean resolution, including in the CNA region, and states that "increasing the model's resolution leads to an overall increase of absolute velocities, not only for western boundary currents but also for the entire domain...". Lin et al. is still under review, and would be removed from the text if Frigola et al. gets accepted and Lin at al. has not been accepted yet.

New text: "The other is a cold bias in the CNA (2–5º C), which earlier studies have linked to an unrealistic position of an overly weak NAC (Marzocchi et al., 2015) and an underestimation of the horizontal heat transport into the CNA domain (Lin et al., in review)."

Old text: "The other is a cold bias in the CNA (2–5º C), which earlier studies have linked to an overly weak NAC and an underestimation of the horizontal heat transport into the CNA domain (Lin et al., in review)."

-L237 and Fig 6 Using correlation of profiles to assess the shape seems rather flawed – if one profile has twice the slope of a second profile then the correlation would be perfect. It would be better to assess the stratification itself.

We agree that the use of correlation coefficients as a stand-alone metric to assess the resemblance between two different vertical profiles should be avoided, for the reason stated by the reviewer, i.e., one curve could have point-to-point slopes that are a (large) multiple of the other curve's slopes. Nevertheless, when a high correlation between two curves is observed in conjunction with a small RMSE, this is an indication that we are not in that case, i.e., that the slopes of the two curves are similar and that the distance between curves is small. That's the reason we considered both metrics together.

We have edited our text to ensure that correlation coefficients are not used as a stand-alone metric in our statements (see below). Additionally, we have added this text at the end of line 240:

"We note that the use of vertical correlation coefficients to assess resemblance between two vertical profiles should come in conjunction with other metrics, such as RMSEs, or direct visual inspection of profiles, as a high correlation coefficient alone does not ensure a small distance between curves."

Lines 236-238 have been rephrased as:

New text: "Indeed, the density profile for HR-HIST is closer in shape to the EN4-derived one, as supported by the Pearson correlation coefficients in Fig. 6c, which are very close to one in all HR-HIST models, and by the relatively small RMSEs."

Old text: "Indeed, the density profile for HR-HIST is closer in shape to the EN4-derived one, as supported by Pearson correlation coefficients in Fig. 6c, which are very close to one in all HR-HIST models.

Additionally, we have slightly edited Lines 227-229:

New text: "Overall, the vertical salinity profile exhibits a more realistic shape in HR-HIST, a higher correlation coefficient, and a slightly smaller RMSE against EN4 (Figs. 5b, 6b)."

Old text: "Overall, the vertical salinity profile is more realistic in HR-HIST, as supported by the higher correlation coefficient and smaller RMSE against EN4 indicated in Fig. 6b."

-L243 'expected to impact on'

Changed.

-Fig 6 What region are these for?

The Pearson correlation coefficients and RMSEs are associated to the vertical profiles in Fig. 5, which are averaged over the region (35º–60º W, 50º–65º N), shown in Figs. 7 and 8 (blue box). The captions in Fig. 6 have been edited for clarification.

New text:

Figure 6. Pearson correlation coefficient (horizontal axis) and Root Mean Square Error (RMSE) (vertical axis; units as in Fig. 5) for the vertical (a) temperature, (b) salinity and (c) density profiles in Fig. 5 against EN4. Profiles are averaged over the region (35º–60º W, 50º–65º N), the LIS box shown in Figs. 7 and 8.

Old text:

Figure 6. Pearson correlation coefficient (horizontal axis; units as in Fig. 5) and Root Mean Square Error (RMSE) (vertical axis) of the (a) temperature, (b) salinity and (c) density profiles in Fig. 5 against EN4, in the vertical dimension.

-L256 'Its' -> 'The'

Changed.

-Fig 8 Might be clearer if you adjusted the scale

Done.

-L287 I don't really understand this sentence.

The better agreement of the AMOC between the HR-HIST ensemble mean and ORAS5m (compared to the AMOC between the LR-HIST ensemble mean and ORAS5m) might not only be due to a more realistic representation of the AMOC in HR-HIST. The fact that ORAS5m AMOC data were produced with an eddy-permitting version of the ocean model NEMO, which in this

sense is closer to the models in the HR-HIST ensemble (eddy-resolving), than to the ones in the LR-HIST ensemble (eddy-parameterized), might also contribute to the resemblance of the HR-HIST ensemble mean and ORAS5m AMOC representations, as part of the AMOC variability will by driven by the model physics.

The old text has been rephrased for clarification.

New text:

"The larger resemblance of the AMOC streamfunction in ORAS5m and the HR-HIST ensemble mean (compared to the LR-HIST ensemble mean) might be to some extent related to the fact that ORAS5m is run with an eddy-permitting model, and is thus potentially more similar to the models in the HR-HIST ensemble (eddy-resolving), than to the models in the LR-HIST ensemble (eddy-parameterized)."

Old text:

"This improved agreement with ORAS5m can partly be attributed to the use of an eddy-permitting ocean model in ORAS5m, allowing for the representation of some eddies."

-L305 It's not entirely clear what you mean by methodological approach and what it affects – expand a little on this.

Details on the methodological approaches have been added to the text. Please, see below.

[revised manuscript text omitted]

-L397 temperature signals are often damped by interactions with the atmosphere reducing the amplitude of the signal. This would explain why there are stronger correlations for salinity.

The sentence has been rephrased to include the reviewer's suggestion.

New text:

"The correlation between the SST biases of the LS and CNA regions is also significant, although weaker compared to the SSS biases ($r = 0.54$, $p < 0.001$; Fig. A2), which could be related to a damping of the SST signal through interactions with the atmosphere (which are usually more important than for salinity), or to mixing with Arctic waters."

Old text:

"The correlation between the SST biases of the LS and CNA regions is also significant, although weaker compared to the SSS biases ($r = 0.54$, $p < 0.001$; Fig. A2), which might indicate additional differences between the mechanisms exerting control over the SSTs in both regions, like the local atmospheric forcing."

-L401-404 I don't understand what the authors are getting at here – please explain more.

The text has been rephrased and extended.

New text:

"We find improved temperature and salinity stratification in the LIS box for HR-HIST compared to LR-HIST, with the HR-HIST ensemble mean curve closer in distance and shape to EN4. The warm and salty biases present at the subsurface in both ensembles over most of the column (below ~150m) are reduced in the HR-HIST ensemble mean. On the other hand, surface biases (above ~150 m) are more pronounced in the HR-HIST ensemble mean compared to the LR-HIST ensemble mean. The increased (reduced) biases at the surface (subsurface) for HR-HIST might be related to the fact that ocean mesoscale eddies increase vertical (upwards) heat and salt transports

in the ocean (Hewitt et al., 2017). Although vertical transport by eddies provides a plausible explanation for the warm and salty biases in the LS, additional factors might be needed to correct for those biases."

Old text:

"Interestingly, we find improved vertical profiles of temperature and salinity in the LIS box for HR-HIST: despite the larger biases found at the surface with respect to LR-HIST, the subsurface is colder and fresher in HR-HIST. This might be related to increased vertical (upwards) heat and salt transports by ocean mesoscale eddies (Hewitt et al., 2017), providing a potential explanation for the warm and salty surface biases in the LS."

-L410-418 Different definitions of MLD can give very different results. Do all the studies mentioned use the same density criteria? Also you don't make it very clear that using monthly mean densities to calculate a MLD can give very different (and likely shallower) estimates than means of instantaneous profiles used by Yashayaev etc.

Yashayaev and Loder (2016; 2017), Holte et al. (2017), and our study, all use a different methodology (a description of the different methods is provided below in the new text). We believe that the exact effect of combined temporal and spatial data smoothing in the different studies might be difficult to assess. Nevertheless, we have highlighted the different temporal and spatial characteristics of the data in the studies as one of the causes for the differences in the range of MLD estimates.

New text:

"A range of methods is used in the different studies. The yearly estimates by Yashayaev and Loder (2016; 2017) are winter maximum values of "aggregate" maximum convection depths, defined as the 75th percentile of the depth of the base of the pycnostad in the set of available individual LS profiles at each time. To estimate the depth of the pycnostad for each profile, first, layer thicknesses of $\sigma_1$ potential density classes (binned in 0.005 kg m$^{-3}$ intervals) are calculated (Fig. 3b in Yashayaev and Loder, 2016); subsequently, the lower boundary of each pycnostad is defined as the depth corresponding to the $\sigma_1$ value with the largest layer thickness (plus a constant). Holte et al. (2017)'s MLD estimates (shown in their Fig. 3a) correspond to individual Argo profiles and are obtained with a density algorithm (Holte and Talley, 2009) that uses temperature, salinity, and density data from individual profiles to calculate MLD through a combination of methods and elements, including temperature and density threshold methods (with threshold values of 0.2ºC and 0.03 kg m$^{-3}$, respectively), temperature and density gradient methods, maximum/minimum values of temperature, salinity and density over the profiles, estimates of thermocline linear fits, etc. Meanwhile, in our study, MLD values are a climatology of March MLD monthly means obtained from gridded temperature and salinity data through a density threshold method based on monthly data (threshold value = 0.03 kg m$^{-3}$). Overall, the maximum MLD values in our HR-HIST ensemble mean for the LS (1800–2000 m) are larger than the observational estimates, excluding the record values in Yashayaev and Loder (2017). These differences might arise from the different methodological approaches, from the different temporal and spatial characteristics of the profile data, as well as from differences in the time intervals analyzed."

Old text:

    "Overall, the maximum values in the HR-HIST ensemble mean for the LS (1800–2000 m) are slightly larger than the direct observational estimates, excluding the record values in Yashayaev and Loder (2017). However, it is important to note that some differences are expected as our HR-HIST values are computed from 1980 to 2014, and use generally smoother profiles associated with the coarser temporal resolution of the model data compared to the individual profiles from observational studies. "

-L463 'southward propagation of the MLD signal into the AMOC' – what does this mean?

We have expanded the original text. The results described are visible in Fig. 7 in Martin-Martinez et al. (2024).

New text:

"Recent work by Martin-Martinez et al. (2024) shows that ocean resolution also affects the timescales governing large-scale dynamical processes in the North Atlantic. In that study, Labrador Sea mixed layer depth is found to be positively correlated with overturning streamfunction strength at high latitudes at 0 time lags in all resolutions of the HighResMIP EC-Earth3P-VHR control simulations. This imprinted signal of the mixed layer in the overturning streamfunction at high latitudes propagates to lower latitudes at subsequent lags. Interestingly, the propagation speed is significantly larger in eddy-resolving models, compared to coarser resolution models (Fig. 7 in Martin-Martinez et al., 2024)"

Old text:

"Results by Martin-Martinez et al. show a faster southward propagation of the MLD signal into the AMOC in eddy-resolving models, indicating that ocean resolution affects also the timescales of the dynamics of the North Atlantic."

---

## Author Comment (AC2)

**Summary**

Frigola et al. investigate the role of resolving mesoscale ocean eddies on the representation of the mean state & circulation of the North Atlantic Ocean in coupled climate models. The authors present a largely qualitative comparison of four coupled historical simulations with nominal ocean resolutions of at least 1/10° to an ensemble of 39 coupled simulations configured at coarser horizontal resolution. The study concludes that the vertical stratification & deep convection in the subpolar North Atlantic, and both the meridional overturning and barotropic circulations agree more closely with ocean observations at mesoscale eddy-resolving resolution. The manuscript is generally well written and includes a valuable final discussion; however, I have significant concerns regarding:
(1) the use of the phrase 'eddy-resolving',
(2) the ocean observations and reanalysis products used to evaluate model performance, and
(3) the originality of the study & its wider implications.
I would not recommend the manuscript for publication until major revisions have been made to address each of the comments below.

We would like to sincerely acknowledge the reviewer for their constructive comments that have greatly contributed to improving this manuscript. The above-mentioned points have been addressed. The use of the phrase "eddy-resolving" has been refined to "mesoscale eddy-resolving" throughout the manuscript, new observations and reanalysis have been included as suggested, as well as new diagnostics exploring inter-model differences and relationships between the different variables analyzed (please, see below for details).

**General Comments**

**- Use of 'eddy-resolving':**
In both the title and throughout the manuscript, the authors use 'eddy-resolving' to refer to coupled model simulations with sufficiently fine horizontal ocean resolution to resolve mesoscale eddies. Given the recent emergence of both submesoscale permitting / resolving ocean model configurations (e.g., Chassignet & Xu, 2017; Lévy et al., 2010; Pennelly & Myers, 2020; Pennelly & Myers, 2022; Li et al., 2023) and a growing awareness that representing or resolving submesoscale processes is integral for the accurate simulation of North Atlantic mean state (see Jackson et al, 2023 for a review), I would strongly recommend that the authors refine their use of 'eddy-resolving' to 'mesoscale resolving' throughout.

Done. We have rephrased "eddy-resolving" as "mesoscale eddy-resolving" throughout the manuscript.

Similarly, I would suggest revising the manuscript title to: 'The North Atlantic Ocean mean state in mesoscale eddy-resolving coupled models: a multi-model study' or equivalent.

Done.
New title: 'The North Atlantic Ocean mean state in mesoscale eddy-resolving coupled models: a multi-model study'.

Furthermore, the authors should explicitly address the role of submesoscale features and the implications of their (justifiable) absence in current generation coupled climate models for the North Atlantic mean state; for example, their important role in restratifying the Labrador Sea (Clément et al., 2023; Frajka-Williams et al., 2014) and reducing deep convection (e.g., Tagklis et al. (2020)) and Gulf Stream penetration (e.g., Chassignet and Xu (2017) and Chassignet et al. (2020)).

Thanks for this remark. We have added a paragraph in the introduction (right before old line 71) discussing the contribution of submesoscale eddies to the North Atlantic mean state and the implications of their absence.

New text:

"Also submesoscale processes have an impact on the North Atlantic mean state. Tagklis et al. (2020) show a significant reduction in deep water convection in the Labrador Sea (and an increase in vorticity) when increasing the grid resolution in a regional model from 15 km (mesoscale permitting in the Labrador Sea) to 1 km (submesoscale resolving in the Labrador Sea). That study finds that the simulated reduction in convection is caused by eddy heat advection from the Irminger Current and by local submesoscale eddy buoyancy fluxes from the Labrador Sea basin itself. Similarly, restratification of the Labrador Sea convective areas at the end of winter has been associated with both mesoscale and submesoscale eddies (Clément et al., 2023). Another example of the importance of the submesoscale in the representation of the North Atlantic dynamics is a further eastward penetration of the NAC and its eddy variability at 1/50º resolution, in closer agreement with observations, compared to mesoscale resolving scales (Chassignet and Xu, 2017). Omitting submesoscale eddies contributions might thus imply biases in the representation of the NAC and deep water convection. Current computational resources allow for multidecadal global coupled runs at mesoscale resolving resolutions, so that future research efforts should be aimed at parameterizing submesoscale-related processes to the extent possible."

The manuscript would also benefit from greater effort to contextualise inter-model differences within the LR & HR ensembles; for example, do all HR simulations use the same z-level vertical coordinate system & what impact would any difference have on the representation of the overflows (e.g., Colombo et al., 2020, Bruciaferri et al., 2024) and Labrador Sea stratification downstream (MacGilchrist et al. 2020).

In order to help identify and explain inter-model differences, we have added and discussed new figures (which can be found in the updated manuscript) relating the different variables analyzed in our study. More specifically, the new analyses include scatterplots of modelled Labrador Sea SSS biases versus AMOC strength, mixed layer depth, and SPG strength, as well as scatterplots of mixed layer depth vs AMOC strength, and vs SPG strength (please, see Figure F6 below as an example).

Regarding the vertical coordinate system, EC-Earth3P-VHR and HadGEM3-GC31-HH share the same z-level vertical coordinates, with a total of 75 depth levels. CESM1-CAM5-SE-HR uses a different z-level vertical grid and it has 62 depth levels. MPI-ESM1-2-ER has the coarsest vertical grid of all, with 40 levels. As for the horizontal resolutions, the finest are again those of HadGEM3-

GC31-HH and EC-Earth3P-VHR (1/12º), followed by CESM1-CAM5-SE-HR and MPI-ESM1-2-ER (1/10º).

Ocean models using fixed vertical levels (z-models) present difficulties in correctly representing the densities of the Arctic overflows downslope the Greenland-Scotland Ridge (Bruciaferri et al., 2024; Colombo et al., 2020; Jackson et al., 2023), which in turn affects stratification in the subpolar North Atlantic and transport in the AMOC lower limb (Bruciaferri et al., 2024).

Results by Colombo et al. (2020) suggest that increasing horizontal resolution to submesoscale scales in combination with an increase in vertical resolution improves the representation of the Denmark Strait overflow, although at a fixed horizontal resolution of 1/12º, increasing the vertical resolution alone does not lead to any improvement (Colombo et al. 2020). If the properties of the overflows were exclusively based on the model resolution and not dependent on the model physics, we could hypothesize that the EC-Earth3P-VHR and HadGEM3-GC31-HH models might have a better representation of the overflows compared to MPI-ESM1-2-ER and CESM1-CAM5-SE-HR, due to their slightly finer combined horizontal and vertical resolutions. However, a detailed analysis of the differences in the representation of the overflows in the HR-HIST models was beyond the scope of this study.

Nonetheless, we have added a paragraph in the discussion section of the manuscript explaining the challenges presented in z-models in representing the overflows (as described above) and how the introduction of local terrain-following coordinates near the Greenland-Scotland Ridge in models could represent a source of improvement (Bruciaferri et al., 2024).

**- Datasets used for model validation:**

My second major concern is both the choice of ocean observations and reanalysis datasets used to validate the ocean model components & the details absent from their methodology descriptions.

We have addressed each of these points right below.

More specifically, I could not find justification for why a coarse resolution ocean analysis product (EN4.2.2) is used to validate mesoscale-resolving ocean models, when at least eddy-permitting resolution products are available (e.g., ARMOR3D [https://doi.org/10.48670/moi-00052] or ASTE [Nguyen et al., 2021]). While no observational product is a 'true' representation of reality, I would argue it is more appropriate to compare the property fields simulated in mesoscale resolving models with observational products that can, at least partially, represent them.

Thanks for this remark. The ARMOR3D dataset at 0.25º resolution (Guinehut et al. 2012), which we were not aware of, has now been added to our analysis of the MLD to complement the previous EN4 estimates and to provide further robustness to our results (please, see Fig. F1). We would like to note, though, that the EN4 1º data have been previously employed to validate both model MLD fields at eddy-permitting and eddy-resolving resolutions (Koenigk et al., 2021; Martin-Martinez et al., 2024), and model temperature and salinity fields in mesoscale eddy-resolving models (Jackson et al. 2023; Chassignet et al. 2020; Roberts et al. 2020; Gutjahr et al. 2019; Roberts et al. 2019; Moreno-Chamarro et al. 2025; and Martin-Martinez et al. 2024 use EN4; Marzocchi et al. 2015 use EN3). For that reason, we have kept it for better comparability with those previous

studies. Besides, EN4 data span the entire historical period covered in our analysis (i.e. 1980-2014), making it especially suitable for bias assessment in the historical period, meanwhile ARMOR3D data only covers from 1993 onwards.

[Figure]

**Figure F1.** March MLD (in m) by groups, for (a) LR-HIST, (b) HR-HIST, (c) EN4, and (d) ARMOR3D. In all cases, including ARMOR3D, MLD has been calculated from temperature and salinity fields using the density threshold method of 0.03 kg m$^{-3}$ described in the manuscript. The time interval covered is 1980-2014 in (a), (b), (c), and 1993-2014 in (d).

We would like to note a characteristic feature of the ARMOR3D dataset, namely a distinct stripe of deep mixing attached to the shelf along the East Greenland Current (Fig. F1).

Similarly, the authors could have used the mesoscale-resolving GLORYS12 ocean reanalysis product (https://doi.org/10.48670/moi-00021 - on its original NEMO grid) rather than the eddy-permitting ORAS5m reanalysis product to validate the mean meridional overturning stream function in depth space.

The GLORYS12 overturning and barotropic streamfunctions have been added to our analysis for further robustness of the results, to complement the ORAS5m dataset (0.25º resolution)(please, see Figs. F2 and F3). We have calculated both streamfunctions using GLORYS12 velocity fields (https://doi.org/10.48670/moi-00021), which are provided on a regular grid at 1/12° (0.083°).

[Figure]

**Figure F2.** AMOC streamfunction (in Sv) by groups, for (a) LR-HIST, (b) HR-HIST, (c) ORAS5m, and (d) GLORYS12. The time interval covered is 1980-2014 in (a), (b), (c), and 1993-2014 in (d).

[Figure]

**Figure F3.** BSF (in Sv) by groups for (a) LR-HIST, (b) HR-HIST, (c) ORAS5m and (d) GLORYS12. The time interval covered is 1980-2014 in (a), (b), (c), and 1993-2014 in (d).

There are also some important details missing from Section 2.3 of the methodology; for example, are model property fields regridded onto the observations or vice versa for validation? And what type of interpolation is used: bilinear, conservative etc? The current use of interpolating colour contour plots does not make this obvious to readers.

For direct visual comparison of mean-state properties across models (e.g. in Figs. 7, 9, 12) data are plotted on the original grid. When some computation between models or against observations is required (e.g., multi-model means, model biases), model data are mostly regridded onto the observation's grid or, in some cases, onto a common regular grid. Methods employed are linear/bilinear (according to the dimension of the data), and nearest neighbour interpolation (closest source point), depending on the case. In all cases, a comparison between the regridded and the original data was performed, to ensure suitability of the interpolation scheme.
More specifically, in Figs. 1-4 model data are regridded to the EN4 regular grid using nearest neighbour interpolation (closest source point). In Fig. 5 individual model profiles are plotted in the original grid, meanwhile ensemble means are computed after linear regridding to the EN4 profile. All bias metrics in Fig. 6 use linear regridding to EN4. In Fig. 8 all data have been regridded to a 1º regular grid using nearest neighbour interpolation (closest source point). In Fig. 10 data are regridded to the ORAS5m grid, using bilinear interpolation. Figure 11 is analogous to Figs. 5 and 6, but using ORAS5m instead of EN4. In Fig. 13 data have been regridded to a 1/4º regular grid using bilinear interpolation.

**- Original contribution to our understanding:**

My final concern is regarding the manuscript's original contribution to our understanding of the representation of the North Atlantic Ocean in coupled climate model simulations. The authors do a good job of placing their largely qualitative findings into wider context in Section 4, however, I still remain unsure which of the study's findings are original since the impact of model resolution on sea surface biases is addressed in Roberts et al., 2019, Gutjahr et al., 2019 & Marzocchi et al., 2015, and the strength and structure of the AMOC in Talandier et al., 2014, Roberts et al., 2020, Hirschi et al., 2019, Jackson et al., 2019, Jackson et al. 2023 (see references within) and Reintges et al., 2024.

We have added new diagnostics to the manuscript (please, see next point). Nevertheless, although we acknowledge the highly valuable studies cited here, we would like to note that, compared to most previous studies, our study employs a larger, state-of-the-art mesoscale eddy-resolving model ensemble (4 models) and a larger low-resolution ensemble of CMIP6 models (39 models). This allows for a more robust characterization of the role of resolution in the representation of the North Atlantic. For example, Marzocchi et al. (2015) focus on one single, non-coupled model (NEMO). Similarly, the studies by Gutjahr et al. (2019) and Roberts et al. (2019) are based on single coupled models: MPI-ESM1.2 and HadGEM3-GC3.1, respectively. Talandier et al. (2014) employ a non-coupled ocean model (OPA), at 0.5º resolution with a nest at 0.125º resolution in the North Atlantic. Roberts et al. (2020) employ 7 coupled models but only 2 of them include mesoscale eddy-resolving configurations. Hirschi et al. (2020) use a total of 10 models, out of which, 4 are coupled, and the study focuses exclusively on AMOC analyses. The study by Reintges et al. (2024)

employs a total of ~47 different model configurations, although most are at eddy-parameterized scales: 3 of them are mesoscale eddy-permitting, but none is eddy-resolving, and that study does not investigate the effect of resolution on the representation of the AMOC. Similarly, Jackson et al. (2020)*'s study includes 5 models at eddy-parameterized and mesoscale eddy-permitting resolutions, but none at eddy-resolving resolutions.

Furthermore, the fact that we are using different models in the ensembles, compared to previous studies, constitutes an additional source of value. For example, the AMOC results obtained in our study are different from those obtained in Roberts et al. (2020) and Hirschi et al. (2020)(see lines 438-452 in the discussion of our manuscript).

Finally, we would like to note that, apart from the assessment of the surface biases and the AMOC, we have also conducted analysis on the Labrador Sea profiles and winter mixed layers in the North Atlantic. Previous work by Heuzé (2021) has investigated the mixed layer representation in a multimodel context, although that study has a different aim: the role of resolution is not assessed, and no mesoscale eddy-resolving models are considered. Koenigk et al. (2021) have successfully investigated the role of resolution on the depth of the mixed layer in a multimodel context, but they use a different definition of mixed layer and they employ 7 models, out of which one includes a mesoscale eddy-resolving configuration.

To summarize, we believe that the larger ensembles in our study, as well as the different models, methods, and combined analysis of different related quantities constitute a meaningful contribution to our common understanding of the effect of resolving the mesoscale in the North Atlantic that additionally avoids drawing model-dependent conclusions.

*We have searched for Jackson et al. (2019) but that manuscript focuses on ocean reanalysis, so we have assumed that the correct reference was Jackson et al. (2020). Please, correct us if otherwise.

To progress beyond identifying differences between ensembles and ocean observations, the study should place greater emphasis on the reasons why these differences exist, including those differences between ensemble members; for example, why is HadGEM3-GC3.1-HH often an outlier in the HR ensemble? The authors begin to address this in Section 4 by identifying an interesting 'potential link between LS salinity biases and the NAC, through the effect of the NAC on the northward salinity transport' and I would strongly encourage them to pursue this further since developing diagnostics to better understand common model biases would be a valuable contribution of this research.

We have added and discussed scatterplots (which can be found in the updated manuscript) exploring relationships between the different variables in the study, putting a special focus on salinities. More specifically, the new analyses include scatterplots of modelled Labrador Sea SSS biases versus AMOC strength, mixed layer depth, and SPG strength, as well as scatterplots of mixed layer depth vs AMOC strength, and vs SPG strength (please, see Figure F6 below as an example). Besides, these new plots have been analyzed to identify and explain inter-model differences.

**Specific Comments**

**Abstract**
Lines 10: Suggest clarifying what 'standard resolution models' are? This description could be clearer for readers.

We have rephrased "which are parameterized in standard resolution models" as "which are parameterized in models with standard resolutions on the order of 1° or coarser".

**Introduction**

Line 39-45: Suggest revising this paragraph from one long sentence to demonstrate the interconnectivity between water mass processes. As it stands, deep convection, surface forced water mass transformation and densification along the SPG boundary are highlighted separately, yet both deep convection and boundary current densification are a result of surface forced water mass transformation. It may be beneficial to frame this discussion in terms of surface forced water mass transformation and mixing and their importance for deep convection and dense water formation – and the role of horizontal ocean model resolution in representing these processes.

The old text has been rephrased to reflect the interconnectivity between water mass processes.

New text (replacing old lines 39-45):
"The North Atlantic circulation is influenced by a series of elements and processes that are strongly interconnected. The strength and path of the NAC affect the heat and salinity content of the waters reaching the subpolar North Atlantic (SPNA; Marzocchi et al., 2015). There, the relatively warm and saline AMOC upper limb waters undergo a process of densification associated to 1) surface water mass transformation through air-sea buoyancy fluxes (Petit et al., 2020; Jackson and Petit, 2023) and 2) mixing with denser (colder) waters from the Greenland–Scotland Ridge overflows (Dey et al., 2024), together triggering deep water convection in the SPNA basins (Koenigk et al., 2021). Sinking of deep waters forming the AMOC return flow occurs at the boundaries of the SPG and it has been associated with densification of waters along the boundary current (Katsman et al., 2018; Straneo, 2006; Spall and Pickart, 2001).

The role of horizontal ocean model resolution in representing these processes is addressed in the next paragraph (from old line 46 onwards). In particular, we have added the following text to the old version:

"In the SPNA, ocean eddies contribute to the downwelling along the boundary current of the SPG through advection of density and vorticity from the interior basins (Straneo 2006; Brüggemann et al. 2017). They also contribute to restratification of Labrador Sea convective areas at the end of winter (Clément et al. 2023)."

Lines 46-55: Suggest including a brief discussion of submesoscale eddies in this paragraph and using mesoscale-resolving models to be more precise (see general comments above).

This point has been addressed in the "General Comments" section (please, see above for further details).

Lines 59-61: The Introduction appears to depend heavily on the single model study of Marzocchi et al. (2015), however, the role of ocean model resolution on the Gulf Stream position is also explored in the more recent studies of Chassignet and Xu (2017) and Chassignet et al. (2020). Suggest extending the references cited here.

We have added the reference Chassignet et al. (2020) at line 61. Regarding the discussion on the work by Chassignet and Xu (2017), we think that it fits better in the paragraph about the ocean submesoscale (see previous point and "General Comments").

Lines 78-79: Suggest rephrasing this sentence to more accurately reflect the number of multi-model comparisons that have been performed; for example, Jackson et al. (2022), Jackson et al. (2023), Reintges et al. (2024) all consider coupled climate models in a North Atlantic context.

Done. Regarding the study by Jackson et al. (2022), we have preferred not to include it in this particular list of citations, because it mostly considers forced models (only one of their figures considers coupled models). Similarly, Jackson et al. (2023) mostly focuses on the HadGEM3 model (only one figure is multimodel). The latter study will though be cited below in the first point of the "Discussion and Conclusions".

The sentence: "To our knowledge, only a few multimodel comparisons of coupled historical experiments with a focus on the North Atlantic ocean exist, that include eddy-resolving simulations (e.g. Roberts et al., 2020; Koenigk et al., 2021), although none of them specifically addresses the impact of resolving mesoscale ocean eddies.",

has been rephrased as:
"Although a wide range of multimodel studies considering coupled climate models in a North Atlantic context have been published (e.g. Reintges et al., 2024; Jackson and Petit, 2023; Bellomo et al., 2021; Heuzé, 2021; Roberts et al., 2020; Koenigk et al., 2021), only a few of them include eddy-resolving simulations (e.g. Koenigk et al., 2021; Roberts et al., 2020) and none of them specifically addresses the impact of resolving mesoscale ocean eddies."

**Methods**

Lines 112-119: Are the three-dimensional temperature and salinity fields used in the study stored on the original model grid or the regularly interpolated tracer fields?

This question has been addressed in the "General Comments" section (regridding details).

Here and throughout, suggest being more precise in the use of AMOC. The AMOC is a phenomenon and the overturning stream function in depth-space is a diagnostic used to understand one aspect of this phenomenon. Suggest using vertical overturning or overturning in depth-space throughout since the diapycnal overturning is not considered in this study (although I would argue is a more relevant diagnostic to consider due to its close relationship to sea surface property biases explored later in this study).

The term "AMOC" has been replaced with the terms "overturning in depth-space" or "AMOC streamfunction" throughout the manuscript.

Lines 141-144: The authors make a strong case for diagnosing the MLD from the time-averaged potential density anomaly field following de Boyer Montegut et al. (2004), so I was surprised that the authors did not then compare this result to the available de Boyer Montegut et al. (2004) MLD climatology.

We aimed at producing MLD estimates both from models and observations that 1) were calculated using the same methodology and that 2) were based on data with similar characteristics, i.e., gridded data. That is why we made the choice of using EN4 gridded data instead of MLD observational products based on individual profiles. Now ARMOR3D MLDs calculated with the same methodology as in models (from temperature and salinity data and a density threshold of 0.03 kg m$^{-3}$) have also been added to the MLD analysis to provide an additional comparison (please, refer to Fig. F1)
However, qualitative comparisons to observational MLD data are carried out in the discussion section of our manuscript (lines 410-418).

Lines 152-156: How is the AMOC & barotropic stream function calculated in the ORAS5m reanalysis data? Is the calculation performed on the original model grid or using interpolated model fields?

Both the AMOC and the barotropic stream function in the ORAS5m data were calculated on the original model grid. We have added this information in the text.

**Results**

Figures 2-4: Here and throughout the manuscript text, suggest discussing the statistical significance of the differences between the LR & HR ensembles. For example, there is considerable discrepancies between SST & SSS bias within the HR ensemble, especially around the NAC. Is the improvement in SST bias in the Central North Atlantic region in the HR ensemble simply due to the warm bias exhibited by HadGEM3-GC31-HH counteracting the cold biases in the other ensemble members?

To address the statistical significance of the differences in means between the LR & HR ensembles, we have applied "bootstrapping" to different single number metrics associated with the variables analyzed in the manuscript: SST and SSS biases in the LS, CNA, and NCH regions; max. AMOC,

as well as AMOC RMSE and AMOC correlation to RAPID at 26.5ºN; RMSE and correlation of temperature, salinity and density profiles in the LIS box compared to EN4; max. MLD in the LIS box; and max. strength of the SPG (first column in Table T1).

We have allowed repetition (replacement) in the bootstrapping samples from both the LR and HR ensembles, to better describe the variability of the ensembles. Significance has been assessed by calculating the 95% confidence intervals (CIs) of the distribution of the differences in means between the two ensembles: mean(HR) - mean(LR).

As a first analysis, bootstrapping has been applied using maximum ensemble sizes in both the LR and the HR samples, i.e. by taking LR ensemble samples with the total size of the LR ensemble, and HR ensemble samples with the total size of the HR ensemble (second column in Table T1). Here is the pseudocode:

*size_LR* = total size of the LR ensemble

*size_HR* = total size of the HR ensemble (it is usually 4)

*distribution* = {}

for i from 0 to 9999:

    *sample_LR* = random sample from the LR ensemble, of size size_LR, taken with replacement

    *sample_HR* = random sample from the HR ensemble, of size size_HR, taken with replacement

    *difference* = mean(*sample_HR*) - mean(*sample_LR*)

    add *difference* to *distribution*

*confidence_interval* = [2.5th percentile of *distribution*, 97.5th percentile of *distribution*]

Subsequently, a second analysis has been performed after reducing the size of the LR ensemble samples to the size of the HR ensemble, i.e. by assigning *size_LR* = total size of the HR ensemble (third column in Table T1). This second analysis is aimed at investigating whether the differences in means between the LR & HR ensembles are still significant when the LR ensemble is considered as subsamples of the same size as the HR ensemble (which is considerably smaller than the LR ensemble), i.e. whether by randomly picking subsamples from the LR ensemble of the same size as the HR ensemble (usually 4), the differences in means between LR & HR are still significant.

If the CI obtained in this second analysis did not contain 0, we repeated the analysis by gradually increasing the size of the LR ensemble samples until a CI falling entirely to the right (or to the left) of zero was obtained. This allowed us to determine the minimum size of the LR ensemble required for significance (last column in Table T1).

For the LS SST and SSS biases, when bootstrapping is applied employing LR samples with the total LR ensemble size, the difference in means between the two ensembles is significant (i.e. the CI obtained does not include 0; Table T1). By contrast, when the size of the LR ensemble samples

is reduced to the size of the HR ensemble (i.e., to 4), the CI does include 0. Sizes of 19 and 25 for the LR ensemble samples are required for SST and SSS, respectively, for the difference in means to become significant. This happens because there are several models in the LR ensemble with LS SST and SSS biases of comparable magnitude to those of the HR ensemble (see Fig. A2 in the original manuscript). We would like to note, though, that results are significant if the whole LR ensemble size is considered, and that even in the case of a reduced LR sample size, the corresponding CI is clearly centered to the right of 0 (Table T1).

As pointed out by the reviewer, the reduction in the CNA SST and SSS biases observed in the HR ensemble mean is not significant, as the CI of the difference in means between the HR and LR ensembles does contain 0 (for all LR ensemble subsample sizes)(Table T1). We note though that CIs are notably centered to the right of 0. In the case of SSTs, the lack of significance is associated with the cold biases in MPI-ESM1-2-ER and EC-Eart3P-VHR still present in that area (Fig. A2). In order to test whether the reduction in the CNA SST bias in HR-HIST is related to the warm bias shown by HadGEM3-GC31-HH in that region, we have removed the HadGEM3-GC31-HH model from the HR ensemble. In that case, the SST HR mean in the CNA is still larger than the LR mean, but the difference is very small (LR-HIST SST mean = -2.83 ºC vs HR-HIST mean = -2.47 ºC) and again not significant at the 95% confidence level. For SSS, the lack of significance is related to the fact that several LR models have a similar performance to the HR models in that region. Also, we note that the MPI-ESM1-2-ER model still presents a significant SSS bias in the CNA (Fig. A2).

As for the NCH region, the analysis shows that both SST and SSS biases are significantly reduced in the HR ensemble compared to the LR ensemble (Table T1). However, in the case of SSTs, samples of at least size 8 are required from the LR ensemble to achieve this significance, which is due to the warm bias that CESM1-CAM5-SE-HR presents in that area (Fig. A1).

Regarding max. AMOC at 26.5 ºN, although CIs are notably centered to the left of 0, the reduction in strength in the HR ensemble is not significant (Table T1), since several LR models show values within the range of the HR ensemble (Fig. 11a). Interestingly, the distance to the RAPID profile (as measured by the RMSE) is significantly reduced in the HR ensemble (for all LR ensemble subsample sizes; Table T1). The increase in correlation to the RAPID curve in the HR ensemble becomes significant when samples considered in the bootstrapping from the LR ensemble have a minimum size of 14, due to several LR models presenting correlation values in the same range of the HR ensemble (Fig. 11b).

In terms of temperature profiles in the LIS box, RMSEs relative to EN4 are significantly reduced in the HR ensemble compared to LR, even when considering small subsamples of size four in the bootstrapping analysis. The reduction in RMSEs is particularly pronounced for the EC-Earth3P-VHR and MPI-ESM1-2-ER models (Fig. 6a). The increase in correlation with respect to the EN4 temperature profile in the HR ensemble is not significant, which is due to the low correlation exhibited by HadGEM3-GC31-HH (Fig. 6a). By removing this model from the bootstrapping calculations, correlation becomes significant even with a reduced LR subsample size (not shown). Regarding the salinity and density profiles, improvements in the HR ensemble related to both RMSE and correlation to EN4 become significant already with relatively small LR sample sizes (Table T1).

The increase in max. MLD observed in the HR ensemble compared to LR is not significant (Table T1), since several LR models present max. MLD within the same range displayed in the HR ensemble (Fig. 7). We note though that CIs are again centered well to the right of 0.

The increase in SPG strength in the HR ensemble is also not significant (Table T1), again because several LR models present values within the same range as the HR ensemble (Fig. 12).

The manuscript will be edited to reflect all the findings described in this point.

| Metric | total LR ensemble size | reduced LR ensemble size | min. LR size for 95% sign. |
|---|---|---|---|
| LS SST (ºC) | **[0.21 1.86]** | [-0.79  3.11] | 19 |
| LS SSS | **[0.04 1.00]** | [-0.39  1.78] | 25 |
| CNA SST (ºC) | [-1.05  3.07] | [-1.43  3.66] | - |
| CNA SSS | [-0.18  1.01 ] | [-0.63   1.70] | - |
| NCH SST (ºC) | **[-4.58 -0.16]** | [-5.19  0.23] | 8 |
| NCH SSS | **[-2.54 -1.02]** | **[-2.99 -0.38]** | 4 |
| max AMOC (Sv) | [-5.10  0.66] | [-7.75  2.61] | - |
| RMSE AMOC (Sv) | **[-1.59 -0.60]** | **[-2.69  -0.05]** | 4 |
| correl AMOC | **[0.01 0.11]** | [-0.02  0.20] | 14 |
| RMSE temp profile (ºC) | **[-1.31 -0.39]** | **[-1.57 -0.12]** | 4 |
| correl temp profile | [-0.15  0.16 ] | [-0.16  0.20] | - |
| RMSE salt profile | **[-0.41 -0.09]** | [-0.68   0.04] | 8 |
| correl salt profile | **[0.01 0.04]** | [-0.00  0.07] | 6 |
| RMSE density profile (kg m-3 ) | **[-0.26 -0.06]** | [-0.48  0.03 ] | 8 |
| correl density profile | **[0.01 0.04 ]** | [-0.00  0.06] | 7 |
| max MLD (m) | [-310.52 1530.64] | [-703.04 1858.65] | - |
| max SPG (Sv) | [-11.17  9.94] | [-17.10 14.09] | - |

**Table T1**: The first column indicates the single numeric metrics analyzed (units in parenthesis). The second column shows the 95% CI of the differences in means between the HR and LR ensembles, calculated from a distribution of bootstrapping samples with repetition. The size of the samples coincides with the total size of their respective ensembles. The third column is analogous to the second one but in this case the size of the LR samples coincides with the total size of the HR ensemble. The fourth column indicates the minimum size of the LR samples in the bootstrapping required to obtain a CI not containing the value 0. Text in bold indicates when this is the case.

Lines 255-256: Is the correlation between MLD (deep convection) and AMOC (assuming you are referring to vertical overturning strength) in models in Martin-Martinez et al.? This relationship is much less clear in observations (see Li et al., 2021 for discussion in relation to the OSNAP observing system).

Yes, Martin-Martinez et al. (2024) show a correlation between March MLDs in the Labrador/Irminger Sea and the AMOC streamfunction at different time lags and latitudes for the

EC-Earth-3P model across different resolutions (Fig. 7 in that manuscript; link to preprint: https://egusphere.copernicus.org/preprints/2024/egusphere-2024-3625/egusphere-2024-3625.pdf). Results from a multimodel study (Li et al., 2019) also show correlations between March Labrador Sea MLDs and AMOC strength at different latitudes (Fig. 11 in that study), as well as between winter-averaged Labrador Sea Water density and AMOC at different latitudes (Fig. 9 in that study). Similarly, Menary et al. (2020) show a lagged correlation between SPNA densities (between 500-1500 m depth) and AMOC strength at 45ºN (Fig. 3a in that manuscript) in the HadGEM3-GC3.1-MM model.

Li et al. (2021)'s study supports a relationship between density anomalies in the SPNA and AMOC variability, which we believe does not contradict our original statement.

We have extended the references in the old text to include that of Li et al. (2019) and added a remark on the interesting results by Li et al. (2021).

New text:
"MLD is generally used as a proxy for deep water convection, it is correlated to AMOC strength (Martin-Martinez et al., in review; Li et al., 2019), and in the North Atlantic it achieves its maximum in March. We note that results from a recent observational study suggest that AMOC variability depends on combined density anomalies from different areas of the SPNA rather than from one location alone (Li et al., 2021), such that MLD analyses in the SPNA should not be limited to one single region (e.g. the central Labrador Sea).

Old text:
"MLD is generally used as a proxy for deep water convection, it is correlated to AMOC strength (Martin-Martinez et al., in review), and in the North Atlantic achieves its maximum in March"

Lines 264-265: Given that EN4.2.2 is too coarse & has insufficient observational data (Argo etc.) to resolve subpolar boundary currents, can you really assess if the convection region along the Irminger Sea western boundary current is better represented in the HR ensemble?

To improve our picture of the representation of mixed layers in the Irminger Sea and in the SPNA in general, we have added the finer resolution ARMOR3D dataset to our analysis (please, see the "General Comments" section). We agree with the reviewer that the 1º EN4 dataset cannot resolve all the details of the East Greenland current, but it can characterize it broadly. We have tested the sensitivity of the EN4 MLDs to the observational period (Argo vs pre-Argo) (Fig. F4) and the convection patterns are similar in both cases. However, we observe that the western IS and the LS convection areas are less connected for the 2000-2014 (Argo) period compared to the 1980-1999 (pre-Argo) period in EN4, which could be related to natural variability but also to the more restricted availability of EN4 profiles prior the year 1999. We have edited our old text accordingly (see below).

Note: this new figure will be added to the manuscript as a supplementary figure.

New text:
"Notice too, that the convection area along the East Greenland current, in the western IS, is also deeper in the HR-HIST ensemble with respect to LR-HIST, better resembling the EN4 pattern."

Old text:
"Notice too, that the convection area along the East Greenland current, in the western IS, is also deeper and better connected with the mixed layers of the LS in the HR-HIST ensemble with respect to LR-HIST, better resembling the EN4 pattern."

[Figure]

**Figure F4.** EN4 March MLD (in m) for the (a) 1980-2014, (b) 1980-1999 (pre-Argo) and (c) 2000-2014 periods.

Figure 9: Is the AMOC vertical overturning stream function in the HR ensemble statistically significantly different from the LR ensemble, given the wide range of AMOC mean states shown in Figure 9 (LR ensemble panels).

This remark has been addressed in a previous point (see bootstrapping analysis above).

Lines 293-294: When comparing model results to the RAPID-MOCHA array is the 2004-2022 period used or the 2004-2014 period overlapping the end of the historical simulations?

In Fig. 11 we compared the 1980-2014 period in models and ORAS5m reanalysis with the 2004-2022 period in RAPID. We have now produced an additional plot to test the sensitivity of the results to the choice of the RAPID period (Fig. F5). In this new figure the RAPID and GLORYS12 data cover the intervals 2004-2014 and 1993-2014, respectively, overlapping the end of the historical simulations. Figure F5 will replace old Fig. 11 in the manuscript. However, differences with respect to the original analysis do not appear significant.

[Figure]

**Figure F5.** (a) Climatological AMOC profile at 26.5º N (in Sv). (b) Pearson correlation coefficient (horizontal axis) and Root Mean Square Error (RMSE; in Sv) (vertical axis) of AMOC profiles at 26.5º N against RAPID, both estimated across the vertical dimension. In both subplots, models and ORAS5m reanalysis data correspond to the interval 1980–2014, RAPID observations are averaged over the period April 2004 – December 2014, and GLORYS12 reanalysis over the interval 1993-2014.

Lines 305-306: Suggest being more specific on the differences in the methodological approach; was the RAPID overturning stream function calculated using the METRIC package or are you comparing the model 'truth' to the RAPID calculation applied to observations?

We compare the model "truth" to RAPID calculations. More details on the methodological approach have been added to the text. Please, see below.

New text (replacing lines 296-306):
" The HR-HIST mean profile shows a particularly good fit with the RAPID array one above ~1000 m, although, in general, the AMOC streamfunction is too shallow both for LR-HIST and HR-HIST. Some differences between models and observations might stem from the methodologies used to derive the AMOC profiles. While in models the AMOC streamfunction is obtained by integrating model velocities, which are simulated in every grid point, this approach is not possible with observations, since direct velocity measurements are scarce. RAPID data combines

measurements of four separate AMOC components. The first is the Florida Current transport, which has been inferred from cable voltage measurements west of the Bahamas since 1982 (Larsen and Sandford, 1985). The second one is the western boundary wedge transport, which measures elements of the Antilles and the deep western boundary currents using current meters west of 76.75ºW to Abaco Island. The third term is the near-surface AMOC Ekman transport, calculated itself from wind stress reanalysis data. The fourth term is the upper mid-ocean return transport, derived for the region east of 76.75º W from density profiles using zonal gradient of dynamic heights (Roberts et al., 2013; Danabasoglu et al., 2021; McCarthy et al., 2015). The calculation of this gradient in RAPID makes use of a reference depth (4820 m), which represents a level-of-no-motion. Some studies report sensitivity of the estimated RAPID profile, particularly in the deep ocean, to the choice of this reference depth (Fig 3.2 in McCarthy et al., 2015; Fig. S3 in Roberts et al, 2013), which might explain some of the differences between the RAPID and model profiles in the deep ocean (Fig. 11). However, uncertainties related to the choice of a reference depth are within the range of the accuracy of the RAPID method, and uncertainties in deep transport are a current topic in the literature (McCarthy et al., 2015). A model-based study also suggests that estimating the AMOC via RAPID's physical assumptions could lead to an underestimation of up to 1.5 Sv in its mean value at ~900 m depth (Sinha et al., 2018) compared to its real strength, a result that is, however, not supported in a more recent study based on a different ocean model (Danabasoglu et al. 2021)."

Old text (lines 296-306):
"The HR-HIST mean profile shows a particularly good fit with the RAPID array one above ~1000 m, although, in general, the AMOC is too shallow both for LR-HIST and HR-HIST. This is in part due to differences in the methodological approach (Danabasoglu et al., 2021)."

Lines 328-344: When discussing the Gulf Stream path, the predominant focus is on the location of separation with only limited commentary on the current's structure. Suggest undertaking a more detailed evaluation of the Gulf Stream structure, including exploring its eastward penetration using surface eddy kinetic energy following Xu and Chassignet (2017). Alternatively, an observational product such as COPERNICUS-GLOBCURRENT could be used to validate the model surface current velocities.

Thanks for this remark. We would prefer performing new diagnostics based on the already available/processed variables, because processing an additional variable (such as eddy kinetic energy) for the 44 models analyzed in this study is an arduous task due to ESMValTool constraints, which requires extremely strict cmorization of the data. We believe however, that this analysis is not indispensable for the completeness of our study.
Additionally, we would like to note that in the original manuscript we have analyzed the location of separation of the Gulf Stream from the coast but also the NAC structure through the barotropic streamfunction (Figs. 12 and 13). We have validated the position of the NAC using absolute dynamic topography (old lines 318-321). Besides, in (old) lines 345-350 we have associated a less zonal NAC structure at high resolution with a reduced bias in the CNA region.

Lines 353-356: This is an interesting point, suggest exploring the relationship between interior dense water formation and SPG dynamics further as a potential explanation for ensemble spread / differences.

We have added a scatterplot showing the correlation between MLD and SPG strength in the different models (Fig. F6), and discussed the results (the discussion is included in the updated manuscript).

[Figure]

**Figure F6.** Scatterplot of max. SPG strength (in Sv) vs max. MLD (in m), both referred to the LIS box (shown in Fig. 8). Pearson correlation coefficient and p-value are shown next to the fit line. Horizontal dashed and dot-dashed lines show EN4 and ARMOR3D observation-based values, respectively. Vertical dashed and dot-dashed lines show ORAS5m and GLORYS12 reanalysis values, respectively.

**Discussion and Conclusions**

Lines 388-391: The current discussion of Labrador Sea biases should be revised to cite more recent perspectives (e.g., review by Jackson et al., 2023; Li et al., 2023; Rühs et al. 2021).

The text has been modified to reflect the findings by the above-mentioned studies (see new discussion in the updated manuscript).

Lines 391-394: This is an interesting hypothesis to link the NAC and LS salinity biases. Suggest reading Kostov et al. (2023, 2024) on the connection between the NAC and LS convection to extend these ideas further in the manuscript as suggested in General Comments.

A discussion has been included on the results by Kostov et al. (2023, 2024) in relation to our findings (see updated manuscript).

Lines 408-412: Given the limitations you have identified with the EN4.2.2 product, why not use a 'purpose-built' global mixed layer climatology, such as the LOPS-IFREMER MLD product (https://doi.org/10.17882/98226)? Note, this is still a coarse resolution product, so using ARMOR3D may be more appropriate to compare to HR models.

Done, the ARMOR3D MLDs have been added to the analysis (please, refer to the "General Comments" section).

Lines 467-470: Suggest revising this summary to focus on why the HR ensemble shows improvements compared to ocean observations, and highlight next steps forward in coupled climate modelling; for example, what are the implications of this improved representation of the ocean mean state in mesoscale-resolving ocean models on the atmosphere and societally relevant indicators? This invokes a wider question of whether the improvements in the North Atlantic mean state are sufficient to justify the additional computational cost, and to what extent the mean state determines the ocean's future trajectory in coupled models.

Done, the text has been modified to add a discussion of all these points (see updated manuscript).

---

## Referee Report (RR1)

**Summary**

I would like to thank the authors for having taken all my comments into account and for their considerable efforts in revising the manuscript. This revised version of the manuscript is much improved, especially through the additions of the new observational analysis & reanalysis products, and the more robust statistical assessment of the LR & HR ensembles.

The authors have resolved all my original concerns. Upon addressing my small number of minor comments below, I would recommend this manuscript be accepted for publication in Ocean Sciences.

**General Comments**

- **Manuscript Length:**

  Whilst I'm grateful to the authors for addressing the comments of Reviewer #2 and myself so thoroughly, I'm slightly concerned that the revised manuscript has increased significantly in length from 34 to 47 total pages. I would strongly recommend reviewing the manuscript for opportunities to consolidate the existing figures / text, whilst preserving the central messages. I'd like to highlight one possible approach to do this below (although this is simply a suggestion):
    - Distributing the findings of **3.6 Testing the significance of differences between ensembles** amongst the relevant sections earlier in the Results. As a reader, I felt that this statistical analysis would have been most useful when the diagnostics were first discussed. This would provide an opportunity to reduce the length of the text, and the summary table could be moved to Supplementary Information / Appendix A.
    - Similarly, the contents of **3.7 Characteristic features in the HR-HIST models** could be distributed between the previous Results sections (for quantitative comparisons – which would then be helpfully located near each large multi-panel figure) and the excellent Discussion & Conclusions section (for the more speculative discussion points).

- **3.8 Relations between dynamical and physical properties:**

  This is an interesting addition to the manuscript, which directly addresses my previous concerns regarding the originality of the study. My only concern is whether it is appropriate to use a composite of the LR-HIST and HR-HIST ensembles to explore the correlations between state variables, since earlier you highlight important differences between these ensembles. For example, might it be the case that the relationship between the SPG strength and the maximum overturning at 26.5N is different between the LR & HR ensembles? Hirschi et al. (2020) highlighted that, at HR, the more realistic SPG circulation projects more strongly onto the diapycnal rather than vertical overturning at subpolar latitudes

(horizontal circulation across sloping isopycnals), whereas LR models exhibit a more classical vertical "conveyor" like overturning cell.

Suggest commenting that the relationships between dynamical & physical properties may hence also depend on horizontal resolution (although a larger HR ensemble would be needed to perform this analysis). This may be a case of bringing some of the excellent discussion on Lines 765-774 forward in the text.

**Specific Comments**

**Methods**

**Use of GLORYS12v1:** I'm grateful to the authors for including the mesoscale eddy resolving GLORYS12v1 reanalysis product in their model evaluation, however, I have some concerns regarding the implications of using the regridded outputs for the purpose of calculating meridional overturning and barotropic stream functions. Given that GLORYS12v1 is originally simulated on a curvilinear ORCA12 grid, the use of linear interpolation to regrid the model velocity will inevitably introduce multiple sources of error into a volume transport calculation (including estimation of grid cell areas, interpolation errors at high-latitudes and considerations of bottom topography). This likely also applies to the ORAS5m outputs shown. Given that GLORYS12v1 is available on its original NEMO model grid, it would at least be worth briefly emphasising to readers the limitations of using a regridded velocity field, although I suspect this will not alter your qualitative results.

**Results**

**Figure 5**: Suggest using a different colour / marker to identify the LR-HIST mean profile than dark grey, this is quite difficult to distinguish from the large number of LR-HIST ensemble member profiles (Fig. 5b especially) - although I do recognise the need to relate these visually.

**Lines 392-419**: Suggest revising this discussion on the methodological concerns regarding comparisons between models and observations at the RAPID array. I would highlight two important points that are missing:

1. There is an existing approach to compare ocean models with RAPID observations in an equivalent manner, accounting for the four separate AMOC components: using the Meridional ovErTurning ciRculation diagnostIC (**METRIC**) package originally developed by Castrucio et al. (https://github.com/AMOCcommunity/metric).
2. The chosen treatment of the net throughflow across the RAPID 26.5N section can be a important control on the magnitude of the depth-space overturning calculated in models. It would be useful to inform readers whether a uniform

volume transport compensation term was applied to each of the models (as done in observations) prior to the stream function calculation or if these diagnostics include the net throughflow across the section. This would be especially relevant for the reanalysis data used, since the regridded velocity field does not properly represent bathymetry.

---

## Author Response (AR2)

The authors sincerely acknowledge both reviewers for their comments, as well as for the time and effort spent in revising our responses.

Responses are in blue and changes in the manuscript in green.

**Responses to referee#1**

**Summary**

I would like to thank the authors for having taken all my comments into account and for their considerable efforts in revising the manuscript. This revised version of the manuscript is much improved, especially through the additions of the new observational analysis & reanalysis products, and the more robust statistical assessment of the LR & HR ensembles.

The authors have resolved all my original concerns. Upon addressing my small number of minor comments below, I would recommend this manuscript be accepted for publication in Ocean Sciences.

**General Comments**

**§ Manuscript Length:** Whilst I'm grateful to the authors for addressing the comments of Reviewer #2 and myself so thoroughly, I'm slightly concerned that the revised manuscript has increased significantly in length from 34 to 47 total pages. I would strongly recommend reviewing the manuscript for opportunities to consolidate the existing figures / text, whilst preserving the central messages.

The length of the manuscript has been shortened. The following changes have been made to shorten the text:

1- The discussion about MLD values in the LS (old lines 339-347 + old lines 698-722) has now been shortened (now lines 346-367).

2- Table 2 has been moved to Appendix A and renamed as Table A1.

3- Fig. 16 has been moved to Appendix A (as it was in the original draft, before the revisions) and renamed as Fig. A3. Old Fig. A3 is now Fig. A4.

4- The discussion about the RAPID method (old lines 392-413) has been shortened.

New text replacing old lines 392-413 (now at new lines 424-431):

"Some differences between models and observations might stem from the methodologies used to derive the AMOC profiles. While in models the AMOC streamfunction is obtained by integrating model velocities, which are simulated at every grid point, this approach is not possible with observations, since direct velocity measurements are scarce. The calculation of the upper mid-ocean return transport in RAPID is based on the zonal gradient of dynamic heights from density profiles, which makes use of a reference depth (4820 m), representing a level-of-no-motion (Roberts et al., 2013; McCarthy et al., 2015; Danabasoglu et al., 2021). Some studies report sensitivity of the estimated RAPID profile, particularly in the deep ocean, to the choice of this reference depth (Fig 3.2 in McCarthy et al., 2015; Fig. S3 in Roberts et al, 2013), which might explain some of the differences between the RAPID and model profiles in the deep ocean (Fig. 11)."

5- Old lines 775-780 have been removed following a suggestion by another reviewer.

I'd like to highlight one possible approach to do this below (although this is simply a suggestion):

o Distributing the findings of 3.6 Testing the significance of differences between ensembles amongst the relevant sections earlier in the Results. As a reader, I felt that this statistical analysis would have been most useful when the diagnostics were first discussed. This would provide an opportunity to reduce the length of the text, and the summary table could be moved to Supplementary Information / Appendix A.

The summary table has been moved to Appendix A following the reviewer's suggestion. Regarding redistributing section 3.6 into the previous sections, we believe it would probably not contribute to shorten the length of the manuscript, since, for example, in section 3.6, the significance of changes in the SST and SSS biases between ensembles are treated jointly (and thus occupy less space), meanwhile in the previous analyses they are split between sect. 3.1 and sect. 3.2.

o Similarly, the contents of 3.7 Characteristic features in the HR-HIST models could be distributed between the previous Results sections (for quantitative comparisons – which would then be helpfully located near each large multi-panel figure) and the excellent Discussion & Conclusions section (for the more speculative discussion points).

We believe it is useful to have a section where the different metrics of a specific model are discussed altogether, it provides the opportunity to relate them to each other (which is difficult in separate sections analysing separate metrics). For this reason, we would prefer to keep sect. 3.7 as a whole.

Importantly, we would also like to note that the length of the manuscript reflects a comprehensive analysis, including several multimodel figures and mesoscale-resolving models.

**§ 3.8 Relations between dynamical and physical properties:**

This is an interesting addition to the manuscript, which directly addresses my previous concerns regarding the originality of the study. My only concern is whether it is appropriate to use a composite of the LR-HIST and HR-HIST ensembles to explore the correlations between state variables, since earlier you highlight important differences between these ensembles. For example, might it be the case that the relationship between the SPG strength and the maximum overturning at 26.5N is different between the LR & HR ensembles? Hirschi et al. (2020) highlighted that, at HR, the more realistic SPG circulation projects more strongly onto the diapycnal rather than vertical overturning at subpolar latitudes (horizontal circulation across sloping isopycnals), whereas LR models exhibit a more classical vertical "conveyor" like overturning cell.

Suggest commenting that the relationships between dynamical & physical properties may hence also depend on horizontal resolution (although a larger HR ensemble would be needed to perform this analysis). This may be a case of bringing some of the excellent discussion on Lines 765-774 forward in the text.

We have added the following paragraph at former line 612 (now line 663):

"Horizontal resolution might play a role in the representation of the relationships between the different dynamical and physical properties in the North Atlantic through differences in model dynamics. For example, work by Katsman et al. (2018) shows differences in deep water sinking mechanisms at mesoscale-permitting resolutions (see Sect. 4 for further details), which might affect relationships involving overturning. In this study we cannot properly assess whether such relationships change with resolution given the limited size of the HR-HIST ensemble, but future studies might be able to address it as new mesoscale-resolving simulations become available".

**Specific Comments**

**Methods**

**Use of GLORYS12v1:** I'm grateful to the authors for including the mesoscale eddy resolving GLORYS12v1 reanalysis product in their model evaluation, however, I have some concerns regarding the implications of using the regridded outputs for the purpose of calculating meridional overturning and barotropic stream functions. Given that GLORYS12v1 is originally simulated on a curvilinear ORCA12 grid, the use of linear interpolation to regrid the model velocity will inevitably introduce multiple sources of error into a volume transport calculation (including estimation of grid cell areas, interpolation errors at high-latitudes and considerations of bottom topography). This likely also applies to the ORAS5m outputs shown. Given that GLORYS12v1 is available on its original NEMO model grid, it would at least be worth briefly

emphasising to readers the limitations of using a regridded velocity field, although I suspect this will not alter your qualitative results.

We have now commented on the limitations of the GLORYS12v1 overturning and barotropic streamfunction data (at old line 202; now line 207). Regarding ORAS5m data, these were available in the original grid, which was the one used in our calculations, and thus those data do not present the same limitations as our GLORYS12v1 dataset.

New text (at new line 207):

"As a second reference, in addition to ORAS5m, GLORYS12 reanalysis data at mesoscale eddy-resolving resolution (1/12º; period 1993–2014; Lellouche et al., 2018) are also employed to add robustness to our analyses. In the case of ORAS5m, the overturning and barotropic streamfunctions are calculated from velocity fields in the original reanalysis ocean model grid, while for GLORYS12, they are calculated based on the regridded velocity fields available from Copernicus. We note that the use of regridded fields for volume transport calculations might introduce some errors related to, for example, the estimates of grid cell areas. However, the GLORYS12 data still constitute a valuable qualitative reanalysis reference."

Old text:

"As a second reference, in addition to ORAS5m, GLORYS12 reanalysis data at mesoscale eddy-resolving resolution (1/12º; period 1993–2014; Lellouche et al., 2018) are also employed to add robustness to our analyses. In the case of ORAS5m, the overturning and barotropic streamfunctions are calculated from velocity fields in the original reanalysis ocean model grid, while for GLORYS12, they are calculated based on the regridded velocity fields available from Copernicus."

**Results**

**Figure 5:** Suggest using a different colour / marker to identify the LR-HIST mean profile than dark grey, this is quite difficult to distinguish from the large number of LR-HIST ensemble member profiles (Fig. 5b especially) - although I do recognise the need to relate these visually.

We would prefer to keep the original colour scheme, which yields the best compromise we could achieve to provide visibility to the HR models against the benchmark of LR models (and still relate them visually). The final jpg figure will have a much improved resolution than it has now in the submitted pdf. The fact that the different profiles might not be so easy to distinguish from one another in the deep ocean has to do with the fact that there is significant overlapping at depth, but this would also be the case with a different colour scheme. More importantly, note that

differences between the HR and the LR ensemble means occur in the upper ocean, and those are clear in the current figure.

**Lines 392-419:** Suggest revising this discussion on the methodological concerns regarding comparisons between models and observations at the RAPID array. I would highlight two important points that are missing:

1. There is an existing approach to compare ocean models with RAPID observations in an equivalent manner, accounting for the four separate AMOC components: using the Meridional ovErTurning ciRculation diagnostIC (METRIC) package originally developed by Castrucio et al. (https://github.com/AMOCcommunity/metric).

2. The chosen treatment of the net throughflow across the RAPID 26.5N section can be a important control on the magnitude of the depth-space overturning calculated in models. It would be useful to inform readers whether a uniform volume transport compensation term was applied to each of the models (as done in observations) prior to the stream function calculation or if these diagnostics include the net throughflow across the section. This would be especially relevant for the reanalysis data used, since the regridded velocity field does not properly represent bathymetry.

We have added the following text at old line 417 (now at lines 457-462):

"Model and reanalysis overturning profiles at 26.5º N in our study are computed from full velocities and do not include the application of a uniform volume transport compensation term. This might be relevant for the GLORYS12 overturning profile, which is based on regridded velocities (see Sect. 2.3). We would like to note the existence of the software package Meridional ovErTurning ciRculation diagnostIC (METRIC), for calculating RAPID observations-equivalent AMOC diagnostics in models using different model output variables (Castruccio, 2021; Danabasoglu et al., 2021)".

New citation in references:

Castruccio, F. S.: NCAR/metric: metric v0.1., Zenodo [code], https://doi.org/10.5281/zenodo.4708277, 2021.

We have also moved old lines 417-419 to new lines 422-424.

**Responses to referee#2**

Thank you for the revised manuscript "The North Atlantic mean state in mesoscale eddy-resolving coupled models: a multimodel study". The paper is well written and much improved on the previous version. Though there are no major new findings, it is a useful reference for impacts of resolution, particularly for mesoscale eddy-resolving, which hasn't been covered by many studies. I have a few minor comments.

L53 'it has' -> have

Here 'it' refers to the 'sinking', which is singular, not to the 'deep waters'. For this reason, we would prefer to keep the singular form of the verb.

New text (at new line 52):

"Sinking of deep waters forming the AMOC return flow occurs at the boundaries of the subpolar gyre (SPG) and has been associated with densification of waters along the boundary current (Katsman et al., 2018; Straneo, 2006; Spall and Pickart, 2001)."

Old text:

"Sinking of deep waters forming the AMOC return flow occurs at the boundaries of the subpolar gyre (SPG) and it has been associated with densification of waters along the boundary current (Katsman et al., 2018; Straneo, 2006; Spall and Pickart, 2001)."

L279-288 Do models also generally have the same SST and SSS biases in regions outside the LS?

In the regions outside the LS, i.e. in the CNA and NCH regions, models present biases of the same sign, positive in the NCH region and negative in the CNA region. In this sense it is not possible to separate models according to the sign of their biases.

L340 'in consistency' -> consistent

Changed (now line 347).

L341-345 I think the discussion about MLD datasets in the discussion should be moved here. At the moment you start by saying that HR MLD is in better agreement with EN4 than LR. I think

you need to say that the observations of MLD vary a lot (probably because of differences in methodology – this could probably be shortened). Then either say that you can't assess because of the wide range of observations, or make an argument than EN4 is most like the models (because you're using the same methodology) and then compare to the models.

The discussion about the MLD datasets in Section 4 has been moved here. Besides, details about the different methodologies have been shortened. The conclusions that we have been able to extract are also described in the new text below.

New text in the results section (replacing old lines 339-347; now at lines 346-367):

"In the LS, the multi-model mean of HR-HIST shows a deeper (although not significantly deeper; see Sect. 3.6) mixed layer than the LR-HIST mean (Fig. 8), consistent with a relatively weaker density stratification (Sect. 3.2). If we check the individual models (Fig. 7) we note that all the HR-HIST models show deep mixed layers in the LS, while ~25 % of the LR-HIST models show little or no convection. LS mixed layers in the HR-HIST mean (1800–2000 m) are closer to EN4 estimates (2000–2200 m), whereas in ARMOR3D (1000–1200 m) they are in the same range as in the LR-HIST mean (1000–1200 m). For the overlapping time interval of ARMOR3D and EN4 (i.e., for 1993–2014), EN4 values for the LS are still larger (1800–2000 m; not shown) compared to ARMOR3D. The wide range in the observation-derived estimates for the LS in our analyses leads us to review the literature for observational studies. Work by Holte et al. (2017) based on individual Argo density profiles shows mixed layers down to 1400–1800 m in the LS for the 2000–2016 period. Time-varying estimates of winter maximum MLDs in the LS obtained from Argo floats, the AR7W line, and moored measurements, suggest values mostly around 1100–1500 m in the 2002–2015 interval (Yashayaev and Loder, 2016), showing an intensification in recent years, with a record value of 2100 m in 2016 (Yashayaev and Loder, 2017). Therefore, the MLD values in our HR-HIST (1800–2000 m) ensemble mean are slightly too large compared to observational studies, and the LR-HIST (1000–1200 m) ensemble mean values are slightly too shallow. The differences in the MLD estimates obtained across the different studies might arise from the different temporal and spatial characteristics of the profile data, differences in the time intervals analysed, and from the different methodological approaches employed. The yearly estimates by Yashayaev and Loder (2016; 2017) are winter maximum values of "aggregate" maximum convection depths, defined as the 75[th] percentile of the depth of the base of the pycnostad in the set of available individual LS profiles at each time. Holte et al. (2017)'s MLD estimates (shown in their Fig. 3a) correspond to individual Argo profiles and are obtained with a density algorithm (Holte and Talley, 2009) that uses a combination of methods and elements (including temperature and density threshold methods, gradient methods, estimates of thermocline linear fits, etc). In our study, instead, MLD values are a climatology of March MLD monthly means obtained from gridded temperature and salinity data through a density threshold method."

New lines in the discussion (replacing old lines 698-722; now lines 781-785):

"In the LS the wide range of observation-derived MLD estimates, with values of 1000–1200 m in ARMOR3D and 2000–2200 m in EN4, makes model assessment challenging. Analyses from additional observational studies show values between 1100–1500 m (Yashayaev and Loder, 2016) and down to 1400–1800 m (Holte et al., 2017) in the LS, suggesting that HR-HIST mean values for the LS (1800–2000 m) might be slightly too deep, and LR-HIST mean values (1000–1200 m) slightly too shallow."

L355 For this 'distinct stripe', is there any evidence whether it's real or an artifact of the method/data? Is it discussed in other studies?

To our knowledge, it is the first time that a 3D temperature and salinity dataset at the resolution of the ARMOR3D dataset (1/4º) is used to derive MLD estimates. Typically, coarser resolutions of 1º are used, which cannot resolve that feature. The 'distinct stripe' is most probably linked to the resolution of the observational dataset. We have modified the text as follows:

New text (at new lines 383-386):

"A remarkable feature in the ARMOR3D dataset is a distinct stripe of deep mixing (1200–1400 m deep) attached to the shelf along the East Greenland Current, which is also slightly visible in some of the individual HR-HIST models (e.g. HadGEM3-GC31-HH; Fig. 7) and might be absent in EN4 due to its coarser resolution."

Old text:

"A remarkable feature in the ARMOR3D dataset is a distinct stripe of deep mixing (1200–1400 m deep) attached to the shelf along the East Greenland Current, which is also slightly visible in some of the individual HR-HIST models (e.g. HadGEM3-GC31-HH; Fig. 7)."

Fig 12 I know from looking at the BSF in CMIP6 models that the way they are often calculated means that there is a large offset from integrating from the Southern Ocean northwards. You should comment on this somewhere and how you removed this offset since it affect the zero line.

BSF model data in our analyses were directly downloaded from the Earth System Grid Federation portal (ESGF; we did not perform the calculations). Some of the ESGF CMIP6 models in the LR-HIST ensemble presented unrealistic values for the BSF symptomatic of a problem in the integration and/or of differences in the integration approaches. Those models were discarded from the analyses. We have added a figure below (Fig. W) that we produced during the selection of the BSF models, which includes all the models that were downloaded

from the ESGF nodes (also the ones showing problems). That figure is just a snapshot from year 1980 but it shows which were the models that presented problems/differences. There is no evidence of any significant offset in the calculation of the global BSF (including in the Southern Ocean area) in the models kept in our analysis (Fig. W; Fig. 12). Also please note that the CMIP6 models kept in our analysis present a realistic structure for the BSF in the North Atlantic, consistent for example with the SST and SSS biases in the Gulf Stream and CNA area as well as with the AVISO SSH 0 contour line (Fig. 12). A note commenting on these points has been added to the text.

New text (at new lines 148-153):

"Some of the LR-HIST models, as downloaded from the Earth System Grid Federation ESGF data portal, present unrealistic values for the barotropic streamfunction (BSF), reflecting problems in the integration and/or different integration approaches (not shown). Such models have been discarded from our analysis. No significant offset in the BSF (including in the Southern Ocean region) is observed between the models kept in our BSF analysis and they all present a realistic BSF structure in the North Atlantic. Differences originating from different integration assumptions are thus expected to be small in our restricted model sample."

[Figure]

**Figure W**. BSF in models available from ESGF (year 1980).

L573-576 Fig 14a shows a huge amount of vertical scatter and, by eye, does not look significant to me. Could you please check the correlation and p value for this relationship? Also you quote a correlation from another study. Comparing correlations is pretty meaningless unless you take into account the number of data points. P values are more meaningful to compare.

The exact p-value associated with the correlation value r=0.47 in our calculations is p=0.0019. We observe that models with a weak AMOC tend to have weak convection in the Labrador Sea,

and the other way round. As the reviewer points out, there is some vertical scatter, which is reflected in a moderate correlation value of only r=0.47. The study by Li et al. (2019) does not present the p-values associated with the correlation values. Their correlation values are based on only 5 different data points, so they might be less significant than ours. Please note though that we relate the differences in the r values between the two studies to the smaller size of the model ensemble in Li et al. (2019).

L577 'in consistence' -> consistent

Changed (now line 624).

Section 3.8 It would also be good somewhere to discuss observational agreement – ie that models can have good agreement with multiple metrics at once. E.g. Models with no LS SSS bias can have no AMOC bias etc. A previous study (Danabasoglu, 2014 https://doi.org/10.1016/j.ocemod.2013.10.005 ) found that although there was a correlation between AMOC and MLD, that models could only agree with one or the other observational constraint.

We have added the following text at the end of old line 581:

New text (at new lines 628-632):

"Note that although the correlations described here are consistent, this consistency does not always translate into full observational fidelity, meaning that some models may agree with observational constraints for one metric but not for the other. For example, whereas several models with max. AMOC values within the range of observational estimates display max. MLD values which are also close to observations, some others exhibit MLD values that are too large compared to observations (Fig. 14a)."

We would like to note though that our study focused on the analysis of the performance of high-resolution models. The observational agreement of each specific high-resolution model has been discussed individually in Section 3.7: Characteristic features in the HR-HIST models as well as in the Discussion section of the manuscript.

L661 'overly weak NAC' – you haven't measured NAC strength only position. Why do you say it is weak?

Figs. 12 and 13 show a weaker barotropic streamfunction (BSF) in the MPI-ESM1-2-ER and EC-Earth3P-VHR models in the CNA box compared to ORAS5m and GLORYS12, as well as a

weaker eastward penetration of the NAC (as represented by the BSF) in these models. The text has now been modified to make these points more explicit.

New Text (at new lines 715-719):

"However, our bootstrapping analysis indicates that the reduction in the CNA surface biases is not statistically significant in our HR-HIST ensemble (Sect 3.6), as some of the HR-HIST models (MPI-ESM1-2-ER and EC-Earth3P-VHR) still present an overly weak NAC in the CNA (as represented in the BSF), with a reduced eastward penetration compared to reanalyses (Figs. 12, 13; Sect 3.7)."

Old Text:

"However, our bootstrapping analysis indicates that the reduction in the CNA surface biases is not statistically significant in our HR-HIST ensemble (Sect 3.6), as some of the HR-HIST models still present an overly weak NAC (MPI-ESM1-2-ER and EC-Earth3P-VHR; Sect 3.7)."

L689-693 Some studies also suggest that the NAC position is important for LS biases – see Jackson et al 2023, Marzocchi et al 2015, Treguier et al 2005 https://doi.org/10.1175/JPO2720.1 This seems relevant here since you've shown the NAC position improves at higher resolution.

Old lines 686-689 have been modified to reflect this.

New text (at new lines 744-748):

"Our study hints that LS and CNA biases might be actually related to each other through northward salinity/heat transport by the NAC, as supported by the correlations between the SSS (and SST) biases of those two regions. Note that northward transports depend both on NAC strength as well as path (Jackson et al., 2023). Studies such as Chang et al. (2020) and Roberts et al. (2019) report increased heat transport by the AMOC in mesoscale eddy-resolving models, further supporting the idea of increased northward transport as a potential origin for the LS biases."

Old text (old lines 686-689):

"Our study hints that LS and CNA biases might be actually related through northward salinity/heat transport by the NAC, as supported by the correlations between the SSS (and SST) biases of those two regions. Studies such as Chang et al. (2020) and Roberts et al. (2019) report

increased heat transport by the AMOC in mesoscale eddy-resolving models, further supporting the idea of increased northward transport as a potential origin for the LS biases."

L761 Should this be 'western continental boundaries'? I don't think it includes those in the east Atlantic.

We have checked Fig. 13, showing BSF multi-model means for HR-HIST and LR-HIST, and we observe that BSF contour lines are closer to each other in the HR-HIST mean compared to the LR-HIST mean, from the western tip of Iceland to the west, which indicates a narrower and locally stronger boundary current. Since these longitudes cover most of the SPG longitudinal extent, we would prefer keeping the original phrasing.

L775-780 This paragraph does not seem relevant to the mean state (the subject of the paper). There might be impacts on the variability/forced evolution, however there are many other studies that would also be relevant then.

This paragraph has now been removed from the text.